# Organismal metabolism regulates the expansion of oncogenic *PIK3CA* mutant clones in normal esophagus

Albert Herms [1,2,3,12], Bartomeu Colom [1,11,12], Gabriel Piedrafita [1,4,5], Argyro Kalogeropoulou[1], Ujjwal Banerjee[1], Charlotte King [1], Emilie Abby [1], Kasumi Murai[1], Irene Caseda[2,3], David Fernandez-Antoran [1,6], Swee Hoe Ong [1], Michael W. J. Hall [1], Christopher Bryant[1], Roshan K. Sood [1], Joanna C. Fowler [1], Albert Pol[2,3,7], Christian Frezza [8], Bart Vanhaesebroeck [9] & Philip H. Jones [1,10] ✉

Oncogenic *PIK3CA* mutations generate large clones in aging human esophagus. Here we investigate the behavior of *Pik3ca* mutant clones in the normal esophageal epithelium of transgenic mice. Expression of a heterozygous *Pik3ca^{H1047R}* mutation drives clonal expansion by tilting cell fate toward proliferation. CRISPR screening and inhibitor treatment of primary esophageal keratinocytes confirmed the PI3K–mTOR pathway increased mutant cell competitive fitness. The antidiabetic drug metformin reduced mutant cell advantage in vivo and in vitro. Conversely, metabolic conditions such as type 1 diabetes or diet-induced obesity enhanced the competitive fitness of *Pik3ca^{H1047R}* cells. Consistently, we found a higher density of *PIK3CA* gain-of-function mutations in the esophagus of individuals with high body mass index compared with those with normal weight. We conclude that the metabolic environment selectively influences the evolution of the normal epithelial mutational landscape. Clinically feasible interventions to even out signaling imbalances between wild-type and mutant cells may limit the expansion of oncogenic mutants in normal tissues.

The accumulation of somatic mutations is a hallmark of aging[1–9]. One of the most mutated tissues is the esophagus, which develops into a patchwork of mutant clones by middle age[1]. The most prevalent mutant genes are under strong positive selection, suggesting that they confer a proliferative advantage over wild-type cells[10]. However, little is known about the mechanisms driving mutant clone expansion. Understanding these processes may open opportunities for cancer prevention by limiting the number of transformable cells in normal tissues.

*PIK3CA* encodes the p110α catalytic subunit of phosphoinositide 3-kinase (PI3K) and is recurrently mutated in normal human esophagus[1,4,8]. PI3K is activated by insulin and other growth factors, and regulates multiple processes including cell proliferation, survival,

growth and metabolism, mainly through the activation of Akt–mTOR signaling[11,12]. Gain-of-function *PIK3CA* mutations, such as *PIK3CA^{H1047R}*, are recurrently found in tumors, including esophageal squamous cell carcinoma (ESCC), benign overgrowth syndromes and vascular malformations[12–15].

Analysis of published data identified 57 missense *PIK3CA* mutant clones in 17 cm² of histologically normal human esophageal epithelium[1]. Missense *PIK3CA* mutations had the second-highest average variant allele fraction (VAF) of 72 cancer-related genes analyzed after inactivating *NOTCH1* mutations, indicating that they form large clones (Fig. 1a,b)[16]. *PIK3CA* mutant clones were significantly enriched for pathogenic and/or gain of function (Path/GoF) missense mutations (MMs)

**Fig. 1 | *PIK3CA* mutant clones in human esophageal epithelium. a**, Schematic representation of mutant clones in an average 1 cm² of normal esophageal epithelium from a 48–51-year-old male donor from ref. 1. To generate the figure, a number of samples from the donor are randomly selected and the mutant clones detected are represented as circles and randomly distributed in space. **b**, The average VAF (top graph) and frequency (bottom graph) of missense mutations (MMs) detected more than once per gene, arranged from largest to smallest. *PIK3CA* highlighted in red. *n* = 844 samples from 9 donors. **c**, The distribution of *PIK3CA* MMs classified into pathogenic/gain of function (Path/GoF) or unknown/ no effect (Unkn/NE) (Methods). VAF distribution of synonymous mutations in all genes is also shown. Medians (red) and quartiles (gray lines) are represented. Two-tailed Mann–Whitney test. *n* = 23, 26 and 603 mutant clones, respectively, from 9 donors. **d**, The frequency of MM codons in the p110α protein. Path/GoF mutations are shown in red. *n* = 41 mutant clones from 9 donors. **e**, A comparison of the VAF distribution of *PIK3CA*^H1047R with other *PIK3CA* MMs classified as Path/ GoF or Unkn/NE. Medians (red) and quartiles (gray lines) are represented. Two-tailed Mann–Whitney test. *n* = 8, 15 and 26 mutant clones, respectively, from 9 donors.

(Methods), with 65% (37/57) Path/GoF mutations observed, well above the ~2% expected under neutrality (*P* = 1.22 × 10⁻⁴⁷, two-tailed binomial test)[1]. Clones with *PIK3CA* Path/GoF mutations were significantly larger compared with unknown/no effect *PIK3CA* mutations or synonymous mutations in all genes (Fig. 1c). In particular, *PIK3CA*^H1047R mutations generated larger clones than other *PIK3CA* missense mutants and were the most prevalent of all Path/GoF mutations (19%, 7/37) (Fig. 1d,e). These findings argue that activating *PIK3CA* mutants, particularly *PIK3CA*^H1047R, drive large clonal expansions in normal human esophagus.

These observations led us to investigate how *PIK3CA*^H1047R mutant progenitors colonize the normal esophagus. By combining lineage tracing and clustered regularly interspaced short palindromic repeats (CRISPR) screens, we demonstrate a role for the PI3K–mTOR and downstream pathways in determining the fitness advantage of mutant clones. We find that the antidiabetic drug metformin and metabolic conditions such as type 1 diabetes and diet-induced obesity or increased body mass index (BMI) modulate the expansion of *Pik3ca* mutant clones in the normal esophagus of mice and humans, respectively.

## Results

### Generation of inducible *Pik3ca*^H1047R-YFP knock-in mice

Most *PIK3CA* mutant clones present in normal human esophagus exhibit activating *PIK3CA*^H1047R mutations (Fig. 1d). To model these, we generated a conditional mouse strain, *Pik3ca*^fl-H1047R-T2A-YFP-NLS (*Pik3ca*^H1047R-YFP),

that allows heterozygous expression of the *Pik3ca*^H1047R mutation from the endogenous locus (Supplementary Note). After recombination mediated by *Cre* recombinase, the wild-type exon 20 of *Pik3ca* is excised and replaced by the mutant exon 20 encoding *Pik3ca*^H1047R (Extended Data Fig. 1a). The mutant protein is co-expressed with a nuclear localized yellow fluorescent protein (YFP) reporter linked to the C-terminus of the *Pik3ca*^H1047R protein by a T2A self-cleaving peptide[17]. *Pik3ca*^H1047R cells and their progeny express YFP, allowing lineage tracing of mutant clones (Methods and Extended Data Fig. 1b). Following T2A cleavage, a peptide remains at the C-terminus of *Pik3ca*^H1047R protein. The C-terminally extended p110α^H1047R protein still activated the PI3K–Akt pathway in vitro and in vivo (Extended Data Fig. 1c–f).

### *Pik3ca*^H1047R mutation drives esophageal tumorigenesis

Genomic alterations in the PI3K signaling axis, including activating mutations in *PIK3CA*, are frequent in esophageal cancer[15]. We therefore investigated whether heterozygous *Pik3ca*^H1047R mutations in normal mouse esophageal progenitors affected tumorigenesis. *Pik3ca*^H1047R-YFP/wt mice were crossed with the *Cre* recombinase line *AhCre*^ERT to generate *Cre-Pik3ca*^H1047R-YFP/wt mice. This strain allows induction of recombination in scattered progenitor cells in esophageal epithelium[18]. Aging experiments revealed that expression of *Pik3ca*^H1047R did not generate tumors, alter the gross appearance of the esophagus or impact mouse survival, as compared with *Cre-YFP*-induced controls (Extended Data Fig. 2a–c). These results are consistent with reports that

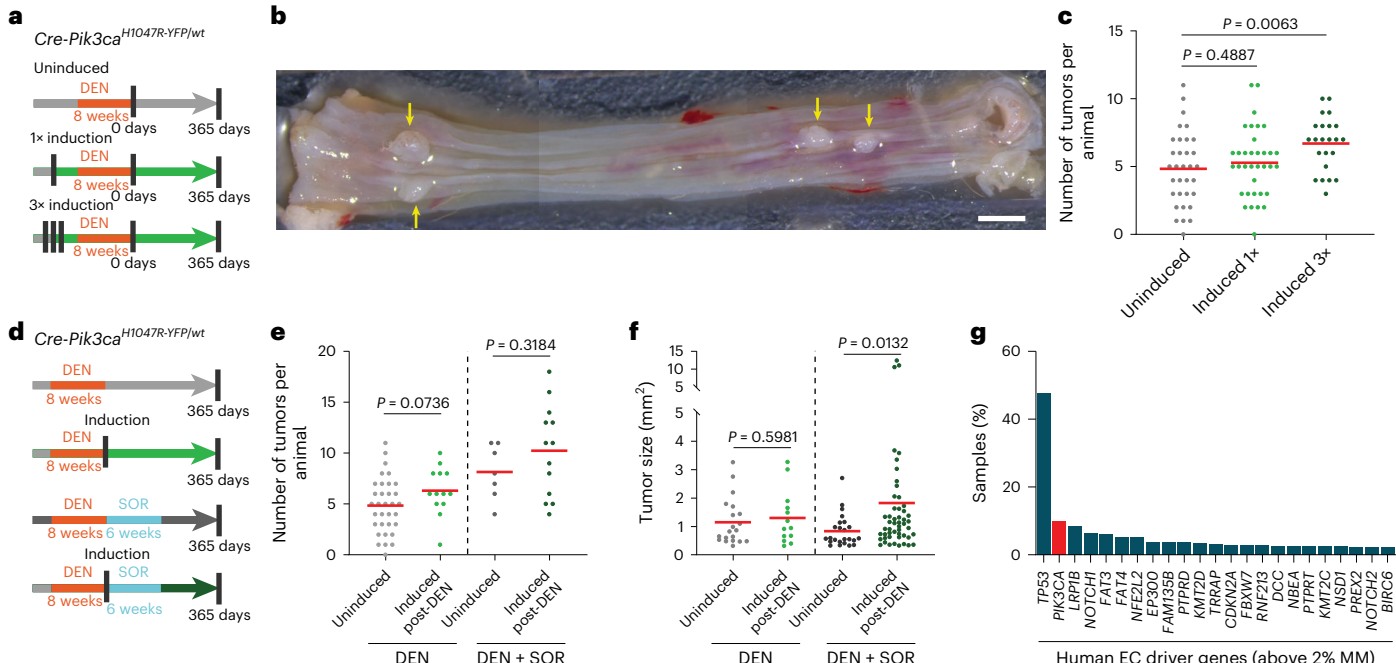

**Fig. 2 | Heterozygous *Pik3ca^H1047R* expression increases esophageal tumorigenesis. a**, Protocol 1: *Cre-Pik3ca^H1047R-YFP/wt* mice were induced one or three times with *Cre*-inducing drugs (Methods) and treated with DEN for 8 weeks starting 4 weeks post-induction. Uninduced mice were used as controls. Tissues were collected before exceeding the permitted humane endpoint or at 1 year after DEN. **b**, Typical DEN-treated esophagus opened and flattened epithelial side up showing four tumors (yellow arrows). Scale bar, 2 mm. **c**, The number of macroscopic esophageal tumors in DEN-treated mice, uninduced or induced one or three times before DEN treatment. Two-tailed Mann–Whitney test versus uninduced (*n* = 33, 35 and 23 animals, respectively). The red lines indicate average values. **d**, Protocol 2. *Cre-Pik3ca^H1047R-YFP/wt* mice were treated with DEN

for 8 weeks followed by *Cre* induction. A subgroup of animals was then treated with the tumor promoter sorafenib (SOR) for 6 weeks. Tissues were collected before exceeding the permitted humane endpoint (Methods) or 1 year post-DEN treatment. Control groups received all treatments but were uninduced. **e,f**, The number (**e**) and size (**f**) of macroscopic esophageal tumors. Two-tailed Mann–Whitney test (*n* = 33, 13, 7 and 13 mice, as they appear in the graph). The red lines indicate average values. **g**, The frequency of MMs for the indicated driver genes detected in human ESCCs from data collected from the TCGA and ICGC databases. Esophageal cancer (EC) driver genes were selected using the Intogen tool (https://www.intogen.org/search). Only driver genes with MM frequency >2% are shown.

heterozygous *Pik3ca^H1047R* expressed at physiological levels requires additional mutations to drive tumor formation[19–21].

We then investigated *Pik3ca^H1047R* in two models of mutagen-induced esophageal tumorigenesis. The first consisted of administration of diethylnitrosamine (DEN), a carcinogen present in tobacco smoke[22,23]. Induction of *Pik3ca^H1047R* did not affect mouse survival following DEN treatment but increased the number of macroscopic tumors compared with controls (Fig. 2a–c and Extended Data Fig. 2d,e). In a second protocol, DEN administration was followed by treatment with the tumor promoter sorafenib (DEN + SOR)[24]. *Pik3ca^H1047R* induction did not affect tumor density but significantly increased tumor size compared with controls (Fig. 2d–f). ESCCs were only detected in *Pik3ca^H1047R*-induced mice (0.032 ESCCs per mouse from 35 DEN-treated and 0.231 ESCCs per mouse from 13 DEN–SOR-treated induced animals; no ESCCs were found in 33 and 7 DEN- and DEN–SOR-treated uninduced animals, respectively) (Extended Data Fig. 2f–i).

These findings suggest a modest esophageal tumor-promoting role for *Pik3ca^H1047R* in mice, consistent with the observation of *PIK3CA* MMs in 10% of human ESCCs (Fig. 2g)[15].

### *Pik3ca^H1047R/wt* mutant outcompete wild-type cells

Given the oncogenic potential of *Pik3ca* mutant clones, we investigated the cellular mechanism underlying their expansion by lineage tracing in *Cre-Pik3ca^H1047R-YFP/wt* mice. *AhCre^ERT Rosa26^flYFP/wt* (henceforth termed *Cre-RYFP*) mice, which express a neutral YFP reporter after induction, were used as controls[25,26]. At time points up to 6 months, the entire esophageal epithelium was imaged in three dimensions, and the number and location of cells in mutant and wild-type clones were

recorded (Fig. 3a). *Pik3ca^H1047R/wt* mutant clones contained more cells than wild-type clones as early as 10 days post-induction, the difference increasing progressively over 6 months (Fig. 3b,c). We concluded that *Pik3ca^H1047R/wt* cells have a competitive advantage over their wild-type neighbors.

To confirm that these results were not due to genetic differences between mutant and control mouse strains, we crossed *Cre-Pik3ca^H1047R-YFP/wt* mice with the *Rosa26^Confetti/wt* strain[27] (Extended Data Fig. 3a). This triple cross allows tracking of *Pik3ca^H1047R/wt* mutant (red fluorescent protein (RFP)^+/YFP^+) and wild-type clones (RFP^+/YFP^−) simultaneously (Methods and Extended Data Fig. 3b,c). Consistent with our previous results, *Pik3ca^H1047R* mutant clones expanded more rapidly than wild types within the same esophagus (Extended Data Fig. 3d,e).

As in humans, once the tissue had been colonized by mutant cells, it appeared normal with no change in the basal layer cell density and no gross tissue disruption (Extended Data Figs. 2c and 3f). Collectively, our results indicated that *Pik3ca^H1047R/wt* mutant progenitors have a competitive advantage over their wild-type counterparts.

### *Pik3ca^H1047R* mutation tilts cell fate toward proliferation

We next investigated the mechanisms of *Pik3ca^H1047R* mutant cell advantage over wild-type cells. One possibility is that mutant cells divide at a faster rate. To investigate this, we induced *Cre-Pik3ca^H1047R-YFP/wt* mice and aged them for 3 months. One hour before tissue collection, animals were injected with 5-ethynyl-2′-deoxyuridine (EdU), which labels S-phase cells. Mutant and wild-type cells within the same esophagus showed a similar proportion of EdU^+ basal cells, arguing that the

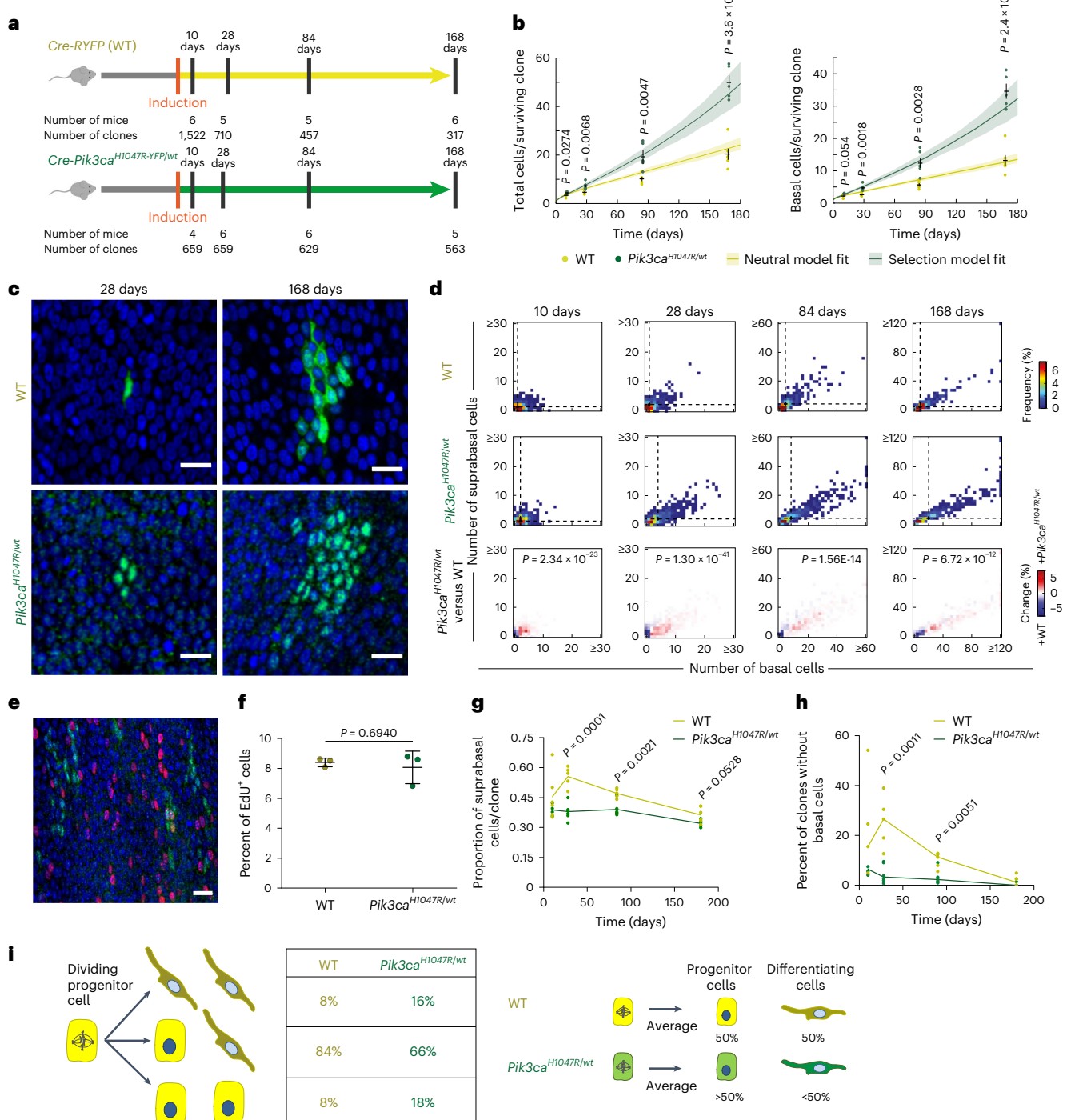

**Fig. 3 | A bias in *Pik3ca^{H1047R/wt}* cell fate drives mutant clone growth. a**, Protocol: *Cre-RYFP* and *Cre-Pik3ca^{H1047R-YFP/wt}* mice were induced, and wild-type and mutant clones were imaged at the time points shown. **b**, Total (basal + first suprabasal layer cells; left) and basal (right) cells per clone over time. Only clones with at least one basal cell were included. The dot indicates the average size of clones in each mouse. The lines and shaded areas represent the best-fitting model of clone size distributions and plausible intervals (Supplementary Note). Lines, mean ± s.e.m. Two-tailed unpaired *t*-test versus wild type (WT) (mice and clone numbers per time point indicated in **a**). **c**, Representative top-down confocal images of basal layer of wild-type (top) and *Pik3ca^{H1047R/wt}* mutant (bottom) clones at indicated time points. Clones, green; DAPI, blue. Scale bars, 20 μm. **d**, Heatmaps of clone size frequency; the number of basal and first suprabasal cells is shown for *Cre-RYFP* and *Cre-Pik3ca^{H1047R-YFP/wt}* animals. The black dots and dashed lines show the geometric median clone size. The graphs in the bottom row show differences between *Cre-Pik3ca^{H1047R-YFP/wt}* and *Cre-RYFP* animals for

each time point. Two-tailed 2D Kolmogorov–Smirnov test. **e**,**f**, Confocal image (**e**) and quantifications (**f**) of EdU+ cells (red) in wild-type or *Pik3ca^{H1047R-/wt}* mutant areas (green) from the basal layer of *Cre-Pik3ca^{H1047R-YFP/wt}* esophagus, 3 months post-induction. EdU was injected 1 h before tissue collection. DAPI is blue. Scale bar, 20 μm. Each dot corresponds to an animal, two-tailed ratio paired *t*-test (52,898 cells from 3 animals, including 3,514 *Pik3ca^{H1047R/wt}* cells). The bars indicate mean ± s.d. **g**,**h**, The proportion of suprabasal cells (**g**) and clones with no basal cells (**h**) in wild-type (*Cre-YFP*) or *Pik3ca^{H1047R/wt}* mutant clones at the indicated time points post-induction, from data in **d**. Each dot corresponds to one animal; the lines connect mean values. Two tailed unpaired *t*-test. **i**, Schematic illustration of WT and *Pik3ca^{H1047R/wt}* cell behavior in esophageal epithelium. Model predictions for the proportions of cell division outcomes for each genotype. *Pik3ca^{H1047R/wt}* cells produce an excess of progenitor over differentiating cells per average cell division, driving clonal expansion even with the rate of mutant cell division being the same as WT cells.

ratio of the length of S phase to total cell-cycle time, and the proportion of cycling basal cells, was not substantially altered by *Pik3ca*[H1047R] expression (Fig. 3e,f). Another potential mechanism of growth advantage is by promoting apoptosis of neighboring cells[28]. However, there was negligible detectable apoptosis in wild-type cells, whether adjacent to or distant from mutant clones in induced *Cre- Pik3ca*[H1047R-YFP/wt] animals (Extended Data Fig. 3g).

The advantage of *Pik3ca*[H1047R/wt] cells may also be explained by altered progenitor cell fate[16,18,29]. The mouse esophagus consists of layers of keratinocytes, with progenitor cells residing in the deepest, basal cell layer. Differentiating cells exit the cell cycle and leave the basal layer, migrating toward the epithelial surface where they are shed[25,26] (Extended Data Fig. 3h). Each progenitor division generates either two progenitor daughters, two nondividing differentiating cells or one cell of each type (Extended Data Fig. 3i)[25,26]. In wild-type cells, the probabilities of these outcomes are balanced across the progenitor population so that, per average cell division, equal numbers of progenitor and differentiating cells are generated, maintaining tissue homeostasis (Extended Data Fig. 3i). Even when mutant and wild-type cell division rates are similar, mutant populations can still expand by producing more progenitor than differentiating daughter cells per average cell division[18,26,29]. We observed that *Pik3ca*[H1047R/wt] clones contained a higher proportion of basal cells and fewer differentiated (suprabasal) cells compared with wild-type clones (Fig. 3d), suggesting that mutant progenitors generate a lower proportion of differentiating and more proliferating progeny than their wild-type equivalents. Mathematical modeling revealed that wild-type clones produced equal proportions of proliferating and differentiating cells (Supplementary Fig. 1a)[25,26]. However, *Pik3ca*[H1047R/wt] clone dynamics could only be explained if the average division generated more proliferating than differentiating progeny (Supplementary Fig. 1b and Supplementary Note). This simple model fits both the observed basal cell and total (basal plus suprabasal) clone size distributions (Fig. 3b, Extended Data Fig. 3e and Supplementary Fig. 1c) and infers two further features of *Pik3ca*[H1047R/wt] clones, a decreased proportion of suprabasal cells per clone (Supplementary Fig. 1d) and a reduction in the fraction of fully differentiated clones lacking any basal cells (Supplementary Fig. 1d and Supplementary Note). These predictions were both confirmed experimentally (Fig. 3g,h). We conclude that the increased fitness of *Pik3ca*[H1047R] over wild-type cells is driven by a bias in mutant progenitor cell fate toward proliferation (Fig. 3i, Supplementary Note and Supplementary Video).

### Increased PI3K pathway activity confers a fitness advantage

To further investigate mutant cell fitness we used epithelioid three-dimensional (3D) primary cultures[30,31]. We generated esophageal epithelioids *Rosa26*[RYFP/RYFP] (*Pik3ca*[wt/wt], henceforth referred to as *WT-RYFP*) and *Pik3ca*[H1047R-YFP/wt] mice, and induced recombination by

infecting these cultures with adenovirus encoding *Cre* recombinase (Extended Data Fig. 4a,b). Due to the much higher level of YFP expression in *WT-RYFP* compared with *Pik3ca*[H1047R/wt] cells, the former can be identified by flow cytometry (Extended Data Fig. 4c).

We mixed *WT-RYFP* keratinocytes with either uninduced (*Pik3ca*[wt/wt]) or induced (*Pik3ca*[H1047R/wt]) cells and tracked the proportion of *WT-RYFP* cells over time (Fig. 4a and Extended Data Fig. 4c). We first confirmed that uninduced *Pik3ca*[wt/wt] cells mixed with *WT-RYFP* cells competed neutrally (Fig. 4b, top and Fig. 4c,d). However, when induced *Pik3ca*[H1047R/wt] and *WT-RYFP* cells were co-cultured, the *Pik3ca*[H1047R/wt] cells almost completely took over the culture within 28 days (Fig. 4b, bottom and Fig. 4c). The suprabasal:basal cell ratio of induced *Pik3ca*[H1047R/wt] cells was lower than for neighbor *WT-RYFP* cells (Fig. 4d), consistent with the reduced differentiation of mutant cells observed in vivo (Fig. 3g). These results indicate that *Pik3ca*[H1047R/wt] mutant cells have an epithelial cell-autonomous competitive advantage over wild-type cells in vitro.

We reasoned that the fitness advantage of *Pik3ca*[H1047R/wt] mutants result from increased activation of the PI3K pathway compared with wild-type cells (Fig. 4e and Extended Data Fig. 1c–f). If so, increasing the PI3K activity in wild-type cells would reduce the signaling differences between the two genotypes and neutralize the mutant cell's competitive advantage. Overactivation of PI3K signaling using supraphysiological doses of insulin (Extended Data Fig. 4d)[32] abrogated the differences in differentiation and gene expression between wild-type and mutant cells (Fig. 4d–f). Indeed, 82% of the genes upregulated in *Pik3ca*[H1047R/wt] mutant cells were also induced in wild-type cells upon insulin treatment (Fig. 4g). Importantly, this treatment significantly reduced the competitive advantage of *Pik3ca*[H1047R/wt] mutant over *Pik3ca*[wt/wt] wild-type cells in mixed cultures (Fig. 4h,i). Adding insulin within the physiological range, or another growth factor, epidermal growth factor (EGF), had no effect on mutant cell advantage (Extended Data Fig. 4e). The inhibitory effect of supraphysiological insulin on mutant cell advantage was reversible, as mutant cells were able to outcompete wild-type cells again when insulin was removed after 15 days of treatment (Extended Data Fig. 4f). Conversely, reduction in PI3K pathway activity in both wild-type and mutant cells by treating mixed cultures with the PI3K inhibitor LY294002 or the mTOR inhibitor rapamycin also reduced the mutant cell advantage (Fig. 4j and Extended Data Fig. 4g).

### PI3K pathway modulates progenitor cell fitness

To confirm the role of the PI3K pathway in regulating cell competition genetically, we performed a targeted CRISPR–Cas9 competitive fitness screen in wild-type 3D mouse primary esophageal cultures (epithelioids) grown in minimal media. Cas9-expressing keratinocytes from *Rosa26*[Cas9/wt] mice were infected with a lentiviral library expressing 1,080 guide RNAs (gRNAs) including nontargeting gRNAs, gRNAs

---

**Fig. 4 | Differential PI3K pathway activation modulates the competitive advantage of *Pik3ca*[H1047R/wt] cells. a**, Protocol: *Rosa26*[flYFP/flYFP] (WT-RYFP) or *Pik3ca*[H1047R/wt] (*Pik3ca*[mut]) cells were mixed with uninduced *Pik3ca*[wt/wt] cells (*Pik3ca*[wt]), from the same animal, cultured at confluence and analyzed at 14 and 28 days (Extended Data Fig. 4c). **b**, A typical confocal basal layer section of mixed culture. WT-RYFP cells, yellow; DAPI, blue. Scale bar, 20 μm. **c**, Proportion of WT-RYFP cells versus *t* = 0. **d**, The suprabasal:basal cell ratio at 14 days. +INS, treated with 5 μg ml⁻¹ insulin. In **c** and **d**, *n* = 10–11 cultures from individual animals per condition. Two-tailed unpaired *t*-test. Mean ± s.d. **e**, Uninduced or induced cells were cultured overnight in starvation medium (STV) and then cultured for 1 h in STV, or STV plus LY294002 50 μM, or STV with insulin 5 μg ml⁻¹, then lysed. Western blots for P-AKT(S473), P-AKT(T), AKT, P-GSK3β, GSK3β, P-S6, S6 and α-tubulin are shown, representative of three biological replicates. **f**, M, log ratio and A, mean average (MA) plots of RNA sequencing (RNA-seq) of induced *Pik3ca*[H1047R/wt] and uninduced *Pik3ca*[wt/wt] (WT) cultures comparing CTL and +INS treatments; red, differentially expressed transcripts (Wald test corrected for

multiple testing, adjusted *P* < 0.05). **g**, Venn diagram of genes upregulated in *Pik3ca*[H1047R/wt] cells also upregulated by insulin treatment of wild-type cells. **h**, A representative basal layer section of WT and *Pik3ca*[H1047R/wt] mixed cultures after 28 days +INS or CTL. WT-RYFP cells, yellow; DAPI, blue. Scale bar, 20 μm. **i**, The proportion of WT-RYFP cells in mixed culture with *Pik3ca*[H1047R/wt] cells, versus *t* = 0. +INS, treatment with 5 μg ml⁻¹ insulin. Each dot represents a primary culture from an animal, lines connect means. *n* = 10–11 cultures from individual animals. Two-tailed paired *t*-test. **j**, The proportion of WT-RYFP cells mixed with *Pik3ca*[H1047R/wt] cells, versus *t* = 0. Cells treated either in minimal medium or 0.5 μM LY294002. Each dot represents a primary culture from an animal; the lines connect means. *n* = 4–16 primary cultures from individual animals, per condition. Two-tailed paired *t*-test. **k**, Cell competition and PI3K activation. Increased PI3K pathway activity gives *Pik3ca*[H1047R/wt] cells a competitive advantage over wild-type cells at physiological insulin levels. Leveling-up or leveling-down PI3K activity between *Pik3ca*[H1047R/wt] mutant and wild-type cells with supraphysiological insulin (high INS) or PI3K inhibitor, respectively, reduces mutant competitive advantage.

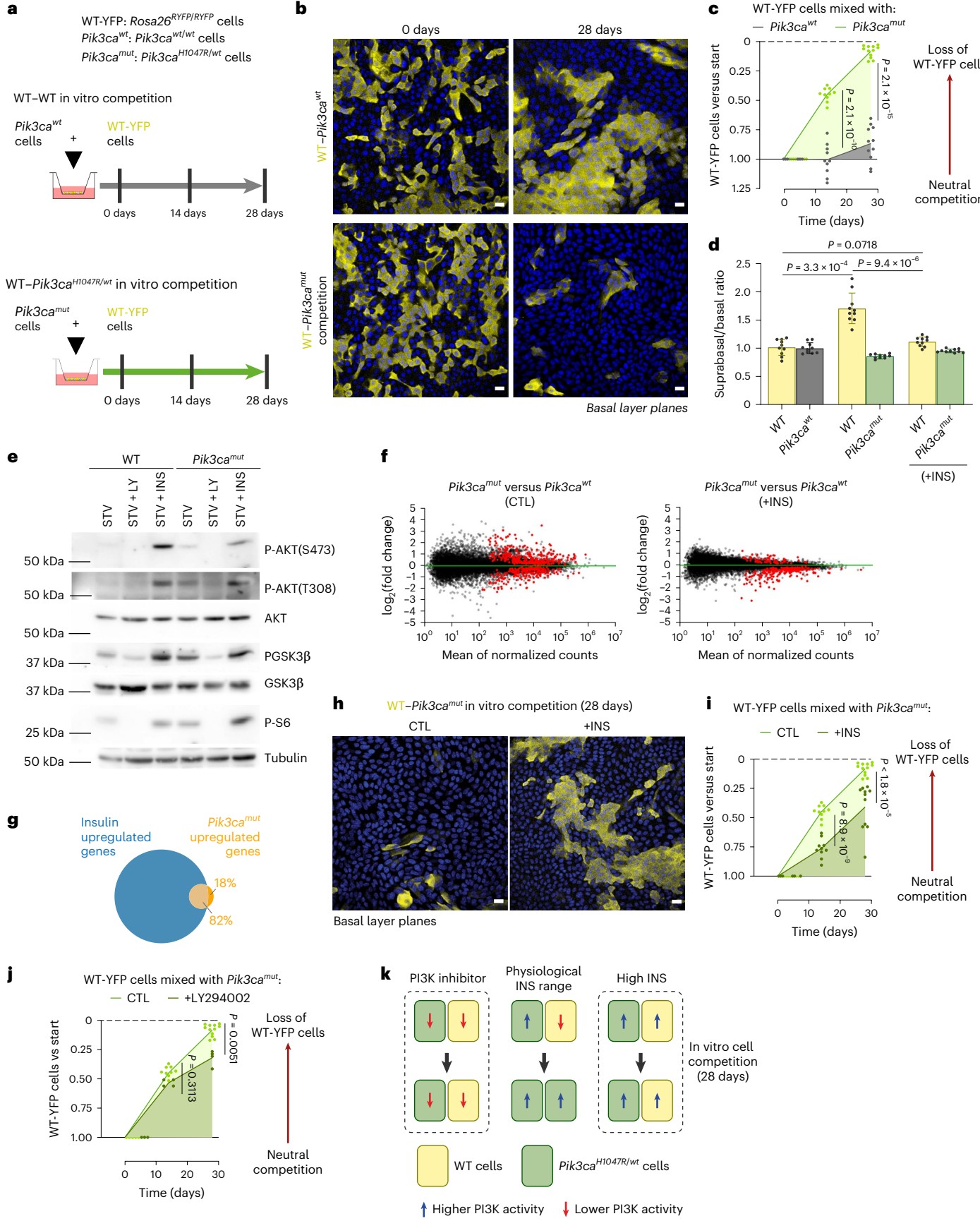

targeting essential genes and gRNAs targeting proteins related to the PI3K pathway (10 gRNAs per gene) (Fig. 5a and Supplementary Table 1). The gRNA representation at time 0 strongly correlated with that in the library (Extended Data Fig. 5a), indicating that representation was not affected by transduction or cell plating. The low infection rate ensured gene edited cells competed against wild-type neighbors over 3 weeks without passaging, after which the abundance of each gRNA was determined by DNA sequencing and compared with time 0 (Fig. 5a). As expected, there was no enrichment of nontargeting gRNAs and a strong depletion of the essential genes (Extended Data Fig. 5b,c). Targeting positive regulators of the PI3K pathway such as *Igf1r*, *Irs2*, *Pik3ca*, *Pdpk1* and *Akt1* reduced cell fitness, while the depletion of the pathway inhibitor *Pten* increased fitness (Fig. 5b and Extended Data Fig. 5c). Downstream of PI3K, depletion of mTOR pathway components such as *Mtor*, *Rheb*, *Rptor* or *Rictor* reduced cell fitness, while depleting mTOR pathway inhibitors such as *Tsc1*, *Tsc2* and *Tbc1d7* significantly enhanced competitiveness, confirming that the PI3K–mTOR pathway promotes esophageal progenitor cell fitness (Fig. 5b and Extended Data Fig. 5c).

The same CRISPR screen performed in *Rosa26*^Cas9/wt *Pik3ca*^H1047R/wt mutant cells in minimal medium yielded similar results to wild-type cells, with log₂ fold changes of gRNAs targeting PI3K signaling after 3 weeks strongly correlated in both genotypes (Fig. 5b,c and Extended Data Fig. 5c). The log₂(fold changes) of gRNAs targeting components of the mTOR pathway were less correlated (Fig. 5c), indicating that wild-type cells appear more dependent on the mTOR pathway for competitive fitness than mutant cells. Similar correlations were found in CRISPR–Cas9 screens performed in wild-type cells treated with supraphysiological doses of insulin that overactivated the PI3K pathway, confirming that the fitness of cells with higher PI3K activity is less dependent on the mTOR pathway. In agreement, differences between wild-type and mutant cells were reduced when screens were performed under insulin treatment (Fig. 5d,e). These results are consistent with *Pik3ca*^H1047R mutation promoting cell fitness by PI3K activation via mTOR-dependent and mTOR-independent pathways.

## HIF1α and glycolysis contribute to mutant cell fitness

The PI3K–mTOR pathway modulates signaling and metabolism at transcriptional and posttranscriptional levels through multiple downstream effectors[33]. Although gRNAs targeting the Foxo transcription factors did not largely modify cell fitness, gRNAs targeting *Atf4*, *Hif1α*/*Hif1β*, *Myc*, *Srebf1* or *Srebf2* reduced cell fitness in both wild-type and mutant cells (Fig. 5b and Extended Data Fig. 5c), suggesting that PI3K might partially regulate cell fitness modulating gene expression through multiple transcription factors in parallel[33,34]. To identify the downstream pathways affected by the mutation, we compared the gene expression profiles of induced (*Pik3ca*^H1047R/wt) and uninduced (*Pik3ca*^wt/wt) primary cultures generated from the same mice. RNA sequencing revealed 301 upregulated and 195 downregulated

transcripts (adjusted *P* value <0.05) in mutant cells (Extended Data Fig. 6a,b and Supplementary Table 2). As expected, gene set enrichment analysis showed that mutant cells have increased activation of the PI3K–mTOR pathway (Extended Data Fig. 6c,d). In addition, the expression of genes in both the HIF1 and MYC pathways[33] was upregulated in mutant cells by gene set enrichment analysis (Extended Data Fig. 6e,f). Consistent with activation of HIF1 signaling, Hif1α/hypoxia signaling was one of the most enriched pathways in a Kyoto Encyclopedia of Genes and Genomes analysis of RNA sequencing data (Extended Data Fig. 6g). In total, 47% of the upregulated genes in mutant cells (transcripts with an adjusted *P* value <0.01) were known or predicted direct targets of the HIF1α transcription factor (Extended Data Fig. 6h). Messenger RNA of *Hif1α* and its canonical target gene *Vegfa* were upregulated in mutant cells or insulin-treated cells by RNA sequencing (Extended Data Fig. 6i).

A small increase in HIF1α protein levels was detected in mutant cells, which was dependent on PI3K activity (Extended Data Fig. 6j). Consistent with an upregulation in HIF1α signaling, we observed upregulation of glycolysis-related genes in mutant cells (Extended Data Fig. 6g,k,l). Such gene expression differences were abrogated by overactivating the PI3K pathway using supraphysiological insulin doses, which highly induce glycolysis gene expression in wild-type cells (Extended Data Fig. 6l,m). High-resolution respirometry demonstrated that such gene expression differences translated into metabolic differences, as mutant cells are significantly more glycolytic (lower OCR/ECAR ratio[35]) and differences are abrogated after insulin treatment (Extended Data Fig. 6n). In summary, we conclude that mutant cells show an upregulation of Hif1α signaling and a metabolic switch to aerobic glycolysis, as described in multiple *PIK3CA*^H1047R mutant cell lines[36].

To test if HIF1α pathway activation was one of the effectors of *Pik3ca*^H1047R/wt cell phenotype, we generated *Pik3ca*^wt/wt and *Pik3ca*^H1047R/wt primary cells stably expressing short hairpin RNA (shRNA) targeting *Hif1a* (sh*Hif1a*), or a nontargeting shRNA (shNT) (Extended Data Fig. 7a,b) and assessed their competitive fitness over *WT-RYFP* cells. Silencing *Hif1a* expression significantly reduced the advantage of *Pik3ca*^H1047R/wt mutant over wild-type cells (Extended Data Fig. 7c,d). These findings were corroborated pharmacologically, by treating mixed cultures of induced *Pik3ca*^H1047R/wt and *WT-RYFP* cells with the HIF1α inhibitor PX478, which reduced the advantage of mutant over wild-type cells (Extended Data Fig. 7e,f)[37]. We concluded that activation of HIF1 signaling partially contributes to the competitive advantage of *Pik3ca*^H1047R/wt mutant over wild-type cells, consistent with the results of the CRISPR screen.

We further investigated the metabolic changes in *Pik3ca* mutant cells (Extended Data Fig. 8a–k, Supplementary Fig. 2 and Supplementary Note). This led us to conclude that, as well as increased HIF1α signaling, glycolysis and lipogenesis also contribute to increased mutant cell fitness. Multiple metabolic changes are thus implicated in the mutant phenotype.

---

**Fig. 5 | CRISPR screen of genes regulating wild-type and mutant fitness.**
**a**, Protocol: primary cultures from *Pik3ca*^H1047R/wt *Rosa26*^Cas9/wt mice were induced (*Pik3ca*^mut/wt) or uninduced (wild type, WT) and infected with a lentiviral gRNA library targeting PI3K–mTOR-related genes. The 0-week time point was assessed, and the remaining cells were cultured in minimal medium (CTL) or minimal medium with 5 μg ml⁻¹ insulin (INS) for 3 weeks. The relative abundance of gRNA between the 3- and 0-week time points is expressed as log₂(fold change) (LFC). The volcano plot shows the enrichment score and log₂(fold change) of genes screened in WT cells in CTL medium. Significantly depleted or enriched gRNAs (false discovery rate (FDR) <0.1 and >10% fold change) are blue or orange, respectively, and unchanged gRNAs are gray. *n* = 2 biological replicates, 10 gRNA per gene. **b**, CRISPR screening results in WT (left) or *Pik3ca*^H1047R/wt (right) in CTL medium showing PI3K pathway and downstream genes including transcription factors (TF). Bold red: gRNAs that are significantly enriched or depleted (FDR <0.1 and >10% fold change difference). Pathway activation and inhibition is

indicated by green and red arrows, respectively. The yellow boxes indicate the enzyme isoform most expressed in esophageal primary cells. **c**–**e**, Plots showing the correlation between average log₂(fold change) of gRNA targeting indicated genes in the following cells and conditions: *Pik3ca*^H1047R/wt cells (*y* axis) versus WT cells (*x* axis) in control (CTL) condition (the panels indicate gene sets of PI3K pathway genes (top) and mTOR pathway genes (bottom)) (**c**); WT cells in CTL (*x* axis) versus INS (*y* axis) conditions (the panels indicate gene sets corresponding to PI3K pathway genes (left) and mTOR pathway genes (right)) (**d**); INS condition comparing *Pik3ca*^H1047R/wt cells (*y* axis) versus WT cells (*x* axis) (the panels indicate gene sets corresponding to PI3K pathway genes (left) and mTOR pathway genes (right)) (**e**). In **c**–**e**, the yellow and green areas of each graph indicate the higher absolute log₂(fold change) in *x*-axis or *y*-axis condition, respectively. Linear regression (black) with slope and coefficient of determination *R*². Identity line, orange. The error bars indicate the s.d. of *n* = 2–3 screen replicates.

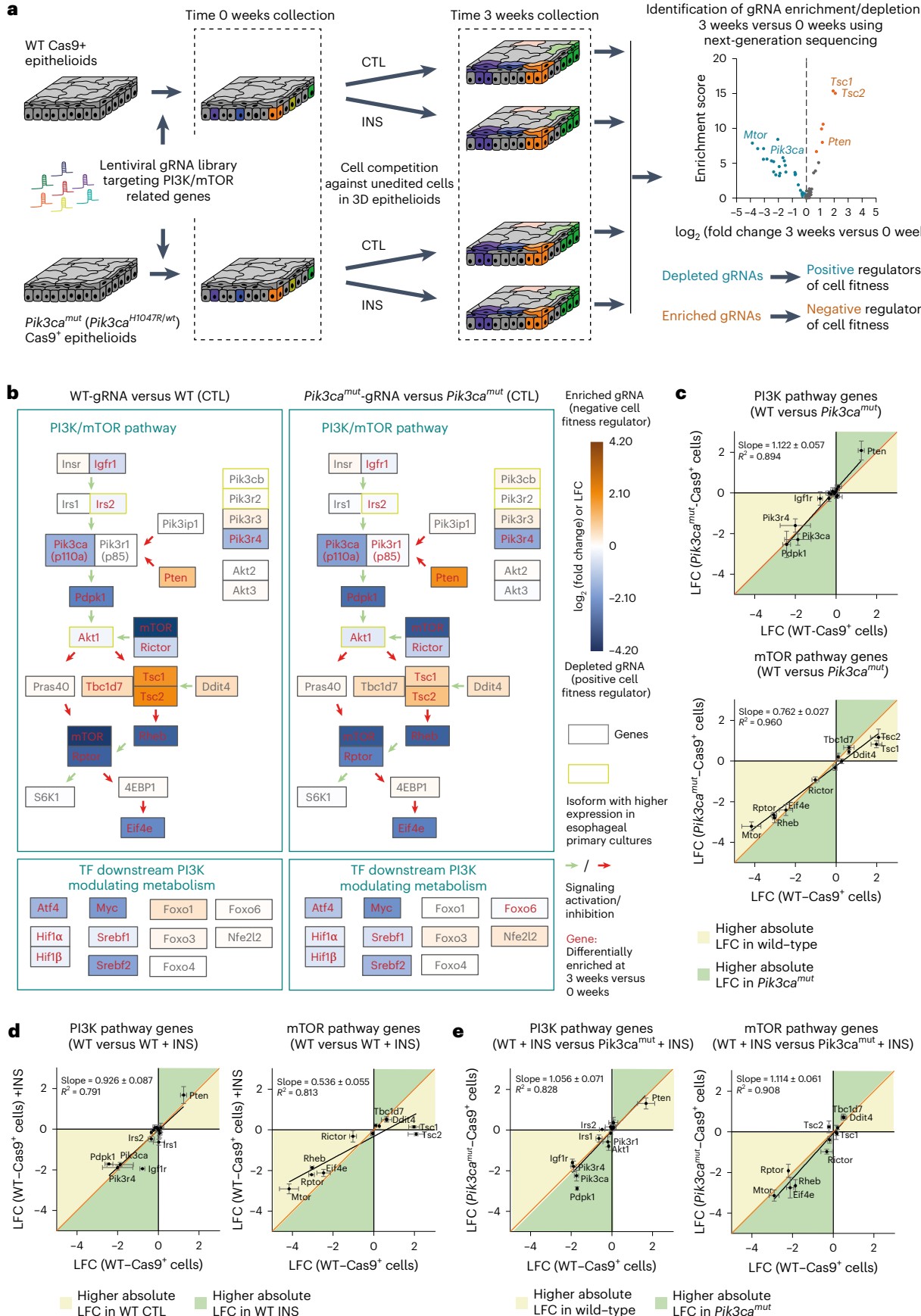

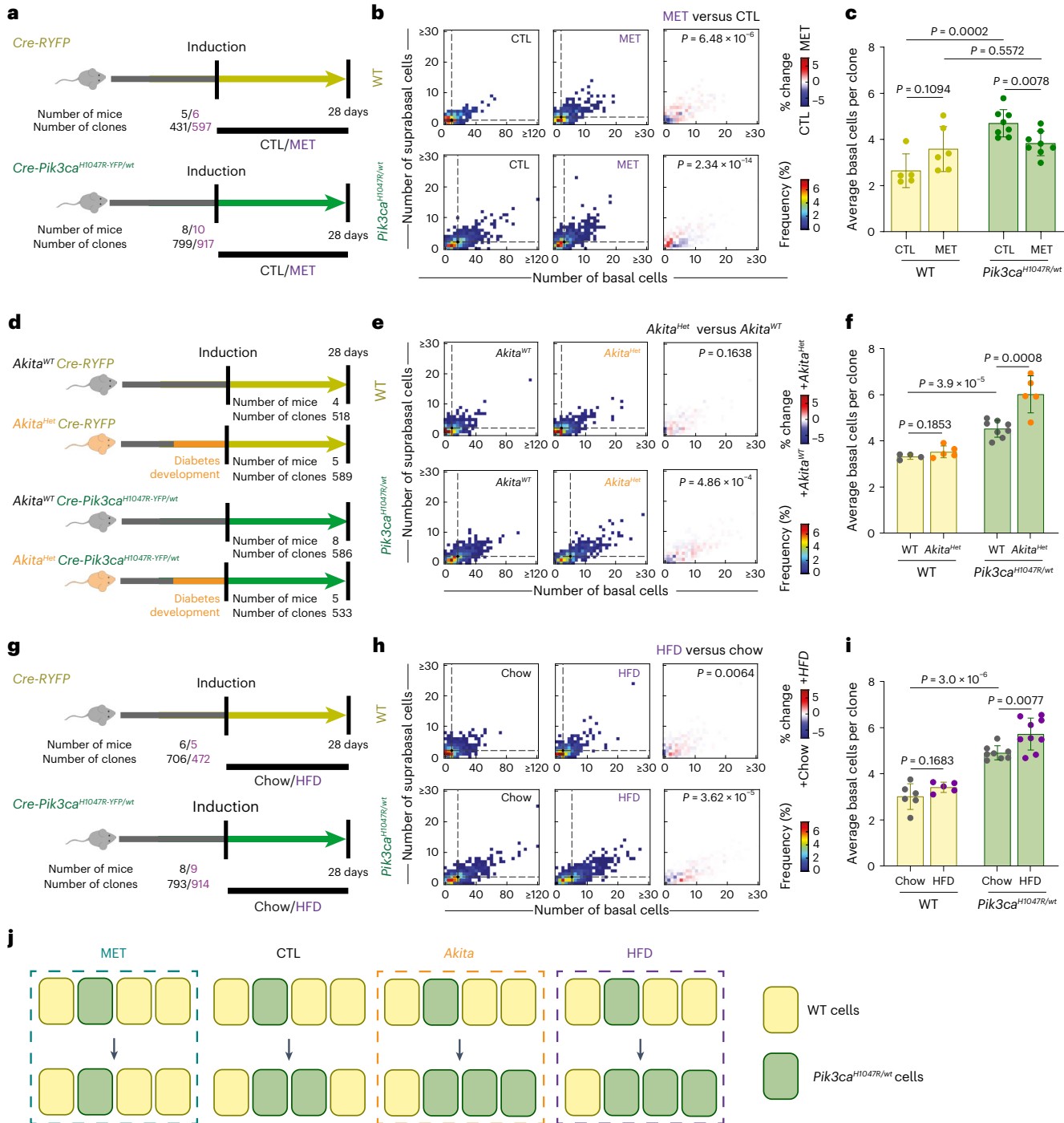

**Fig. 6 | Metabolic conditions alter Pik3ca$^{H1047R/wt}$ competitive advantage.**
**a**, Protocol: *Cre-RYFP* control and *Cre-Pik3ca$^{H1047R-YFP/wt}$* mice were induced and treated with/without metformin (MET). Clones with at least one basal cell were analyzed at 28 days. **b**, Left and middle: heatmaps showing the frequency of basal and first suprabasal layer cells in clones; dots and dashed lines indicate geometric median clone size. Right: heatmaps showing differences between treatment and control in *Cre-RYFP* (top) or *Cre-Pik3ca$^{H1047R-YFP/wt}$* (bottom) animals. Two-tailed 2D Kolmogorov–Smirnov test. **c**, Average basal clone sizes for each strain and treatment. The bars are mean ± s.d. Two-tailed unpaired *t*-test. *n* = 431–917 clones from 5–10 animals per condition. **d**, Protocol: *Cre-RYFP* and *Cre-Pik3ca$^{H1047R-YFP/wt}$* mice in *Akita$^{wt}$* (control) or *Akita$^{Het}$* (diabetic) backgrounds were induced after diabetes development in *Akita$^{Het}$* animals and tissues collected 28 days post-induction. **e**, Left and middle: heatmaps showing the frequency of basal and first suprabasal in clones from **d**. The dots and dashed lines indicate geometric median. Right: heatmaps showing the differences between each treatment and control in *Cre-RYFP* (top) or *Cre-Pik3ca$^{H1047R-YFP/wt}$* (bottom) mice. *n* = 518–589

clones from 4–8 animals per condition. Two-tailed 2D Kolmogorov–Smirnov test. **f**, Average basal clone sizes for each strain and treatment from **d**. The dots indicate the average clone size of a mouse. The bars indicate the s.d. (*n* = 4–8 mice). Two-tailed unpaired *t*-test. **g**, Protocol: *Cre-RYFP* and *Cre-Pik3ca$^{H1047R-YFP/wt}$* mice were induced, fed a chow or HFD, and tissues collected 28 days post-induction. **h**, Left and middle: heatmaps of frequency of basal and first suprabasal layer cells in clones from **g**. The dots and dashed lines indicate geometric median clone size. Right: heatmaps showing differences between treatment and control in *Cre-RYFP* (top) or Cre-*Pik3ca$^{H1047R-YFP/wt}$* (bottom) animals. *n* = 472–914 clones from 5–9 animals per condition. Two-tailed 2D Kolmogorov–Smirnov test. **i**, Average basal clone sizes for each strain and treatment from **g**. The dots indicate the average clone size of a mouse. The bars are s.d. (*n* = 5–9 mice). Two-tailed unpaired *t*-test. **j**, The effects of MET, *Akita* and HFD on mutant cell advantage. In a control situation, mutant clones show advantage that is reduced by MET treatment and increased in *Akita$^{Het}$* background or under HFD.

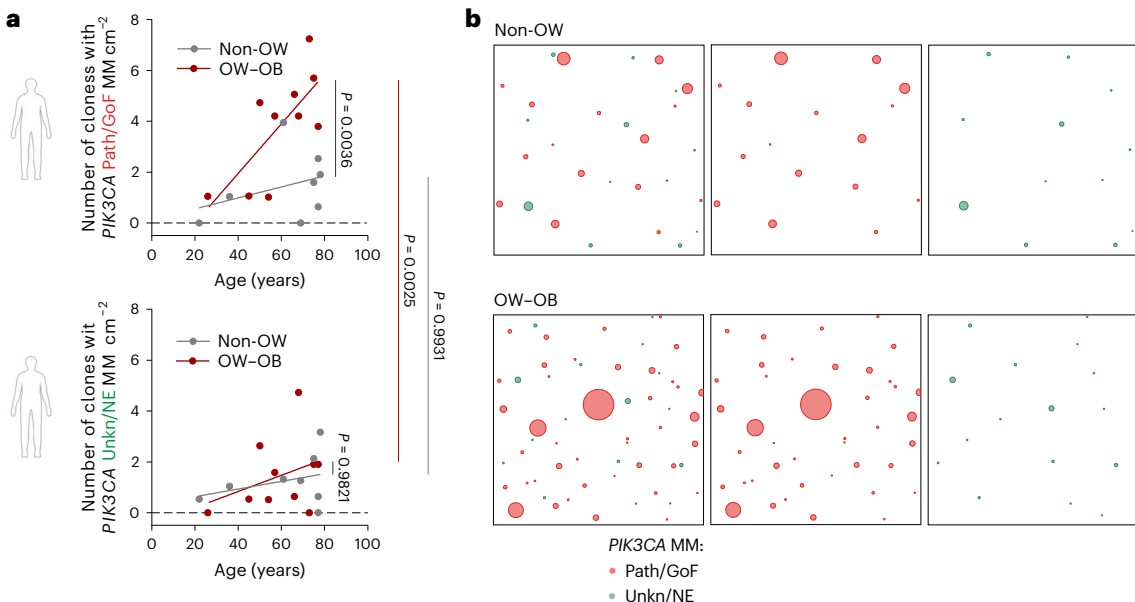

**Fig. 7 | Higher density of *Pik3ca* MMs in overweight humans.** Human donors were classified into overweight, OW–OB (*n* = 10, BMI ≥25 kg m⁻²) and non-OW (*n* = 8, BMI <25 kg m⁻²). *PIK3CA* MMs were classified as Path/GoF and Unkn/NE (Methods) and analyzed separately in OW–OB and non-OW donors. **a**, The number of clones per centimeter squared carrying *PIK3CA* MM Path/GoF (top) or Unkn/NE (bottom) plotted against donor age. Each dot represents a donor. Simple linear regression for OW–OB and non-OW donors is shown. Tukey's test.

**b**, Representative patchwork plots. Each panel is a representation of *PIK3CA* mutant clones in an average 8 cm² area of esophagus from non-OW (top) or OW-OB (bottom) donors. To generate each figure, a sample of biopsies from all donors >60 years of age was randomly selected from each weight category amounting to 8 cm² of tissue. All *PIK3CA* mutant clones are represented as circles and randomly distributed in space.

## Organismal metabolism modulates *Pik3ca^{H1047R}* mutant fitness

Collectively, our results argue that *Pik3ca^{H1047R/wt}* mutant cells have increased activation of the insulin–PI3K signaling pathway, which provides them with higher cell fitness. Therefore, conditions affecting the insulin–PI3K signaling pathway may alter the fitness of *Pik3ca^{H1047R}* mutant clones in vivo[38]. To test this hypothesis, we analyzed mutant clonal expansion in three different organismal metabolic scenarios. These were metformin treatment (a widely used antidiabetic agent that increases insulin sensitivity and mouse lifespan[39,40]), *Ins2* mutant mice (Akita) (which model type I diabetes[41,42]) and a high-fat diet (HFD) model in mice (which alters PI3K signaling and promotes insulin resistance, increased body mass and hyperinsulinemia[43–45]).

Metformin treatment reduced mutant clone size and increased the proportion of differentiated mutant cells toward the levels observed in wild-type clones (Fig. 6a–c and Extended Data Fig. 9a). Metformin also significantly decreased the differences between wild-type and mutant basal cell clone size distributions (Extended Data Fig. 9b and Methods). Metformin treatment in vitro activates glycolysis in wild-type cells (Extended Data Fig. 9c) and reduced the expansion of *Pik3ca^{H1047R/wt}* cells in in vitro competition experiments (Extended Data Fig. 9d–f), suggesting that the in vivo effect of metformin could be partially explained by a direct reduction of the metabolic differences between wild-type and mutant esophageal cells. In summary, we conclude that metformin reduces the fitness advantage of *Pik3ca* mutant clones in mouse esophagus.

Next, we bred the *Cre-RYFP* and *Cre-Pik3ca^{H1047R-YFP/wt}* strains onto *Akita^{Het}* (diabetic) or *Akita^{wt}* (nondiabetic) backgrounds. Diabetes development in *Akita^{Het}* mice was confirmed by measuring urinary glucose (Extended Data Fig. 9g). Clonal recombination was induced in both strains after the onset of diabetes in *Akita^{Het}* mice (Fig. 6d), and the number and size of clones were analyzed after 1 month. Type 1 diabetes did not modify the global clone size distribution or average size of wild-type clones. However, *Pik3ca^{H1047R/wt}* mutant clones were larger and the distribution of basal and suprabasal cells was altered in diabetic

mice compared with nondiabetic littermates (Fig. 6e,f and Extended Data Fig. 9h,i). Differences in basal and suprabasal cell distributions between wild-type and mutant clones were further increased in *Akita^{Het}* mice (Extended Data Fig. 9h). We conclude that the fitness advantage of *Pik3ca^{H1047R}* mutant cells is increased in a diabetic background.

Finally, we examined the effects of HFD. *Cre-RYFP* and *Cre-Pik3ca^{H1047R-YFP/wt}* mice were induced and fed HFD or a control diet with matched ingredients, and the clone sizes were analyzed 1 month later (Fig. 6g). As previously reported, HFD significantly increased body weight and hyperinsulinemia, suggesting insulin resistance, compared with the control diet (Extended Data Fig. 9j,k)[45–47]. HFD caused small changes in the global clone size distributions of wild-type RYFP clones but did not alter average clone size or the proportion of suprabasal cells per clone. However, HFD substantially altered mutant global clone size distribution and average clone size, and decreased the proportion of suprabasal cells in mutant clones, in both males and females (Fig. 6h,i, Extended Data Fig. 9l,m, Supplementary Fig. 3 and Supplementary Tables 3 and 4). HFD further increased the differences between mutant and wild-type basal clone size distributions (Extended Data Fig. 9m and Methods).

Taken together, these results demonstrate that metabolic conditions affecting the insulin–PI3K axis modulate the selection of *Pik3ca^{H1047R}* mutant clones in the mouse esophagus (Fig. 6j). Further work will be needed to understand whether the modulation of mutant cell advantage in these conditions depends on the change in glycaemia, insulinemia or a combination of both.

## *PIK3CA* mutant clone density is increased in overweight humans

These findings led us to explore the relationship between BMI and *PIK3CA* mutant clone size in human esophagus. We sequenced the *PIK3CA* gene in 698 samples of esophageal epithelium (covering a total of 13.96 cm²) from 10 individuals with no previous diagnosis of diabetes. We combined these data with published results shown in Fig. 1

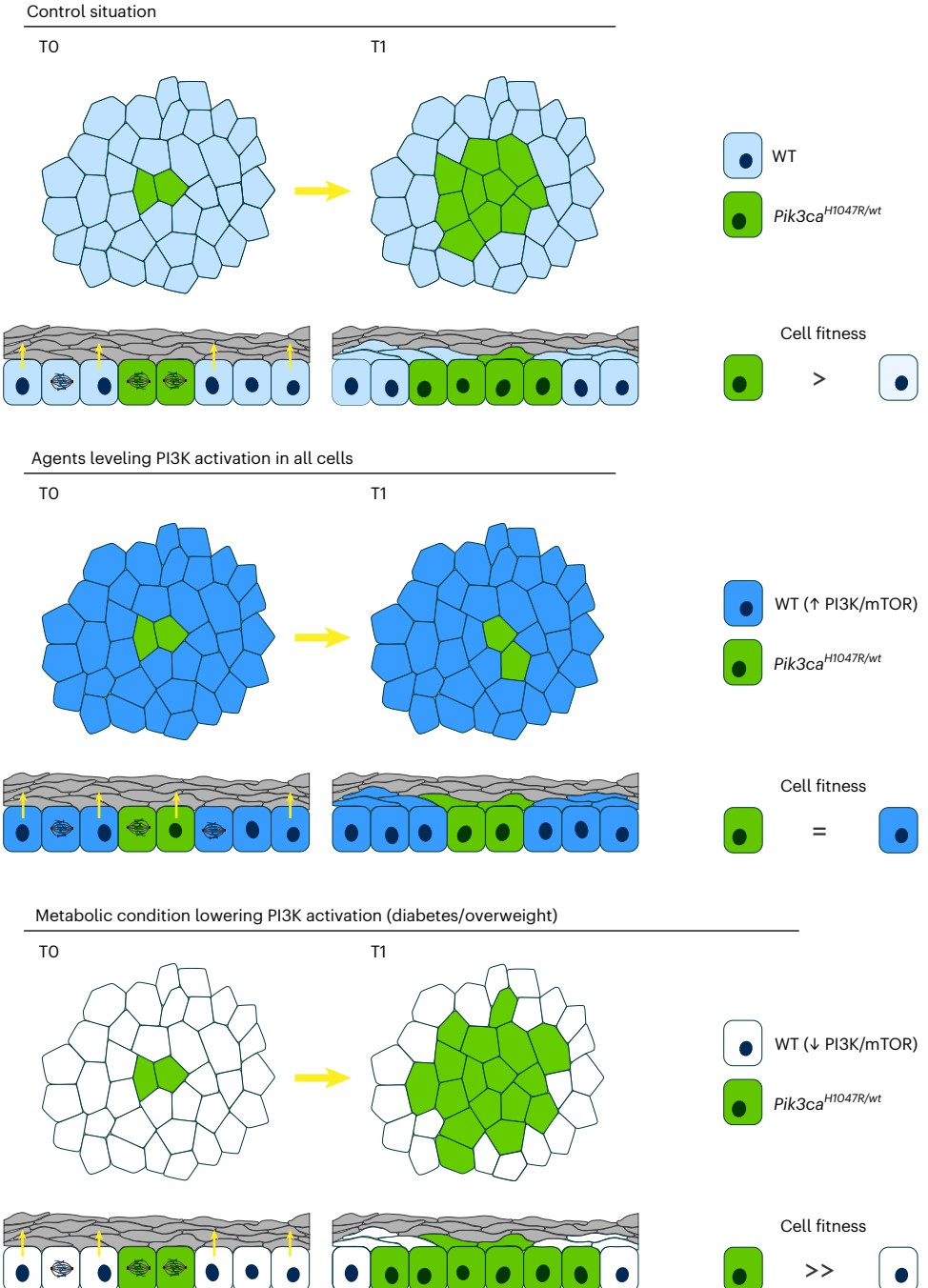

**Fig. 8 | Interventions that alter the competitive fitness of oncogenic PIK3CA mutant clones.** Top: overactivation of the PI3K–mTOR pathway increases the competitive fitness of *Pik3ca*^*H1047R/wt*^ mutant over wild-type cells, by reducing mutant cell differentiation, which drives clonal expansion. Middle: interventions that level out the differential activation of the PI3K–mTOR axis and cell fitness between wild-type and mutant cells neutralize the competitive advantage of the mutants. Bottom: on the contrary, metabolic conditions such as type 1 diabetes and overweight would further increase the cellular differences in the PI3K–mTOR axis between both cell types, resulting in a further increase in the competitive fitness of the mutant over wild-type cells.

(Supplementary Table 5)[1]. Donors were classified into nonoverweight (non-OW, BMI <25 kg m$^{-2}$) and overweight–obese groups (OW–OB, BMI ≥25 kg m$^{-2}$)[48]. The groups had an average age 58 and 62 years, respectively (Extended Data Fig. 10a). *PIK3CA* MMs were classified into Path/GoF or unknown/no evidence (Unkn/NE) (Methods). The distribution of *PIK3CA* mutations was similar between both groups, with *PIK3CA*^*H1047R*^ being the most prevalent.

A multiple regression linear model was used to examine the mixed effects of mutation pathogenicity, donor age and bodyweight on the density of *PIK3CA* MM clones (Methods). The number of mutant clones significantly increased with donor age (analysis of variance (ANOVA) *F*-test, *P* = 0.0133). While the density of Unkn/NE *PIK3CA* mutations was not different between groups, Path/GoF mutations were significantly more frequent in OW–OB than in non-OW donors (Fig. 7a,b and Extended Data Fig. 10b). We estimate that, in the esophagus, Path/GoF *PIK3CA* mutant clones accumulate at a rate of 9.8 mutations dm$^{-2}$ yr$^{-1}$ in OW–OB donors (95% confidence interval (CI) 2.3–17.2), as opposed to 2.2 mutations dm$^{-2}$ yr$^{-1}$ in non-OW donors (95% CI −3.7 to 8.1). Consistently, the effect of pathogenic mutations on clonal density was significantly influenced by BMI (ANOVA *F*-test, *P* interaction 0.0217)

(Fig. 7a and Extended Data Fig. 10b). These results suggest that Path/GoF *PIK3CA* MM are specifically selected in OW–OB individuals. As a result, these mutations cover a larger proportion of tissue than Unkn/NE mutations in OW–OB, while no differences are found in non-OW donors (Extended Data Fig. 10c). An increased density of pathogenic variants of the kinase domain was observed in the BMI ≥25 group (Extended Data Fig. 10d), consistent with the increased fitness of *Pik3ca^H1047R* clones in mice fed a HFD (Fig. 6g–i).

We conclude that clones carrying pathogenic MMs in the kinase domain of *PIK3CA*, such as *PIK3CA^H1047R*, have an additional competitive advantage in the esophagus of people with high BMI.

## Discussion

Our results show that subtle activation of the PI3K pathway, caused by a heterozygous activating mutation in *PIK3CA*, is sufficient to drive clonal expansion in normal esophageal epithelium in humans and mice. The mechanism is a bias in mutant cell fate. We find genetic evidence for the PI3K–mTOR axis regulating progenitor fate. *Pik3ca^H1047R/wt* clones hijack this pathway driving clonal expansion. In vivo models of diabetes and diet-induced overweight increase mutant cell expansion, while the antidiabetic drug metformin decreases it (Fig. 8).

The limitations of the study include the lack of direct evidence for signaling differences between *Pik3ca^H1047R/wt* clones and wild-type cells in vivo and showing that they are altered in the various metabolic states we studied. Changes induced by targeting oncogenic *Pik3ca* alleles to the endogenous allele in mice are difficult to detect with immunostaining, suggesting they are likely to be modest compared with those in *Pik3ca* mutant overexpression studies[49–51]. More work is also needed to characterize the relative contributions of the different metabolic changes induced by the mutant allele to cell fitness, for example, exploring the effects of multiple inhibitors and culture conditions on mutant fitness in vitro. Nevertheless, our results suggest that reducing insulin signaling in peripheral tissues via lower insulin levels or insulin sensitivity may enhance mutant fitness.

Our observations parallel studies in *Drosophila* where the fitness advantage gained by increased PI3K signaling can be modulated by insulin or metformin, suggesting a conserved mechanism across species[52,53]. However, the activity of the pathway is critical. For example, overexpression of *Pik3ca^H1047R* drives clonal expansion in the mouse intestine, while in the *Drosophila* wing disc PI3K overexpression has no effect[54,55]. In humans, heterozygous activating mutations in *PIK3CA* are positively selected in the skin[56], indicating they provide competitive advantage; however, the biallelic expression of *Pik3ca^H1047R/H1047R* in mouse skin drives their elimination from the tissue possibly related to senescence induced by high-PI3K signaling[57]. These observations stress the importance of using mouse models that express heterozygous activating *Pik3ca* mutants from the endogenous promoter to model the corresponding mutants in human epithelia[19,20].

We further demonstrate that overweight alters the selection of activating *PIK3CA* mutations in human esophagus. Indeed, our data argue that metabolic conditions alter the competitive selection of oncogenic mutant clones in normal tissues, which may impact cancer risk. Mutations in the insulin–PI3K pathway are common in breast and endometrial cancers, the incidence of which is positively correlated with BMI[38,48]. This correlation may be related to the expansion of *PIK3CA* mutations in overweight individuals before carcinogenesis. Metformin may suppress the expansion of this subset of oncogenic mutant cell clones in normal adult epithelia.

The concept of remodeling the mutational landscape of normal tissues described here sets the basis for studies on how metabolic conditions and treatments may affect mutant selection in human tissues and represents a new potential point of intervention in aging and cancer prevention[58–60].

## Online content

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

[1]Wellcome Sanger Institute, Hinxton, UK. [2]Department of Biomedical Sciences, Faculty of Medicine, Universitat de Barcelona, Barcelona, Spain. [3]Lipid Trafficking and Disease Group, Institut d'Investigacions Biomèdiques August Pi i Sunyer, Barcelona, Spain. [4]Spanish National Cancer Research Centre, Madrid, Spain. [5]Department of Biochemistry and Molecular Biology, Complutense University of Madrid, Madrid, Spain. [6]Wellcome/Cancer Research UK Gurdon Institute, University of Cambridge, Cambridge, UK. [7]Institució Catalana de Recerca i Estudis Avançats, Barcelona, Spain. [8]Cologne Excellence Cluster on Stress Responses in Ageing-Associated Diseases, Cologne, Germany. [9]UCL Cancer Institute, University College London, London, UK. [10]Department of Oncology, University of Cambridge, Hutchison Research Centre, Cambridge Biomedical Campus, Cambridge, UK. [11]Present address: Cambridge Institute of Science, Altos Labs, Cambridge, UK. [12]These authors contributed equally: Albert Herms, Bartomeu Colom. ✉e-mail: pj3@sanger.ac.uk

## Methods

### Ethical approval for animal experiments

All experiments were approved by the local ethical review committees at the Wellcome Sanger Institute and conducted according to Home Office project licenses PPL70/7543, P14FED054 and PF4639B40.

### Ethical approval for human study

The study protocol was ethically reviewed and approved by the UK National Research Ethics Service Committee East of England, Cambridge South, Research Ethics Committee protocol reference 15/EE/0152 NRES. Esophageal tissue was obtained from deceased organ donors. Written Informed consent was obtained from the donor's next of kin.

### Mouse experiments

Animals were maintained on a C57/Bl6 genetic background. Mice were maintained in a specific-pathogen-free unit under a 12 h light and 12 h dark cycle with lights off at 19:30 and no twilight period. The ambient temperature was $21 \pm 2\,°C$, and the humidity was $55 \pm 10\%$. Mice were housed at three to five mice per cage (overall dimensions of caging: 365 mm × 207 mm × 140 mm (length × width × height), floor area, 530 cm$^2$) in individually ventilated caging (Tecniplast, Sealsafe 1284L) receiving 60 air changes per hour. In addition to aspen bedding substrate and standard environmental enrichment of Nestlets, a cardboard tube/tunnel and wooden chew blocks were provided. Mice were given water and diet ad libitum. Animals were fed on standard chow unless otherwise specified. Experiments were carried out with male and female animals, and no sex specific differences were observed. Animals were randomly assigned to experimental groups. $Pik3ca^{fl\text{-}H1047R\text{-}T2A\text{-}EYFP\text{-}NLS/wt}$ knock-in mice generation is described in the Supplementary Note. For lineage tracing experiments, the relevant floxed reporter lines were crossed onto the $Ahcre^{ERT}$ strain in which transcription from a transgenic $CYP1A1$ (arylhydrocarbon receptor, $Ah$) promoter is normally tightly repressed until is activated by β-napthoflavone[61]. In this model, tamoxifen promotes the nuclear translocation of Cre$^{ERT}$ protein to mediate recombination. For lineage tracing of control clones, the $Rosa26^{flYFP/wt}$ mice, which express YFP from the constitutively active $Rosa26$ locus, were used[62]. To assess the mutant and wild-type clone growth in the same mouse, $Rosa26^{flConfetti}$ animals were used[27]. Homozygous $Ahcre^{ERT}Rosa26^{flConfetti}$ animals were crossed onto $Pik3ca^{fl\text{-}H1047R\text{-}T2A\text{-}EYFP\text{-}NLS/fl\text{-}H1047R\text{-}T2A\text{-}EYFP\text{-}NLS}$ to generate $Ahcre^{ERT}$ $Rosa26^{flConfetti}\text{-}Pik3ca^{fl\text{-}H1047R\text{-}T2A\text{-}EYFP\text{-}NLS/wt}$. Following induction, this strain yields cells expressing one of four colors of reporter from the $Rosa26$ locus (wild-type clones), just the YFP reporter of $Pik3ca$ mutant cells ($Pik3ca^{H1047R/wt}$ clones) or both ($Pik3ca^{H1047R/wt\text{-}Confetti}$ Confetti clones) (Extended Data Fig. 2b). Because green fluorescent protein (GFP), YFP and cyan fluorescent protein (CFP) clones are recognized by the same anti-GFP antibody used to detect the EYFP co-expressed with the mutation, only RFP-expressing clones (wild-type or mutant) were analyzed. Where indicated, mice were treated with a well-tolerated dose of metformin (Sigma-Aldrich 317240, 2 mg ml$^{-1}$) or sodium dichloroacetate (DCA; Sigma-Aldrich 347795, 0.5 mg ml$^{-1}$) in drinking water containing 15% 'Ribena' to enhance palatability, for the duration of the experiment. In those cases, the results were compared with control mice treated with 15% 'Ribena' in water. For the analysis of $Rosa26^{flYFP/wt}$ wild-type or $Pik3ca^{H1047R/wt}$ mutant clones in a hypoinsulinemic background, $Ahcre^{ERT}$ $Rosa26^{flYFP/flYFP}$ or $Ahcre^{ERT}Pik3ca^{fl\text{-}H1047R\text{-}T2A\text{-}EYFP\text{-}NLS/fl\text{-}H1047R\text{-}T2A\text{-}EYFP\text{-}NLS}$ female mice were crossed with heterozygous $C57BL/6\text{-}Ins2^{Akita}/J$ males ($Akita^{Het}$) (The Jackson Laboratory, 003548) to produce $Ahcre^{ERT}\text{-} Rosa26^{flYFP/wt}\text{-}Ins2^{Akita/wt}$ and $Ahcre^{ERT}\text{-}Rosa26^{flYFP/wt}\text{-}Ins2^{wt/wt}$ or $Ahcre^{ERT}\text{-}Pik3ca^{fl\text{-}H1047R\text{-}T2A\text{-}EYFP\text{-}NLS/wt}\text{-}Ins2^{Akita/wt}$ and $Ahcre^{ERT}\text{-}Pik3ca^{fl\text{-}H1047R\text{-}T2A\text{-}EYFP\text{-}NLS/wt}\text{-}Ins2^{wt/wt}$ littermates. The development of diabetes phenotype was confirmed by analyzing the presence of glucose in urine at the start and end of the experiments (Extended Data Fig. 8g) using SURERS2/50 strips (SureScreen

Diagnostics, UK) according to the manufacturer's instructions. For the HFD experiments, mice were fed during the indicated times with irradiated rodent diet with 45 kcal% fat (D12451i, Researchdiets) or a chow diet with matched ingredients (D12450H, Researchdiets).

### Lineage tracing

To induce low-frequency expression of EYFP in the mouse esophagus, 10–16-week-old transgenic mice were given a single intraperitoneal (i.p.) of 80 mg kg$^{-1}$ β-naphthoflavone (MP Biomedicals 156738) and 1 mg tamoxifen (Sigma-Aldrich N3633) ($Ahcre^{ERT}Rosa26^{flConfetti}\text{-}Pik3ca^{fl\text{-}H1047R\text{-}T2A\text{-}EYFP\text{-}NLS/wt}$) or 0.25 mg tamoxifen (other mouse strains), resulting in $2.9 \pm 0.9\%$ and $0.17 \pm 0.04\%$ (mean ± standard deviation (s.d.)) recombined epithelium by area, respectively, 10 days after induction. Following induction, between three and eight mice per time point were culled and the esophagus was collected. Time points analyzed included 10 days and 1, 3 and 6 months after induction. As expression from the endogenous $Pik3ca$ locus is very low, immunostaining was necessary to detect EYFP-nuclear localization sequence (NLS) reporter expression (Extended Data Fig. 2b). Normalized, clone-size distributions were built for each experimental condition and time point from the observed relative frequencies $f_{m,n}$ of clones of a certain size, containing $m$ basal and $n$ suprabasal cells, resulting in two-dimensional (2D) histograms, (displayed as heatmaps using CloneSizeFreq_2Dheat package (https://github.com/gp10/CloneSizeFreq_2Dheat)). A 2D histogram of the residuals or differences observed between conditions in the relative frequencies of each particular clone size (that is, each cell on the grid) was generated when appropriate.

### Chemically induced mutagenesis

To generate mutations in the esophageal epithelium, mice were treated with DEN (Sigma, catalog no. N0756) at 40 mg per 1,000 ml in sweetened drinking water for 24 h on 3 days a week (Monday, Wednesday and Friday) for 8 weeks. When indicated, mice were induced using one or three i.p. injections of 80 mg kg$^{-1}$ β-naphthoflavone (MP Biomedicals 156738) and 1 mg tamoxifen (Sigma-Aldrich N3633), resulting in $2.9 \pm 0.9\%$ and $10.2 \pm 2.8\%$ (mean ± s.d.) recombined epithelium by area, respectively, 10 days after the last induction. After each dosage, mice received sweetened water until the next DEN treatment. Control mice received sweetened water as vehicle for the length of the treatment. After the 8 weeks, all mice were administered normal water.

Sorafenib (LC Laboratories S8599) treatment was performed as described[24]. Briefly, sorafenib was administered by i.p. injection at 10 mg ml$^{-1}$ dissolved in cottonseed oil containing 5% dimethylsulfoxide. Control animals received a vehicle control (5% dimethylsulfoxide in cottonseed oil). Injections were given on alternate days for 6 weeks and starting after DEN treatment.

Animals were collected when the humane endpoint of the license was reached, that is, nontransient signs that deviate from normal health and/or reached a 20% weight loss over maximum weight.

### Analysis of in vivo proliferation by EdU labeling

Three months after induction with one i.p. injection of 80 mg kg$^{-1}$ β-naphthoflavone (MP Biomedicals 156738) and 1 mg tamoxifen (Sigma-Aldrich N3633), 10 μg of EdU in phospate-buffered saline (PBS) was administered by i.p. injection 1 h before culling. Tissues were collected and stained with EdU-Click-iT kit and immunofluorescence as explained below. EdU-positive basal cells were quantified from a minimum of ten $z$-stack images.

### Human samples

A sample of mid-esophagus was removed, placed in University of Wisconsin organ preservation solution (Belzer UW Cold Storage Solution, Bridge to Life) and flash frozen in tissue freezing medium (Leica catalog no. 14020108926)[1]. The BMI of each donor was used to classify them into nonoverweight (BMI <25 kg m$^{-2}$) or overweight–obese

(BMI >25 kg m$^{-2}$). None of the donors used in the study had a previous diagnosis of diabetes.

## Human DNA sequencing

Data in Fig. 1 was from ref. 1. Figure 7 and Extended Data Fig. 9 human data combine the previously published donors with data from PIK3CA locus DNA sequencing from esophagus from ten additional transplant donors. Briefly, human esophagus epithelium was isolated and cut into a gridded array of 2 mm$^2$ samples[1]. DNA was extracted and using QIAMP DNA microkit (Qiagen, 56304). DNA sequencing was performed as described[1]. For mutation calling, we used the ShearwaterML algorithm from the deepSNV package (v1.21.3, https://github.com/gerstung-lab/deepSNV)[1,63]. Instead of using a single-matched normal sample, the ShearwaterML algorithm uses a collection of deeply sequenced normal samples as a reference for variant calling that enables the identification of mutations at very low allele frequencies.

PIK3CA MM classified as pathogenic/gain of function (Path/GoF) were annotated either as pathogenic/likely pathogenic by the ClinVar mutation database (https://clinvarminer.genetics.utah.edu/variants-by-gene/PIK3CA) or as gain of function/predicted gain of function by the Clinical Knowledgebase (https://ckb.jax.org/gene/show?geneId=5290&tabType=GENE_VARIANTS). The remaining mutations, classified as unknown/no effect (Unkn/NE), did not appear in the datasets or were classified as uncertain significance/likely benign by Clinvar or unknown/no effect/no effect predicted by Clinical Knowledgebase. *Pik3ca* mutant clone density was calculated as the number of missense mutant clones classified as above and normalized by the total area of epithelium sequenced in each donor. All analyses were performed per donor and, therefore, are independent of the area of tissue sequenced in each donor.

## Driver mutations in esophageal cancer

To detect the most frequent mutations found in esophageal cancer, we used the Integrative OncoGenomics pipeline (https://www.intogen.org/search) that collects and analyses somatic mutations in multiple datasets including The Cancer Genome Atlas Program (TCGA), to identify cancer driver genes using multiple driver discovery methods[64]. Then, we selected the frequency of MMs in the selected drivers in samples of ESCC from TCGA and International Cancer Genome Consortium (ICGC).

## Primary epithelioid culture

After removing the muscle layer with fine forceps, esophageal explants were placed onto a transparent ThinCert insert (Greiner Bio-One) with the epithelium facing upward and the submucosa stretched over the insert membrane, and cultured in complete FAD medium (cFAD) (50:50) consisting of 4.5 g l$^{-1}$ D-glucose, pyruvate and L-glutamine DMEM (Invitrogen 11971-025):DMEM/F12 (Invitrogen 31330-038), supplemented with 5 µg ml$^{-1}$ insulin (Sigma-Aldrich I5500), 1.8 × 10$^{-4}$ M adenine (Sigma-Aldrich A3159), 0.5 µg ml$^{-1}$ hydrocortisone (Calbiochem 386698), 1 × 10$^{-10}$ M cholera toxin (Sigma-Aldrich C8052), 10 ng ml$^{-1}$ EGF (PeproTech EC Ltd 100-15), 5% fetal calf serum (PAA Laboratories A15-041), 5% penicillin–streptomycin (Sigma-Aldrich, P0781) and 5 µg ml$^{-1}$ apo-transferrin (Sigma-Aldrich T2036). Explants were removed after 7 days once keratinocytes covered half of the membrane. Medium was changed every 3 days. Minimal FAD medium without cholera toxin, EGF, insulin and hydrocortisone was used from 2 weeks after establishing the culture.

## Adenoviral infection

To establish either induced or uninduced mouse primary keratinocyte 3D cultures from *Pik3ca$^{fl-H1047R-T2A-EYFP-NLS/wt}$* mice and *Rosa26$^{flYFP/flYFP}$*, cells were infected with either *Cre*-expressing adenovirus (Ad-CMV-iCre, Vectorbiolabs, #1045) or Null adenovirus (Ad-CMV-Null, Vectorbiolabs, #1300), respectively. Briefly, cells were incubated with

adenovirus-containing medium supplemented with polybrene (Sigma-Aldrich, H9268) (4 µg ml$^{-1}$) for 24 h at 37 °C, 5% CO$_2$. Cells were washed, and fresh medium was added. Infection rates were >90%.

## Generation of cultures stably expressing shRNA

MISSION lentiviral transduction particles expressing shRNA targeting *Hif1a* (*Hif1a*#1: TRCN0000232222) and lentiviral-negative control particles (SHC002V) were purchased from Sigma-Aldrich. Stably transduced cell lines were generated according to the manufacturer's instructions. Briefly, cells were trypsinized and infected in suspension and seeded on a six-well insert. The following day, cells were infected. After 24 h, medium was changed. Cells were selected 7 days later by addition of 1 µg ml$^{-1}$ puromycin (Sigma-Aldrich, P88338). Knockdown efficiency was determined by real-time PCR.

## Cell competition assays

Fully induced *Rosa26$^{flYFP/flYFP}$* cultures (*WT-RYFP*) were trypsinized and mixed, 1:1 for fluorescence-activated cell sorting analysis or 1:3 for microscopy, with *Pik3ca$^{fl-H1047R-T2A-EYFP-NLS/wt}$* either fully induced with Cre-expressing adenovirus (*Pik3ca$^{H1047R/wt}$*) or uninduced (*Pik3ca$^{wt/wt}$*) (infected with null adenovirus). After 1 week, when the cultures are fully confluent, cholera toxin, EGF, insulin and hydrocortisone were removed from the medium and the starting time point was collected and analyzed by flow cytometry on a BD LSRFortessa using Facs DIVA software. At the time points specified, cells were collected and analyzed by flow cytometry or fixed for microscopy. Where indicated, mixed cultures were treated with 2 ng ml$^{-1}$ or 5 µg ml$^{-1}$ insulin (Sigma-Aldrich I5500), 10 ng ml$^{-1}$ EGF (PeproTech EC Ltd 100-15), 10 µM PX478 (Cambridge Bioscience 10005189), 0.5 µM LY294002 (Selleckchem S1105), 5 mM 2-deoxyglucose (Sigma-Aldrich D8375), 500 nM rapamycin (Sigma-Aldrich R0395), 6 µg ml$^{-1}$ betulin (Merck B9757), 10 µM CB839 (Tocris 7591), 30 µM 5-tetradecyloxy-2-furoic acid (TOFA) (Santa Cruz sc-200653), 0.1 µM TVB2640 (Selleckchem S9714), 2.5 mM metformin (Sigma-Aldrich 317240) or 25 mM DCA (Sigma-Aldrich 347795).

## Flow cytometry

Keratinocyte cultures were detached by incubation with 0.05% trypsin–EDTA for 20 min at 37 °C 5% CO$_2$. Cells were pelleted for 5 min at 650*g* and resuspended in PBS to be immediately analyzed using a BD LSRFortessa. Were suprabasal and basal cells needed to be quantified, 2% PFA fixed cells were incubated in blocking solution (0.1% BSA and 0.5 mM EDTA in PBS) for 15 min and then with anti-ITGA6-647 antibody (1:125, BioLegend, UK 313610) in blocking solution for 45 min at room temperature. YFP fluorescence was collected using the 488 nm laser and the 530/30 bandpass filter, and ITGA6-647 fluorescence, to discriminate between basal and suprabasal cells, was collected using the 640 nm laser and the 670/14 bandpass filter. Data were analyzed using FlowJo software (version 10.5.3). Basal cells were defined as ITGA6-positive cells and suprabasal cells as ITGA6-negative cells.

## Immunofluorescence and microscopy

For wholemount staining, the mouse esophagus was opened longitudinally, the muscle layer was removed and the epithelium was incubated for 1 h and 30 min in 20 mM EDTA–PBS at 37 °C. The epithelium was peeled from underlying tissue and fixed in 4% paraformaldehyde in PBS for 30 min. Wholemounts were blocked for 1 h in blocking buffer (0.5% bovine serum albumin, 0.25% fish skin gelatin, 1% Triton X-100 and 10% donkey serum) in PHEM buffer (60 mM PIPES, 25 mM HEPES, 10 mM EGTA and 4 mM MgSO$_4$·7H$_2$O). Anti-GFP antibody (1:4,000, Life Technologies A10262) was incubated over 3–5 days using blocking buffer, followed by several washes over 24 h with 0.2% Tween-20 in PHEM buffer. Where indicated, an additional overnight incubation with anti-caspase-3 (1:500, Abcam ab2302) was performed. Finally, samples were incubated for 24 h with 1 µg ml$^{-1}$ 4′,6-diamidino-2-phenylindole (DAPI) and secondary antibody anti-chicken (1:2,000, Jackson

ImmunoResearch 703-545-155) in blocking buffer. Clones were imaged on an SP8 Leica confocal microscope. Whole tissue or large tissue area images were obtained in most cases with the 20× objective with 1× digital zoom, optimal pinhole and line average, speed 600 Hz and a pixel size of 0.5678 µm per pixel. In YFP tissues at long time points, clones were manually detected in the microscope and individually imaged using 40× objective with 0.75× digital zoom, optimal pinhole and line average, speed 600 Hz and a pixel size of 0.3788 µm per pixel. The numbers of basal and suprabasal cells in each clone were counted manually. Representative images were produced by selecting a 120 × 120 µm area in the images obtained as previously stated. Images were processed using ImageJ software adjusting brightness and contrast and applying a Gaussian blur of 1. EdU incorporation was detected with Click-iT chemistry kit according to the manufacturer's instructions (Invitrogen, 23227) using 555 Alexa Fluor azides. Confocal images for EdU–GFP staining were acquired on a Leica TCS SP8 confocal microscope (objective 20×; optimal pinhole and line average; speed 600 Hz; resolution 1,024 × 1,024, zoom ×2). Images were processed using ImageJ software adjusting brightness and contrast and applying a Gaussian blur of 1. For in vitro culture staining, inserts were fixed in 4% paraformaldehyde in PBS for 30 min, then blocked for 30 min in blocking buffer and incubated overnight with anti-GFP antibody (1:1,000, Life Technologies A10262) in blocking buffer followed by 4× 15 min washes with 0.2% Tween-20 in PHEM buffer. Finally, samples were incubated for 2 h with 1 µg ml$^{-1}$ DAPI (Sigma-Aldrich D9542) and secondary antibody anti-chicken (1:500, Jackson ImmunoResearch 703-545-155) in blocking buffer. Afterward, inserts were washed 4× 15 min with 0.2% Tween-20 in PHEM buffer and mounted in Vectashield (Vector Laboratories H1000). Cultures were imaged on an SP8 Leica confocal microscope, obtained with the 40× objective with 0.75× digital zoom, optimal pinhole and line average, speed 600 Hz and a pixel size of 0.1893 µm per pixel. Then, 3D stacks were generated including all the cell layers of the culture, and where indicated, the basal layer plane was selected.

## Histology

Part of the esophageal tumor was dissected, fixed in 10% formalin for at least 24 h and stored at 4 °C. Tissues were then embedded in paraffin and cut at 5 µm thickness. Sections were stained with hematoxylin and eosin and scanned. Some sections were immunostained. These were blocked for 1 h in blocking buffer (0.5% bovine serum albumin, 0.25% fish skin gelatin, 1% Triton X-100 and 10% donkey serum) in PHEM buffer (60 mM PIPES, 25 mM HEPES, 10 mM EGTA and 4 mM MgSO$_4$·7H$_2$O). Next, sections were incubated for 1 h in blocking buffer with anti-GFP antibody (1:4,000, Life Technologies A10262) and anti-vimentin antibody (abcam ab92547 1:1,000). After three washes with 0.2% Tween-20 in PHEM buffer, sections were incubated for 1 h with 1 µg ml$^{-1}$ DAPI and secondary antibodies anti-chicken (1:2,000, Jackson ImmunoResearch 703-545-155) and anti-rabbit (1:500, Thermo Fisher A32794).

## Respirometry experiments

Fully recombined *Pik3ca*$^{H1047R/wt}$ or uninduced (adenovirus-null infected) parallel cultures from the same mice were treated for at least 1 week in FAD medium supplemented with fetal calf serum, apo-transferrin and penicillin–streptomycin, with or without insulin. At the moment of the experiment, 5 mm circular sections of each culture were obtained using a biopsy punch (Stiefel BC-BI-1600) and transferred to XFe24 Cell Culture microplate wells with the basal layer facing up. Then, 675 µl of bicarbonate-free DMEM (Sigma-Aldrich, D5030) supplemented with 25 mM glucose, 1 mM pyruvate, 4 mM glutamine and 5 µg ml$^{-1}$ apo-transferrin with or without 5 µg ml$^{-1}$ insulin (Sigma-Aldrich I5500) was added to each sample. To eliminate residual carbonic acid from medium, cells were incubated for at least 30 min at 37 °C with atmospheric CO$_2$ in a nonhumidified incubator. Oxygen consumption rate (OCR) and extracellular acidification rate (ECAR) were assayed in a Seahorse XF-24 extracellular flux analyzer by three

measurement cycles of 2 min mix, 2 min wait and 4 min measure. Then, the OCR/ECAR ratio was calculated using XF-24 software. Five technical replicates and four biological replicates were collected per condition.

## Plasma and media biochemical analyses

Plasma samples or media samples collected after 3-day culture in the specified treatments were stored at −80 °C. Mouse insulin was measured simultaneously using a 2-plex Mouse Metabolic immunoassay kit (K15124C-3, MSD). The assay was performed according to the manufacturer's instructions and using recombinant human insulin as calibrator. Lactate was analyzed using a commercial kit (DF16, Siemens Healthcare). All sample measurements were performed by the Medical Research Council Metabolic Diseases Unit Mouse Biochemistry Laboratory.

## Quantification and statistical analysis

Unless otherwise specified, all data are expressed as mean values ± s.d. Differences between groups were assessed by two-tailed unpaired or paired *t*-test (specified in the figure legend) for normally distributed data or two-tailed Mann–Whitney *U* test for skewed data, using GraphPad Prism software V_8.3.1. For paired comparisons of clonal behavior across conditions, Peacock's test was used, a 2D extension of a two-tailed Kolmogorov–Smirnov test (implemented as kstest_2s_2d function for MATLAB). The influence of metformin, DCA, HFD and Akita conditions on differences between *Rosa26*$^{YFP/wt}$ and *Pik3ca*$^{H1047R/wt}$ basal clone sizes was assessed by multiple-regression linear modeling following a nonparametric two-factor design, of type: clone size ~ treatment (or genotype, in the case of Akita) × mutation. Given the skewness of the clone size distributions, we performed analysis of the variance of aligned-rank transformed (ART) data (contrast tests with ART (cran.r-project.org) ARTool (R) package vignette; updated 12 October 2021), reporting the *P* value of the interaction term (omnibus *F*-test). Exact *P* values for statistical tests are specified in the figures. No statistical method was used to predetermine sample size. The experiments were not randomized. The investigators were not blinded to allocation during experiments and outcome assessment.

The analysis of mutations and MM clone colonization in humans was done using a multiple-regression linear model with a three-factor design including possible interactions, of type: age × weight × mutation pathogenicity, using either the number of clones per centimeter squared or the summed VAF as the dependent variable. Age was modeled as a continuous factor, while BMI (below 25 or equal/above 25 kg m$^{-2}$) and pathogenic status (Path/GoF versus Unkn/NE MM) as categorical factors. Results are reported following three-way ANOVA *F*-test, using Tukey's test for post-hoc pairwise comparisons.

## Reporting summary

Further information on research design is available in the Nature Portfolio Reporting Summary linked to this article.

## Data availability

Mouse transcriptomic data can be viewed at https://www.ebi.ac.uk/ena using the following accession numbers: Animal 1 CTL; ERS2647403, *Pik3ca*$^{wt/wt}$ Animal 2 CTL; ERS2647404, *Pik3ca*$^{wt/wt}$ Animal 3 CTL; ERS2647405, *Pik3ca*$^{wt/wt}$ Animal 4 CTL; ERS2647406, *Pik3ca*$^{H1047R/wt}$ Animal 1 CTL; ERS2647407, *Pik3ca*$^{H1047R/wt}$ Animal 2 CTL; ERS2647408, *Pik3ca*$^{H1047R/wt}$ Animal 3 CTL; ERS2647409, *Pik3ca*$^{H1047R/wt}$ Animal 4 CTL; ERS2647410, *Pik3ca*$^{wt/wt}$ Animal 1 INS; ERS2515253, *Pik3ca*$^{wt/wt}$ Animal 2 INS; ERS2515254, *Pik3ca*$^{wt/wt}$ Animal 3 INS; ERS2515255, *Pik3ca*$^{wt/wt}$ Animal 4 INS; ERS2515256, *Pik3ca*$^{H1047R/wt}$ Animal 1 INS; ERS2515257, *Pik3ca*$^{H1047R/wt}$ Animal 2 INS; ERS2515258, *Pik3ca*$^{H1047R/wt}$ Animal 3 INS; ERS2515259, *Pik3ca*$^{H1047R/wt}$ Animal 4 INS; ERS2515260. CTL refers to cultures incubated in minimal medium and insulin to cultures incubated with supraphysiological insulin dose. Human genomic sequencing data can be viewed at https://ega-archive.org, using the accession number EGAD00001008281. Source data are provided with this paper.

## Code availability

Code used for modeling has been made publicly available and can be found via GitHub at https://github.com/gp10/DriverClonALTfate. Code used to generate 2D histograms of clone sizes, displayed as heatmaps, is available in the CloneSizeFreq_2Dheat package via GitHub at https://github.com/gp10/CloneSizeFreq_2Dheat.

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

## Acknowledgements

We thank E. Choolun and T. Metcalf for their assistance in in vivo experiments, and Wellcome Sanger Institute RSF facilities for technical support. This work was supported by grants from the Wellcome Trust to the Wellcome Sanger Institute (098051, 296194 and 108413/A/15/D) and Cancer Research UK Programme Grants to P.H.J. (C609/A17257 and C609/A27326). A.H. benefited from the award of an EMBO long term fellowship (EMBO ALTF885-2015) and a Maria Zambrano Grant to attract international talent from Universitat de Barcelona and Ministerio de Universidades and cofunded with Next Generation EU funds. G.P. is supported by the Agencia Estatal de Investigación of Spain (grant no. PID2020-116163GA-I00). Work in the laboratory of B.V. is supported by Cancer Research UK (C23338/A25722) and the UK NIHR University College London Hospitals Biomedical Research Centre.

## Author contributions

A.H. and B.C. performed the experiments with support from E.A., I.C., A.P., D.F.-A. and C.B. A.H., A.K. and U.B. designed, performed and analyzed the CRISPR screen. K.M., B.V. and P.H.J. designed the mouse model and performed initial validations. C.K. and J.C.F. processed and analyzed the human sequencing data. G.P. analyzed clonal data and performed statistical analysis. M.W.J.H., S.H.O. and R.K.S. performed bioinformatics analysis. A.H., B.C. and P.H.J. wrote the manuscript with input from C.F. and B.V.

## Competing interests

B.C. is an employee of Altos Labs (Cambridge, UK). B.V. is a consultant for iOnctura (Geneva, Switzerland), Venthera (Palo Alto, CA, United States), Olema Pharmaceuticals (San Francisco, CA, United States) and Pharming (Leiden, The Netherlands). The other authors declare no competing interests.

## Additional information

**Extended data** is available for this paper at https://doi.org/10.1038/s41588-024-01891-8.

**Correspondence and requests for materials** should be addressed to Philip H. Jones.

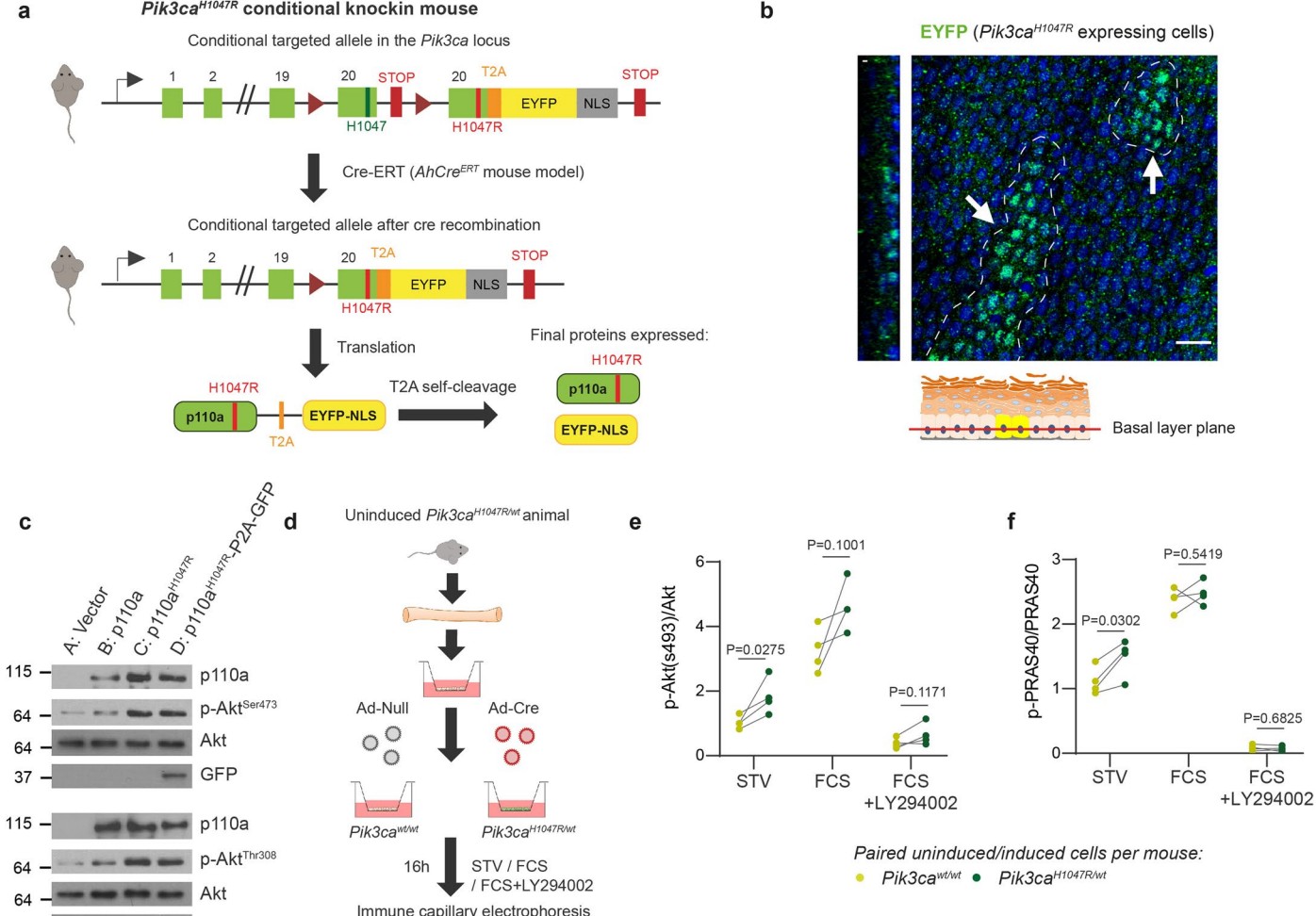

**Extended Data Fig. 1 | Generation of mice expressing a *Pik3ca^{H1047R}* mutation.**
**a**, Schematic of the conditional targeted *Pik3ca* allele. *Pik3ca* exon 20 was flanked
by *loxP* sites (triangles). The duplicate region of exon 20 encodes the H1047R
mutation, a self-cleaving T2A peptide and an enhanced Yellow Fluorescent
Protein (EYFP), followed by a nuclear localization signal (NLS). Wild-type p110α
protein is expressed until *Cre* mediated recombination, after which the allele
co-expresses p110α^{H1047R} mutant protein and EYFP-NLS. *Cre* recombination was
mediated by crossing the conditional mutant strain with *AhCre^{ERT}* mice which
express the Cre recombinase upon induction with β-naphthoflavone (BNF)
and tamoxifen (TAM). **b**, Typical top-down confocal image of esophageal
epithelial whole-mount from an Ahcre^{ERT}*Pik3ca^{H1047R/wt}* mouse, 3 months post-
induction. Optical section through basal cell layer is shown. *Pik3ca^{H1047R/wt}*
clones, showing nuclear EYFP (green) indicated by white arrows. Nuclei are
stained with DAPI (blue). Clones are delimited by dashed white lines. Scale bar,
20 µm. **c**, Immunoblots of protein expression of p110α, GFP and AKT (total and,

phosphorylated Ser^{473} (top) or Thr^{308} (bottom)) in NIH3T3 cells transfected
with empty vector, or vector carrying wild-type p110α, p110α^{H1047R} or
p110α^{H1047R}-P2A-GFP. Cells were starved for 24 h and lysed. Images are
representative of 3 separate experiments. **d-f**, Signaling in induced mutant
cells. Protocol (**d**): primary esophageal keratinocytes from *Pik3ca^{H1047R/wt}* mice
were infected either with null-adenovirus (uninduced *Pik3ca^{wt/wt}* controls) or
*Cre*-adenovirus (induced *Pik3ca^{H1047R/wt}*). Cells were treated in medium containing
either 0.1% serum and no added growth factors (Starved, STV), 20% serum and
growth factors (FCS) or FCS and the PI3K inhibitor LY294002 (50 µM), lysed and
the pAKT(Ser473)/Total-AKT (**e**) and p-PRAS40/Total-PRAS40 (**f**) analyzed by
immune capillary electrophoresis. Two-tailed ratio paired *t*-test. n = 4 paired
(Ad-Null and Ad-*Cre* infected) biological replicates (mice) per condition (paired
samples are linked by lines). Each dot is a biological replicate culture, lines link
uninduced and induced cultures derived from the same mouse.

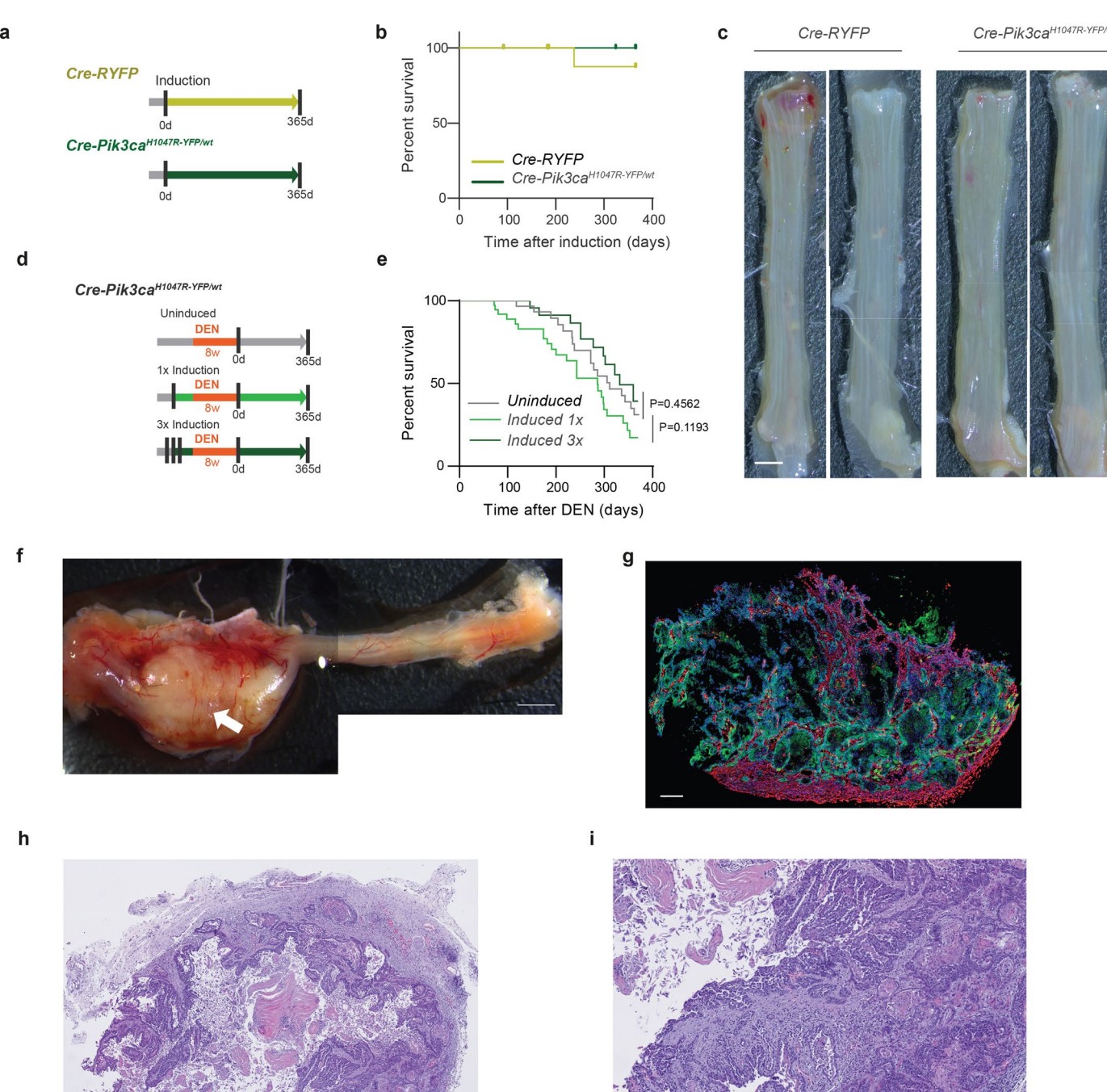

**Extended Data Fig. 2 | Heterozygous *Pik3ca*[H1047R] expression and mouse esophageal tumorigenesis. a-c**. Protocol (a) and survival (b) of *AhCre*[ERT] *Rosa26*[REYFP/wt] (*Cre-RYFP*) and *Ahcre*[ERT] *Pik3ca*[H1047R-YFP/wt] (*Cre-Pik3ca*[H1047R-YFP/wt]) mice induced and followed up to one-year post-induction. **c** Normal appearance of typical esophagi from a after longitudinal opening and flattening out. **d-e**, Carcinogenesis study. **d**. Protocol: *Cre-Pik3ca*[H1047R-YFP/wt] mice were induced 1 or 3 times and treated with diethylnitrosamine (DEN) for 8 weeks starting 4 weeks post-induction. Uninduced mice were used as controls. Tissues were collected before exceeding the permitted humane endpoint or at 1 year after DEN. **e**. Survival curves of animals on protocol in **d**. Log-rank (Mantel-Cox) test. **f-i**, *Pik3ca*[H1047R-YFP/wt] induced ESCC. Image in (**f**) shows unopened esophagus containing an ESCC (arrow) generated by DEN of an induced *Cre-Pik3ca*[H1047R-YFP/wt] mouse. (**g**) Immunofluorescence of a section of the tumor in (**f**) showing DAPI (blue), *Pik3ca*[H1047R/wt] (green) and Vimentin (red). Images in (**h**) and (**i**) depict Hematoxylin-Eosin staining of sections from the tumor in (**f**). Scale bars, 2 mm (**f**), 200 μm (**g**), 500 μm (**h**), 250 μm (**i**).

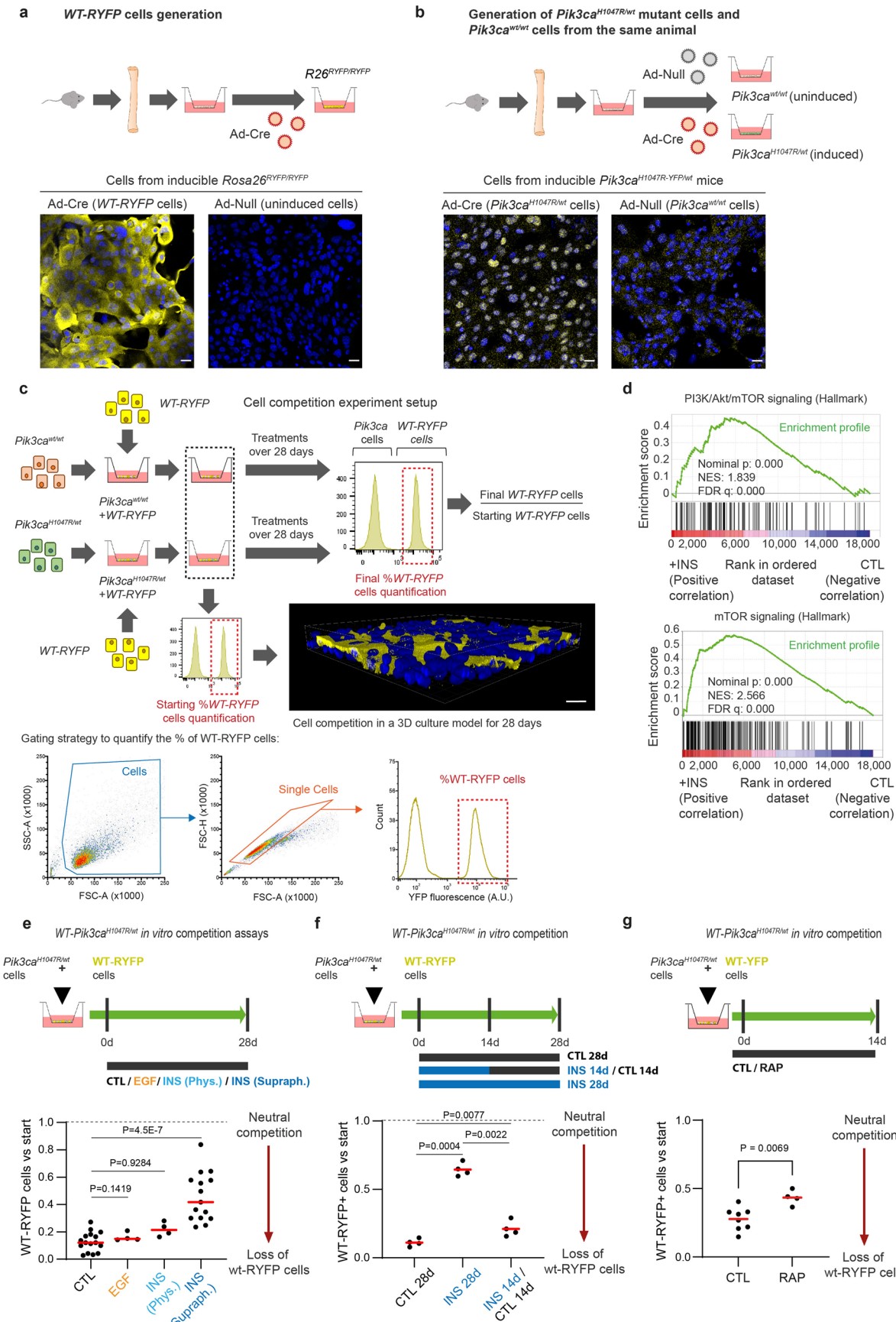

**Extended Data Fig. 3 | See next page for caption.**

**Extended Data Fig. 3 | Mechanism of *Pik3ca^{H1047R-YFP/wt}* clonal expansion.**
**a**, Multicolor lineage tracing in *Pik3ca^{H1047R-YFP/wt}Rosa26^{confetti/wt}AhCre^{ERT}* (*Cre-Pik3ca^{H1047R-YFP/wt}-Confetti*) animals. After recombination, one of four reporters is expressed. In some clones the *Pik3ca* locus is also recombined. **b**, Possible color combinations following immunostaining for GFP to detect *Pik3ca^{H1047R-YFP/wt}*. Only RFP+ in wild-type or *Pik3ca^{H1047R/wt}* clones can be distinguished. **c**, Top-down confocal image of *Cre-Pik3ca^{H1047R-YFP/wt}-Confetti* epithelium 84 days post-induction, showing RFP⁺ *Pik3ca^{wt/wt}* and YFP⁺/RFP⁺ *Pik3ca^{H1047R/wt}* clone. DAPI is blue. Scale bar, 20 μm. **d**, Heatmaps of number of basal and first suprabasal layer cells in RFP⁺ clones in *Cre-Pik3ca^{H1047R-YFP/wt}-Confetti* animals. Black dots and dashed lines show geometric median. Number of clones analyzed is shown, animal numbers are in brackets. Lower panels, differences between *Pik3ca^{wt/wt}* and *Pik3ca^{H1047R/wt}* clones. Two-tailed 2D Kolmogorov-Smirnov test. **e**, Average basal cells per clone from d, in clones with >1 basal cell. Dots indicate the average clone size per mouse. Black lines indicate mean±s.e.m. Lines represent the best fitting model shown in Fig. 3, shaded areas plausible intervals. **f**, Basal cell density in *Pik3ca^{H1047R/wt}* mutant clones and size-equivalent wild-type

areas in esophagus of C*re-Pik3ca^{H1047R-YFP/wt}* mice, 6 months post-induction. Red lines, mean ± S.D. Two tailed unpaired *t*-test (n = 63 areas per group, from 5 animals). **g**, Confocal images showing Caspase 3 (red), *Pik3ca^{H1047R/wt}* cells, green, and DAPI, blue staining in basal layer of a C*re-Pik3ca^{H1047R-YFP/wt}* mouse, 1-month post-high dose induction (Methods). Images representative of 4 mice. Left panel, UV irradiated positive control sample. The middle and right panels are images from the same tissue showing apoptotic cell (red arrow) and *Pik3ca^{H1047R/wt}* clone (green arrow). Scale bars, 20 μm. Architecture **h** and dynamics **i** of mouse esophageal epithelium. **h**, The lowest basal layer contains proliferating progenitor cells and post-mitotic differentiating cells, which will migrate upwards through the suprabasal layers until they are shed. **i**, Progenitor divisions may generate two differentiating cells, two progenitor cells or one of each type. The probabilities of each outcome are balanced, producing equal numbers of progenitor and differentiating cells across the epithelium, maintaining tissue homeostasis. Tilting cell fate towards proliferation, produces excess mutant progenitors resulting in clone growth.

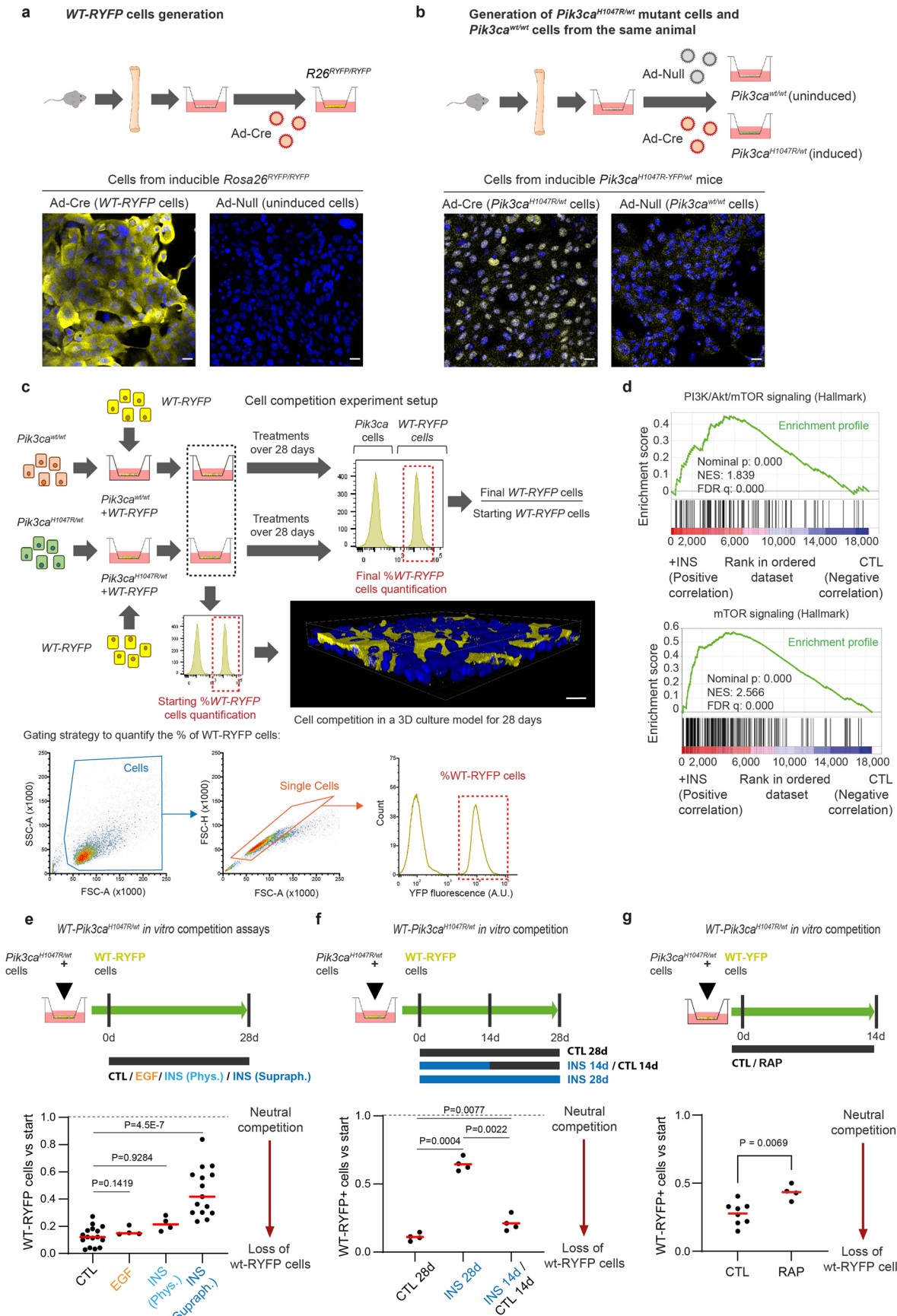

**Extended Data Fig. 4 | See next page for caption.**

**Extended Data Fig. 4 | Mutant versus wild-type cell competition in culture.**
**a-b**, Generation of induced *WT-RYFP* (**a**) and uninduced/induced *Pik3ca^{H1047R/wt}*
(**b**) epithelioid cultures. Primary esophageal keratinocytes from uninduced
*Rosa26^{RYFP/RYFP}* (**a**) or uninduced *Pik3ca^{H1047R-YFP/wt}* animals (**b**) were incubated with
*Cre*-expressing adenovirus (Ad-*Cre*) or null adenovirus (Ad-Null). Right panels,
representative images of Ad-*Cre* or Ad-Null treated cultures stained for YFP
(yellow) and DAPI (nuclei, blue). Scale bars, 20 μm. **c**, In vitro cell competition
protocol. *Pik3ca^{H1047R/wt}* and *Pik3ca^{wt/wt}* keratinocytes from b, are mixed with
*WT-RYFP* cells from a, and maintained at confluence. Proportion of *WT-RYFP*
cells is quantified by flow cytometry, gating strategy is shown. The proportion
of *WT-RYFP* cells after treatment was normalized to the initial *WT-RYFP* cell
proportion. Scale bars, 20 μm. **d**, GSEA histograms of PI3K/Akt/mTOR (top) and
mTOR signaling (bottom) Hallmark gene sets comparing RNA-seq data from
control (CTL) and 5 μg/ml insulin (+INS) treated wild-type cells from the same
animals. The nominal p-value, the normalized enrichment score (NES) and the
false discovery rate (FDR) q-value are indicated. n = 4 independent replicates
per condition from one animal each. **e-g**, Experimental scheme (top panel) and
quantification by flow cytometry (bottom panel) of the proportion of *WT-RYFP*
cells mixed with *Pik3ca^{H1047R/wt}* cells at the end of the experiment *versus* the
start of each experiment. **e**, Cells were treated either in minimal FAD medium
or treated with 10 ng/ml EGF, 2 ng/ml Insulin (physiological) or 5 μg/ml Insulin
(supraphysiological) for one month. Each dot represents a biological replicate
(n = 4-16 primary cultures from individual animals, per condition). **f**, Cells were
cultured for 28 days in minimal medium (CTL), 28 days in minimal medium with
5 μg/ml insulin (INS), or 14 days in minimal medium with 5 μg/ml insulin followed
by 14 days in minimal medium. Each dot represents a biological replicate (n = 4
primary cultures from individual animals, per condition). **g**, Cells were treated
either in minimal FAD medium minus/plus Rapamycin 500 nM for 14 days. Each
dot represents a biological replicate (n = 4-8 primary cultures from different
animals, per condition). Red lines indicate mean values. Two-tailed paired *t*-test.

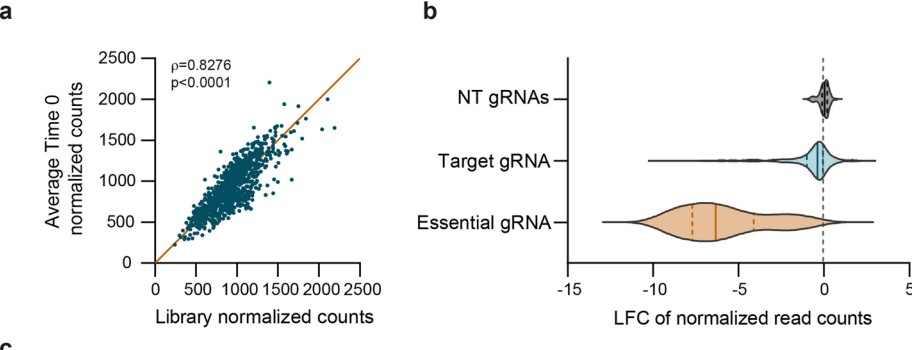

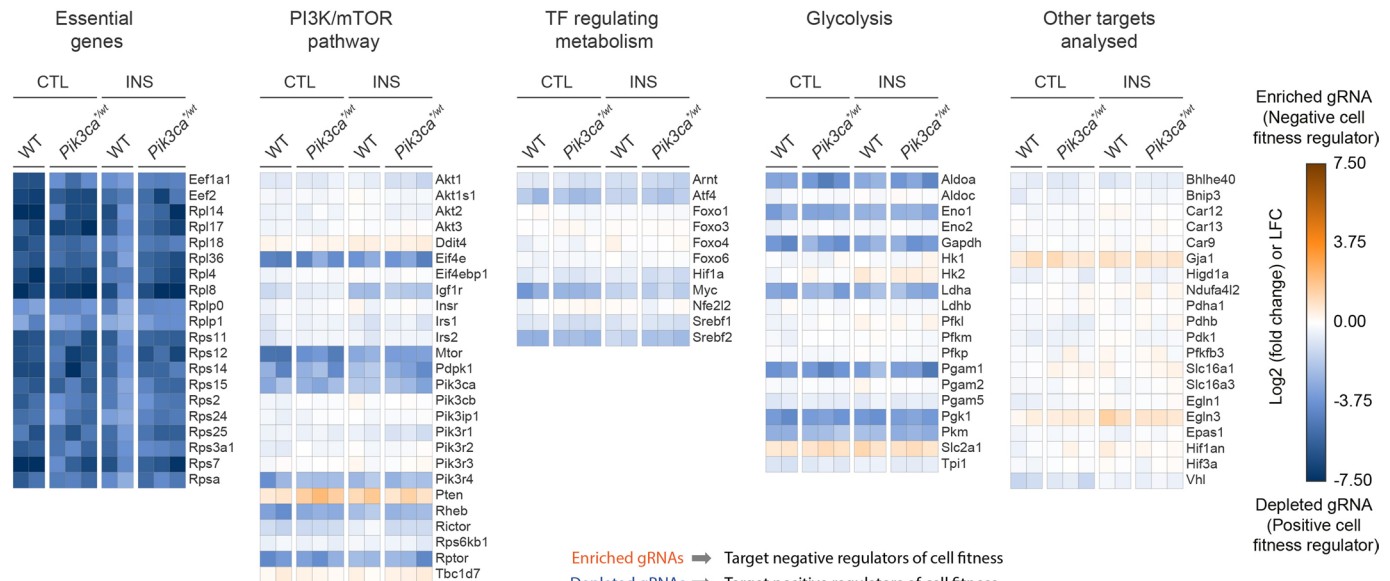

Enriched gRNAs ➡ Target negative regulators of cell fitness
Depleted gRNAs ➡ Target positive regulators of cell fitness

**Extended Data Fig. 5 | CRISPR screening of targets affecting mutant cell fitness. a-b,** CRISPR screening in uninduced (*Pik3ca^{wt/wt}*, WT) cells in minimal medium (CTL). **a,** Correlation between normalized read counts for the sequenced library and the average normalized read counts at Time 0. Orange line, linear regression between samples with the Pearson's coefficient and two-tailed p-value of the correlation. **b,** Violin plots of distribution of average $\log_2$ fold change between Time 3 and Time 0 for each gRNA. n = 2 biological replicates, 10 gRNA

per gene. **c,** Heatmaps of average $\log_2$ (fold change per gene of gRNA abundance after 3 weeks versus Time 0. Genes are grouped by pathway. Heatmaps show the enrichment of gRNAs targeting each gene at 3 weeks over 0 weeks in either minimal medium (CTL) or minimal medium supplemented with 5 µg/ml Insulin (INS), and either uninduced (WT) or induced (*Pik3ca^{+/wt}*) cells from *Rosa26^{Cas9/wt} Pik3ca^{H1047R/wt}* mice. Each column is a biological replicate. n = 2-3 biological replicates, 10 gRNA per gene.

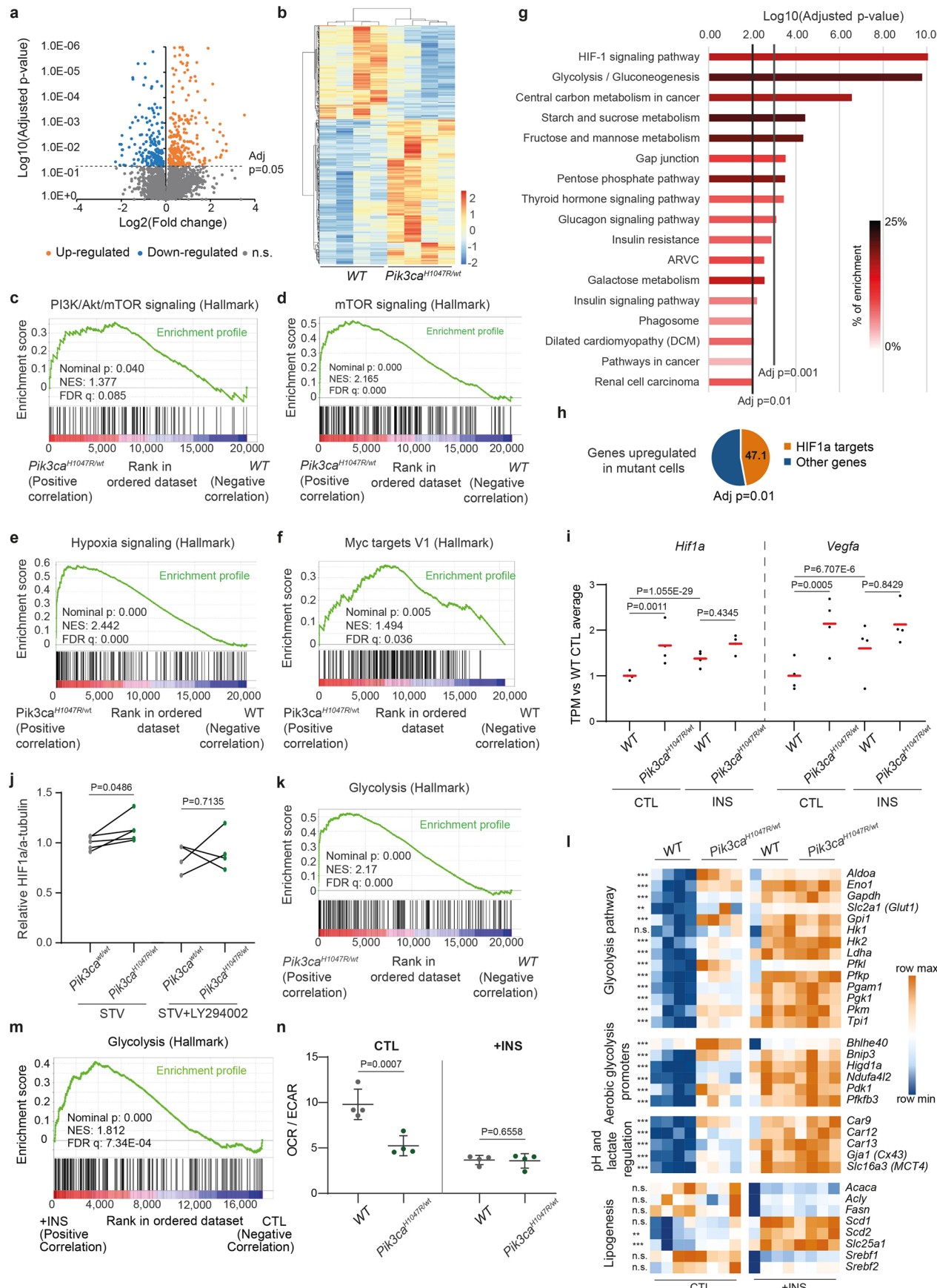

**Extended Data Fig. 6 | See next page for caption.**

**Extended Data Fig. 6 | *Pik3ca^{HI047R/wt}* cells activate Hif1α pathway and glycolysis. a-b**, RNA-seq analysis comparing *Pik3ca^{wt/wt}* (WT, uninduced) and *Pik3ca^{HI047R/wt}* mutant cultures from the same animal, in minimal medium. Heatmap shows altered transcripts (adjusted p < 0.05). n = 4 mice per group, paired induced-uninduced samples. Wald test corrected for multiple testing. **c-f**, Gene set enrichment analysis histograms of PI3K/Akt/mTOR (**c**), mTOR (**d**) Hypoxia (e) and myc targets (**f**) signaling Hallmark gene sets in WT and mutant cultures. Nominal p-value, normalized enrichment score (NES) and false discovery rate (FDR) q-value are indicated. **g**, KEGG pathway enrichment analysis for up-regulated genes in mutant vs WT (p-value adjusted for multiple hypotheses testing using the Benjamini-Hochberg method<0.01). Intensity indicates pathway enrichment. **h**, Proportion of HIF1α target genes significantly up-regulated (adjusted p-value (as in g)<0.01) in mutant vs WT. **i**, Transcripts per million (TPM) relative to WT CTL condition of *Hif1a* and *Vegfa* in *Pik3ca^{wt/wt}* and *Pik3ca^{HI047R/wt}* cells in control (CTL) versus cultures treated with 5 μg/ml insulin (INS). n = 4 biological replicates per condition. Red lines, median

values. Wald test corrected for multiple testing. **j**, Uninduced (*Pik3ca^{wt/wt}*) and induced (*Pik3ca^{HI047R/wt}*) cells were cultured in starvation media (STV) +/− PI3K inhibitor LY294002 (0.5 μM), and HIF1α/α-Tubulin analyzed by immune capillary electrophoresis. HIF1α/α-Tubulin ratio versus the average of the WT STV shown. Two-tailed ratio paired *t*-test. n = 5 paired biological replicates per condition (lines link paired samples). **k**, GSEA histogram of Glycolysis Hallmark gene set comparing induced *Pik3ca^{HI047R/wt}* and WT cells in minimal medium. n = 4 biological replicates per condition. Statistics as in c-f. **l**, Heatmaps comparing WT and *Pik3ca^{HI047R/wt}* cultures in CTL and INS conditions. Two-tailed Wald test corrected for multiple testing comparing WT and *Pik3ca^{HI047R/wt}* CTL conditions. ***p < 0.001, n.s.= not significant. **m**, GSEA histograms of Glycolysis Hallmark set comparing control (CTL) and 5 μg/ml insulin (+INS) treated wild-type cells. n = 4 biological replicates per condition. Statistics as in c-f. **n**, Basal OCR to ECAR ratios of WT and *Pik3ca^{HI047R/wt}* cultures in CTL or INS conditions. Dots, average per animal (n = 4 mice). OCR/ECAR ratios are mean ± SD. Two-tailed paired *t*-test.

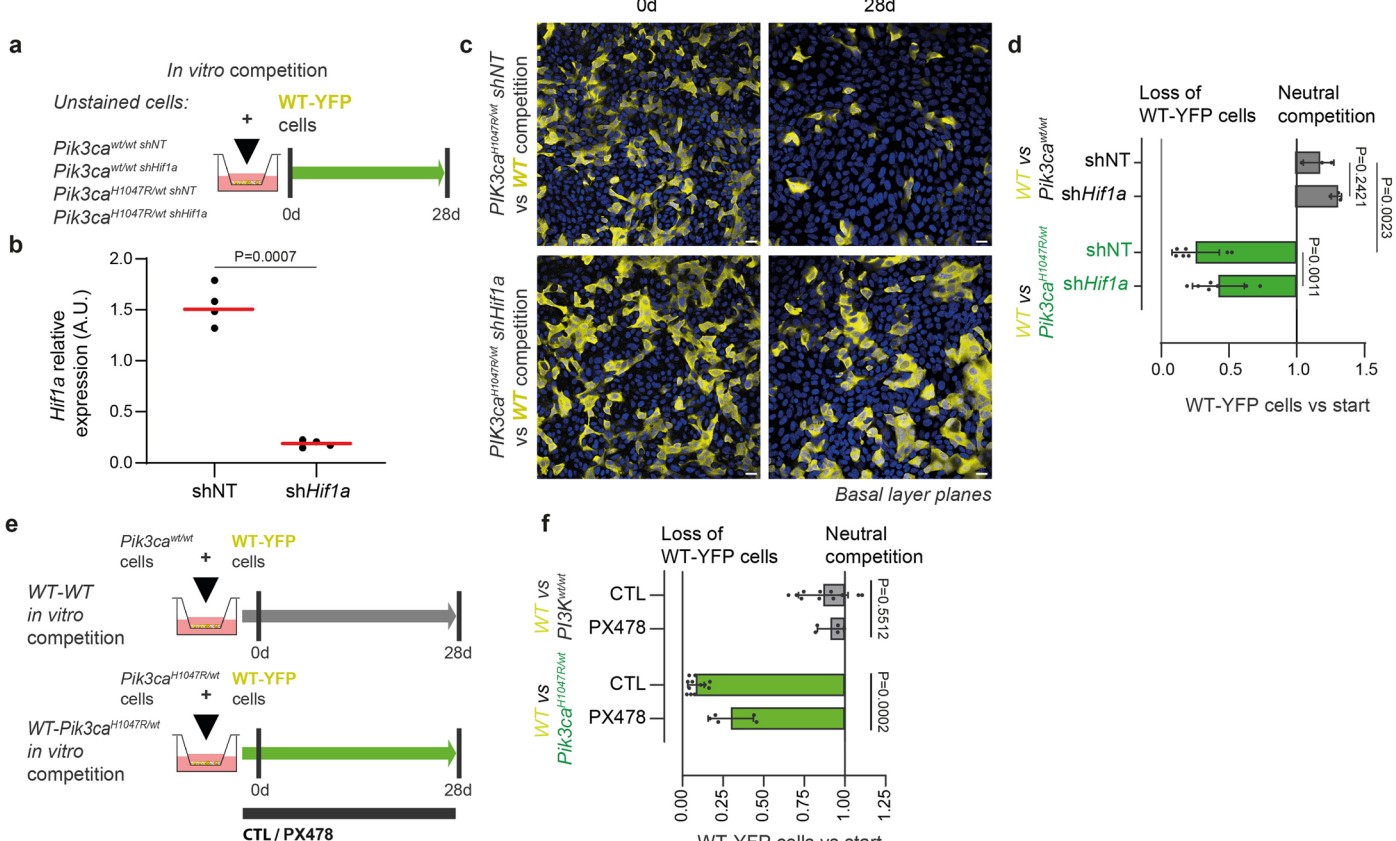

**Extended Data Fig. 7 | Fitness of *Pik3ca^{H1047R/wt}* cells depends on Hif1α.**
**a-d**, Protocol (**a**): *Pik3ca^{wt/wt}* (wild-type) or *Pik3ca^{H1047R/wt}* mutant cells expressing shRNA against *Hif1a* (sh*Hif1a*) or control shRNA (shNT) were mixed with *WT-RYFP* and cultured for 28 days in minimal medium. **b**, *Hif1α* mRNA levels in sh*Hif1a* or shNT cells. Each dot is a culture from a different mouse (n = 4 mice). Red lines, median values. Two tailed paired *t*-test. **c**, Representative confocal images showing basal layer of an epithelioid culture generated as in **a**. n = 3 biological replicates. WT-RYFP cells, yellow, DAPI, blue. Scale bar, 20 μm. **d**, Proportion of *WT-RYFP* cells at 28 days versus day 0. Each dot is a culture from a different animal. n = 3-7 cultures per condition. Two-tailed paired *t*-test. **e-f**, In vitro cell competition of *WT-RYFP* mixed with *Pik3ca^{H1047R/wt}* cells or uninduced controls from the same mice. Cultures were then treated with HIF1α inhibitor PX-478 (10 μM) or vehicle for 28 days. (**f**). Each dot is mean of a culture from a different animal (n = 5-11). Bars are SD. Two tailed unpaired *t*-test.

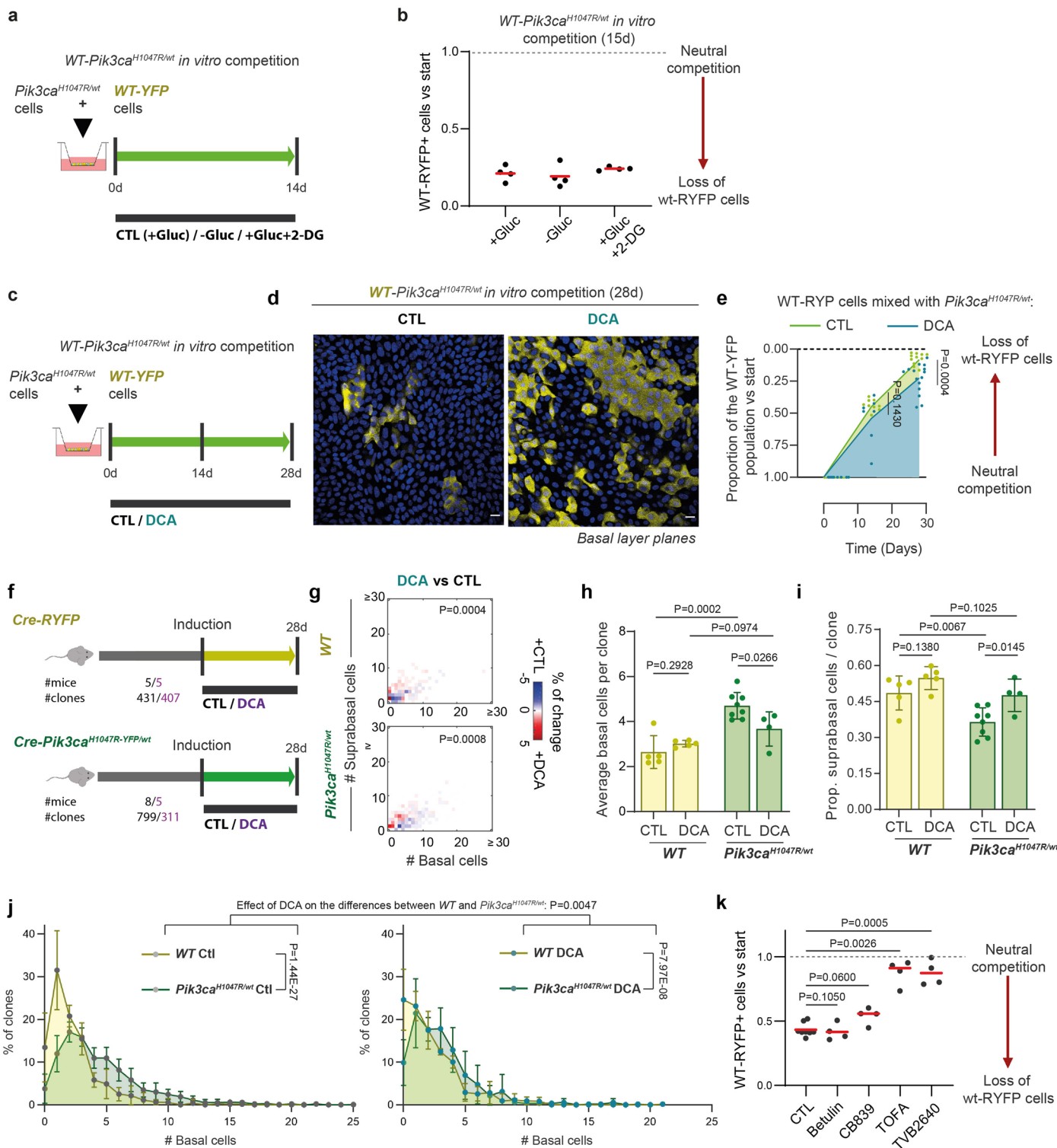

**Extended Data Fig. 8 | See next page for caption.**

**Extended Data Fig. 8 | Metabolic changes downstream of *Pik3ca^{H1047R/wt}* expression. a-b**, Proportion of WT-RYFP cells in mixed cultures with *Pik3ca^{H1047R/wt}* cells, after 15 days in minimal media normalized to baseline value. +/−Gluc indicates culture with/without glucose and 2-DG indicates culture with 5 mM 2-deoxyglucose. Each dot is a culture from a different animal, lines are mean values. n = 4 cultures per condition. **c-e**, *In vitro* cell competition assay. **c**, Protocol: Confluent mixed cultures of *WT-RYFP* and induced *Pik3ca^{H1047R/wt}* cells with/without DCA 25 mM for 28 days in minimal medium. **d**, Representative confocal images of basal cell layer of cultures from **c**, *WT-RYFP* cells, yellow, DAPI, blue. Scale bar, 20 μm. **e**, Flow cytometric analysis from mixed cultures in **c**. Each dot is a culture from a different mouse and lines connect mean values (n = 7-15 cultures). Two-tailed paired *t*-test. **f**, Protocol: *Cre-RYFP* control and *Cre-Pik3ca^{H1047R-YFP/wt}* mice were induced and treated with DCA. Clones with >1 basal cell sizes were analyzed at 28 days. Mice and clone numbers are shown. **g**, Heatmaps showing the differences between each treatment versus control in *Cre-RYFP* (upper panels) or *Cre-Pik3ca^{H1047R-YFP/wt}* (lower panels) mice from **f**. Two-tailed 2D Kolmogorov-Smirnov test. **h-i**, Average basal clone sizes **h** and proportion of first suprabasal layer cells **i** for each strain and treatment from **f**. Data includes clones with at least one basal cell. Bars are SD. Two-tailed unpaired *t*-test. **j**, Wild-type and *Pik3ca^{H1047R}* mutant basal cell clone size distributions from untreated or DCA-treated *Cre-RYFP* and *Cre-Pik3ca^{H1047R-YFP/wt}* mice, respectively, 28 days post-induction. n = 311-799 clones from 5-8 animals per condition. Dots indicate mean and lines standard deviation. Two-tailed Kolmogorov-Smirnov test and Contrast ART-C Post-hoc test of differences of differences between distributions. **k**. Proportion of *WT-RYFP* cells in mixed cultures with *Pik3ca^{H1047R/wt}* cells, at 15 days normalized to baseline value. Cells were culture in minimal media (CTL) or with Betulin (6 μg/ml), CB839 (10 μM), TOFA (30 μM) or TVB-2640 (0.1 μM). Each dot represents a culture from a different animal, lines correspond to mean values. n = 4 cultures from individual animals per condition. Two tailed paired *t*-test.

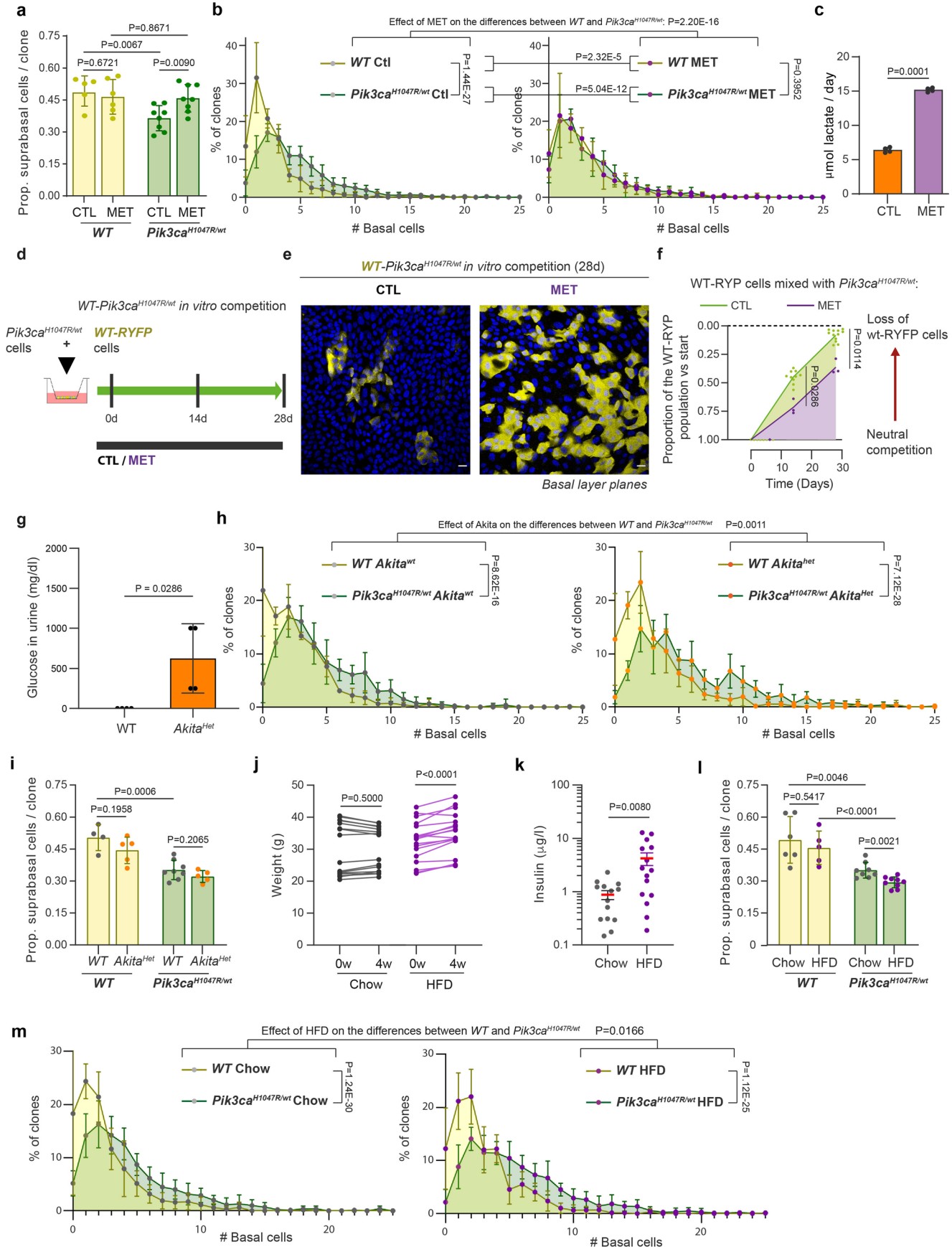

**Extended Data Fig. 9 | See next page for caption.**

**Extended Data Fig. 9 | Metabolic conditions alter the Pik3ca$^{H1047R/wt}$ fitness *in vivo*. a-b**, *Cre-RYFP* and *Cre-Pik3ca$^{H1047R-YFP/wt}$* mice were induced and treated with or without metformin (MET). First suprabasal layer cells per clone (**a**) and basal clone size distributions (**b**) analyzed at 28 days. Dots indicate mean and lines S.D. in **b**. Two-tailed unpaired *t*-test for suprabasal cells per clone. Two-tailed Kolmogorov-Smirnov test to compare wild-type and mutant basal clone distributions. Contrast ART-C Post-hoc test of differences of differences between distributions. n = 431-917 clones from 5-10 animals per condition. **c**, Lactate secretion in wild-type cells in minimal medium in control conditions (CTL) or with 2.5 mM MET. Bars indicate mean. Two-tailed paired *t*-test. **d-f**, *In vitro* competition assay. **d**, Protocol: Confluent cultures of *WT-RYFP* and *Pik3ca$^{H1047R/wt}$* cells in minimal medium with/without MET for 28 days. **e**, Representative confocal image of basal layer of culture from **d**. WT-RYFP cells, yellow, DAPI, blue. Scale bar, 20 µm. **f**, Flow cytometric analysis of cultures in d. Each dot represents a biological replicate, lines connect mean values (n = 3–12). Two-tailed paired

*t*-test. **g**, Urine glucose levels in mice from Fig. 6d. Two-tailed Mann-Whitney test. n = 4 mice per group. Mean ± SD. **h**, Basal cells per clone in Fig. 6d. Dots and bars indicate mean and S.D. Two-tailed Kolmogorov-Smirnov and Contrast ART-C Post-hoc tests. **i**, Average proportion of suprabasal cells per clone for each strain and treatment from Fig. 6d, only the first suprabasal cell layer was counted. Each dot corresponds to one animal. Bars are mean ± SD. n = 431-917 clones from 5-10 animals per condition. Two-tailed unpaired *t*-test. **j**, Body weight measured at weeks 0 and 4 post diets. Each dot corresponds to one animal, lines link weights from same animal (n = 10-16 mice). Two-tailed paired *t*-test. **k**, Blood insulin levels at the end of experiment in **j**. Error bars are mean±s.e.m. Two-tailed unpaired *t*-test. **l**, Average proportion of suprabasal cells per clone for each strain and treatment from Fig. 6g, only first suprabasal cells were counted. Each dot corresponds to one animal. Bars are mean ± S.D. Two-tailed unpaired *t*-test. **m**, Distribution of basal cells per clone from Fig. 6g. Dots and bars indicate mean and S.D. Two-tailed Kolmogorov-Smirnov and Contrast ART-C Post-hoc tests.

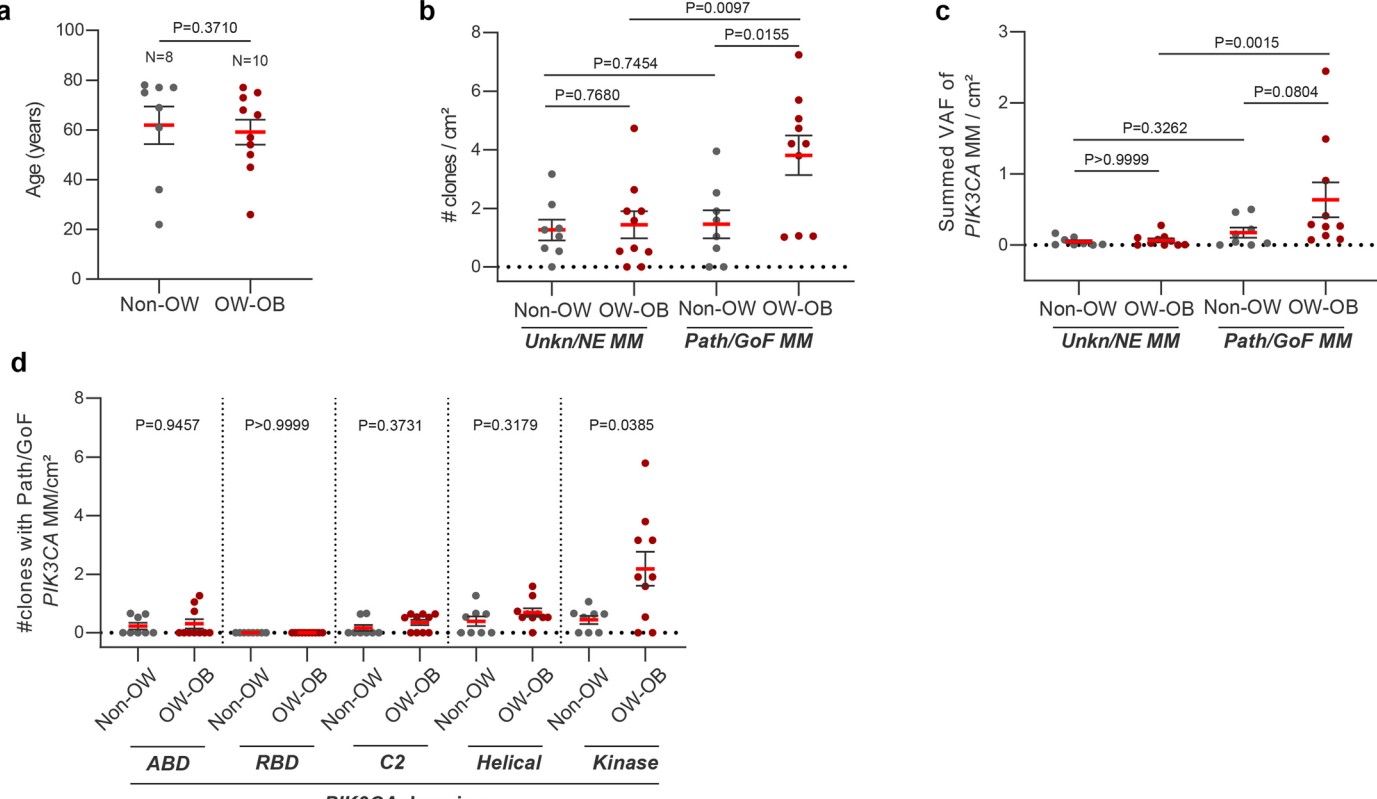

**Extended Data Fig. 10 | Higher density of *Pik3ca* missense mutations in overweight humans. a**, Age distribution of all human donors, including donors from Fig. 1. Lines are mean±s.e.m. Two-tailed unpaired *t*-test. **b-d**, Human donors were classified into OW-OB (n = 10, BMI ≥ 25 kg m²) and non-OW (n = 8, BMI < 25 kg m²). *PIK3CA* missense mutations (MM) were classified as Path/GoF and Unkn/NE (see Methods) and analyzed separately in OW-OB and Non-OW groups. Each dot represents a donor. **b**, Number of *PIK3CA* missense mutant

clones detected/cm² in the specified groups. Bars are mean±s.e.m. Two-tailed unpaired *t*-test. **c**, Summed variant allele frequency (VAF) of all Path/GoF or Unkn/NE *PIK3CA* missense mutations per cm² for each donor. Lines are mean±s.e.m. Two-tailed Mann-Whitney test. **d**, Number of clones/cm² carrying Path/GoF *PIK3CA* missense mutations in each p110α domain. Bars are mean±s.e.m. Two-tailed Mann-Whitney test.

# natureportfolio

# Reporting Summary

## Statistics

For all statistical analyses, confirm that the following items are present in the figure legend, table legend, main text, or Methods section.

| n/a | Confirmed | |
|---|---|---|
| ☐ | ☒ | The exact sample size (*n*) for each experimental group/condition, given as a discrete number and unit of measurement |
| ☐ | ☒ | A statement on whether measurements were taken from distinct samples or whether the same sample was measured repeatedly |
| ☐ | ☒ | The statistical test(s) used AND whether they are one- or two-sided<br>*Only common tests should be described solely by name; describe more complex techniques in the Methods section.* |
| ☐ | ☒ | A description of all covariates tested |
| ☐ | ☒ | A description of any assumptions or corrections, such as tests of normality and adjustment for multiple comparisons |
| ☐ | ☒ | A full description of the statistical parameters including central tendency (e.g. means) or other basic estimates (e.g. regression coefficient) AND variation (e.g. standard deviation) or associated estimates of uncertainty (e.g. confidence intervals) |
| ☐ | ☒ | For null hypothesis testing, the test statistic (e.g. *F*, *t*, *r*) with confidence intervals, effect sizes, degrees of freedom and *P* value noted<br>*Give P values as exact values whenever suitable.* |
| ☐ | ☒ | For Bayesian analysis, information on the choice of priors and Markov chain Monte Carlo settings |
| ☒ | ☐ | For hierarchical and complex designs, identification of the appropriate level for tests and full reporting of outcomes |
| ☐ | ☒ | Estimates of effect sizes (e.g. Cohen's *d*, Pearson's *r*), indicating how they were calculated |

*Our web collection on statistics for biologists contains articles on many of the points above.*

## Software and code

Policy information about availability of computer code

| Data collection | Confocal images were obtained using the Leica Adquisiton Software LAS X. Flow cytometry data was obtained using a BD LSRFortessa and BD FACSDiva•M Software (BO-Biosciences). Immune capillary electrophoresis was performed using Wes Simple'M (ProteinSimple, P/N 031-108) with the Compass software. OCR and ECAR were assayed using the XF24 software in a Seahorse XF-24 analyser. DNA sequencing was performed on an Illumina MiSeq platform. RNA sequencing was performed on an Illumina HiSeq 2500 platform. |
|---|---|
| Data analysis | - Confocal Z stack images were rendered and analyzed with Imaris 4.3 (Bitplane), ImageJ software, or Volocity 6.3 Software (Perkin Elmer).<br>- Flow cytometry data was analysed using the FlowJo software (version 10.5.3).<br>- Immune capillary electrophoresis analysis was performed using the Compass software.<br>- OCR and ECAR were analysed using the XF24 software.<br>- RNA sequencing: Reads were mapped using STAR 2.5.3a; the alignment files were sorted and duplicate-marked using Biobambam2 2.0.54; the read summarization performed by the htseq-count script from version 0.6.lpl of the HTSeq framework; Gene set enrichment was analyzed with GSEA software, using the Hallmarks gene sets of the Molecular Signature Database (MSigDB) version 4.0 provided by the Broad Institute (http://www.broad.mit.edu/gsea/), following the standard procedure described on the GSEA user guide (http://www.broadinstitute.org/gsea/doc/GSEAUserGuideFrame.html); Differential gene expression was analyzed using the DEBrowser tool (https://debrowser.umassmed.edu/); Heatmaps were generated from the TPM values and build using ClustVis (https://biit.cs.ut.ee/clustvis/) and Morpheus tools (https://software.broadinstitute.org/morpheus/); Kyoto Encyclopedia of Genes and Genomes (KEGG) pathway enrichment analysis was performed uploading the significantly upregulated gene list (p<0.05) into the Enrichr tool (https://amp.pharm.mssm.edu/Enrichr/); Venn diagrams were generated using the Venn Diagrams tool (https://www.biotools.fr/misc/venny); MA plots were generated using GraphPad Prism 8v8.3.1.<br>- DNA sequencing analysis: ShearwaterML algorithm from the deepSNV package (vl.21.3, https://github.com/gerstung-lab/deepSNV), ClinVar mutation database (https://clinvarminer.genetics.utah.edu/variants-by-gene/PIK3CA), Clinical Knowledgebase (https://ckb.jax.org/gene/ |

- Lineage tracing: Code used for modelling has been made publicly available and can be found at https://github.com/gpl0/DriverClonALTfate. Code used to generate two-dimensional histograms of clone sizes, displayed as heatmaps is available in the CloneSizeFreq_2Dheat package (https ://githu b.com/gp 10/CloneSizeFreq_2 Dheat).
- Crispr screen: targets were selected using the ChopChop tool v3; Gini coefficients and Lorenz curves were calculated for all samples using the "Ineq" package in R; Enrichment analysis was done using the MAGeCK 0.5.9 software package; Enrichment scores were further analysed using the MAGeCKFLUTE package) and visualised using Graph Pad.
- Statistics: GraphPad Prism software v8.3.l, Matlab, ARTool [R] package vignette, Matlab, CloneSizeFreq_2Dheat package (https://github.com/gp10/CloneSizeFreq_2Dheat.

For manuscripts utilizing custom algorithms or software that are central to the research but not yet described in published literature, software must be made available to editors and reviewers. We strongly encourage code deposition in a community repository (e.g. GitHub). See the Nature Portfolio guidelines for submitting code & software for further information.

## Data

Policy information about availability of data

All manuscripts must include a data availability statement. This statement should provide the following information, where applicable:
- Accession codes, unique identifiers, or web links for publicly available datasets
- A description of any restrictions on data availability
- For clinical datasets or third party data, please ensure that the statement adheres to our policy

The sequencing data sets in this study are publicly available at the European Nucleotide archive (ENA) Accession numbers for RNAseq data on https://www.ebi.ac.uk/ena are as follows: In vivo samples: ERS14340821, ERS14340822, ERS14340823, ERS14340824. In vitro samples: ERS2515249, ERS2515250, ERS2515251, ERS2515252. Accession numbers for targeted DNA sequencing of SCA is ERP107379. Data used to generate each figure is available in Supplementary table 1.

## Human research participants

Policy information about studies involving human research participants and Sex and Gender in Research.

| | |
|---|---|
| Reporting on sex and gender | Our study used human samples from both sexes. |
| Population characteristics | Deceased organ donors from whom organs were being retrieved for transplantation. The samples a small unselected sample of organ donors in the Eastern region of England. |
| Recruitment | Written informed consent was obtained from relatives of deceased organ donors from whom organs were being retrieved for transplantation |
| Ethics oversight | Ethical approval was obtained from the Cambridge South Ethics Committee, Research Ethics Committee reference: 15/EE/0152 NRES Committee East of England - Cambridge South |

Note that full information on the approval of the study protocol must also be provided in the manuscript.

## Field-specific reporting

Please select the one below that is the best fit for your research. If you are not sure, read the appropriate sections before making your selection.

☒ Life sciences ☐ Behavioural & social sciences ☐ Ecological, evolutionary & environmental sciences

For a reference copy of the document with all sections, see nature.com/documents/nr-reporting-summary-flat.pdf

## Life sciences study design

All studies must disclose on these points even when the disclosure is negative.

| | |
|---|---|
| Sample size | Sample size was not predetermined by statistical methods. Sample size for for lineage tracing and carcinogenesis studies was determined by previous studies (PMID: 22821983, PMID: 24814514, PMID: 31327664, PMID: 27548914, PMID: 34646013, PMID: 36266286 and PMID: 32424351). |
| Data exclusions | No data was excluded from the study. |
| Replication | Different epithelioid cultures or mice were considered as independent experimental unit. All attempts at replication were successful. |
| Randomization | Cultures and mice were randomly allocated in experimental groups. |
| Blinding | Investigators were not blinded as to sample IDs during analyses. The staining pattern of mutant clones and the evolving clone size distributions reveal the genotype and the time point, preventing effective blinding. |

# Reporting for specific materials, systems and methods

We require information from authors about some types of materials, experimental systems and methods used in many studies. Here, indicate whether each material, system or method listed is relevant to your study. If you are not sure if a list item applies to your research, read the appropriate section before selecting a response.

## Materials & experimental systems

| n/a | Involved in the study |
|-----|-----------------------|
| ☐ | ☒ Antibodies |
| ☐ | ☒ Eukaryotic cell lines |
| ☒ | ☐ Palaeontology and archaeology |
| ☐ | ☒ Animals and other organisms |
| ☒ | ☐ Clinical data |
| ☒ | ☐ Dual use research of concern |

## Methods

| n/a | Involved in the study |
|-----|-----------------------|
| ☒ | ☐ ChIP-seq |
| ☐ | ☒ Flow cytometry |
| ☒ | ☐ MRI-based neuroimaging |

## Antibodies

| Antibodies used | |
|-----------------|---|
| | **Microscopy/Flow Cytometry** |

| | | | |
|---|---|---|---|
| GFP | Life technologies | A10262 | Polyclonal |
| Caspase 3 | Abcam | ab2302 | Polyclonal |
| Vimentin | Abcam | ab92457 | EPR3776 |
| | | | |
| ITGA6 | Biolegend | 313610 | GoH3 |
| Alexa Fluor 488 | Jackson | 703-545-155 | |
| Donkey Anti-Chicken | ImmunoResearch | | |

**Western Blotting/Protein Simple**

| | | | |
|---|---|---|---|
| P-Akt S473 | CST | 4060S | D9E |
| Akt | CST | 4691S | C67E7 |
| P-Akt T308 | CST | 2965S | C31E5E |
| PRAS40 | CST | 2691T | C77D7 |
| P-PRAS40 | CST | 2997T | D23C7 |
| HIF1A | Novus Biologicals | NB100-134 | Polyclonal |
| GSK3b | CST | 9315S | 27C10 |
| PGSK3b | CST | 9322S | D3A4 |
| PS6 | CST | 2211 | Polyclonal |
| p110a | Abcam | Ab152155 | Polyclonal |
| GFP | Abcam | Ab290 | Polyclonal |
| aTubulin | CST | 2125S | 11H10 |

| Validation | |
|------------|---|
| | GFP: Detects GFP with no cross reactivity with mammalian proteins, data on manufacturer's website. |
| | Caspase 3: staining induced by irradiation of mouse esophagus (PMC6739485) |
| | ITGA6: Staining undetectable in knockout mice (PMID: 8673141) |
| | Vimentin: Knockout validated, Abcam website |
| | P-Akt S473: Validation by inhibitors, CST Website |
| | Akt:  Western blot in multiple cell lines and vs recombinant proteins CST Website |
| | P-Akt T308: Validation by inhibitors, CST Website |
| | PRAS40: siRNA knockdown abolishes WB band PMID: 23460019 |
| | P-PRAS40: Validation by inhibitors, CST Website |
| | HIF1A: siRNA knockdown, this paper. |
| | GSK3b: Data on CST Website. |
| | PGSK3b: Data on CST Website. |
| | PS6: WB and inhibitor validation CST Website |
| | aTubulin: Data on CST Website. |
| | p110a:  Product correct size on WB, Ab no longer available. |

## Eukaryotic cell lines

Policy information about cell lines and Sex and Gender in Research

| Cell line source(s) | NIH 3T3 Cell line (ATCC CRL-1658) |
|---------------------|-----------------------------------|

| Authentication | None of the cell lines used were authenticated |
|---|---|
| Mycoplasma contamination | The cells tested negative for mycoplasma. |
| Commonly misidentified lines<br>(See ICLAC register) | No commonly misidentified cell lines were included in the study. |

# Animals and other research organisms

Policy information about studies involving animals; ARRIVE guidelines recommended for reporting animal research, and Sex and Gender in Research

| Laboratory animals | All mouse strains were maintained on a C57/BL6 genetic background.<br>Strains used included:<br>Cyp1A1creERT<br>Rosa26FlYFP<br>Ins2Akita/wt<br>Pik3caflH1047RT2AYFP-NLS/wt<br>Rosa26Cas9P2AGFP<br>Rosa26FllConfetti<br>and crosses of the above lines. |
|---|---|
| Wild animals | The study did not involve wild animals. |
| Reporting on sex | Cultures were stablished from animals of both sexes.  No sex specific differences were observed. |
| Field-collected samples | The study did not involve samples collected from the field. |
| Ethics oversight | All mouse experiments were ethically reviewed and approved by the Welcome Sanger Institute Ethics Committee and conducted according to UK government Home Office project licences PPL22/2282, PPL70/7543, and PPL4639B40. |

Note that full information on the approval of the study protocol must also be provided in the manuscript.

# Flow Cytometry

## Plots

Confirm that:

☒ The axis labels state the marker and fluorochrome used (e.g. CD4-FITC).

☒ The axis scales are clearly visible. Include numbers along axes only for bottom left plot of group (a 'group' is an analysis of identical markers).

☒ All plots are contour plots with outliers or pseudocolor plots.

☒ A numerical value for number of cells or percentage (with statistics) is provided.

## Methodology

| Sample preparation | Primary cells grown as epithelioids were trypsinized to obtain a cell suspension. |
|---|---|
| Instrument | Becton Dickinson (BD) LSRFortessa |
| Software | FACSDiva™ Software (BD-Biosciences)<br>FlowJo software (version 10.5.3) |
| Cell population abundance | 20000 single cells were analysed per sample. |
| Gating strategy | Single cells were selected using FSC-A/FSC-H and the cells expressing the fluorescent reporter quantified. YFP fluorescence was collected using the 488 nm laser and the 530/30 bandpass filter. ITGA6-647 fluorescence, to discriminate between basal and suprabasal cells, was collected using the 640 nm laser and the 670/14 bandpass filter. |

☒ Tick this box to confirm that a figure exemplifying the gating strategy is provided in the Supplementary Information.

