## [Peer Review File · Nature Genetics]

Peer Review Information

Manuscript Title: Organismal metabolism regulates the expansion of oncogenic PIK3CA mutant clones in normal esophagus

Corresponding author name(s): Professor Philip (H) Jones

Reviewer Comments & Decisions:

Decision Letter, initial version:

23rd Mar 2023

Dear Professor Jones,

First, please accept my apologies for the delay in returning this decision to you. Please also apologise to your team on my behalf, I appreciate that the wait is excruciating.

Your Article, "Organismal metabolism regulates the expansion of oncogenic PIK3CA mutant clones in normal esophagus" has now been seen by 3 referees. You will see from their comments below that while they find your work of interest, some important points are raised. We are interested in the possibility of publishing your study in Nature Genetics, but would like to consider your response to these concerns in the form of a revised manuscript before we make a final decision on publication.

We therefore invite you to revise your manuscript taking into account all reviewer and editor comments. Please highlight all changes in the manuscript text file. At this stage we will need you to upload a copy of the manuscript in MS Word .docx or similar editable format.

*2) If you have not done so already please begin to revise your manuscript so that it conforms to our

Article format instructions, available here.

*3) Include a revised version of any required Reporting Summary:

Please be aware of our guidelines on digital image standards.

[redacted]

We can be flexible about resubmission deadlines (with the usual disclaimers regarding novelty) but please let me know if you anticipate your revisions taking more than 6 months.

Sincerely,

Safia Danovi
Editor
Nature Genetics

Referee expertise:

Referee #1: mouse models, obesity, cancer

Referee #2: mouse models, metabolism

Referee #3: cell competition

Reviewers' Comments:

Reviewer #1:

Remarks to the Author:

In the present manuscript, Herms and Bartomeu et al. report that a heterozygous PIK3CAH1047R mutation in the normal esophageal epithelium of mice leads to clonal expansion of progenitor cells due to increased PI3K/mTOR signaling and glycolysis. Leveraging glycolytic activators/inhibitors, murine models of type-1 diabetes, and high fat diet (HFD) induced obesity, the authors show that the metabolic environment can affect the proliferation of cells harboring oncogenic mutations in normal tissues, and – more importantly – interventions to balance signaling could limit their expansion.

The findings reported here are timely and of great importance as many studies – including from this group – have shown an accumulation of cells harboring oncogenic mutations across a plethora of seemingly healthy tissue types, yet the long-term consequences and underlining mechanisms responsible are not fully understood. There are, however, a few major concerns that should be addressed to strengthen the manuscript, most notably a lack of convincing evidence to support the claim that elevated aerobic glycolysis is responsible for the acquired cellular fitness in PIK3CAH1047R clones and a lack of biochemical studies to support the differences in insulin and PI3K signaling inferred from the experiments.

Major concerns:

1. Fig. 2 and Extended Data Fig. 1 – A strength of the manuscript is the generation of a new Cre-inducible knock-in oncogenic PI3K allele within the endogenous, which overcome some of the concerns related to expression level of prior models (e.g., ROSA26 knock-in), as evident with the signaling studies in 3T3 cells and primary keratinocytes from the model. Confirmation of PI3K/AKT signaling in vivo using immunofluorescence analysis (or western blots on directly sorted esophageal keratinocytes) for downstream pathway activation would strengthen the manuscript. More specific details on the percent recombination (described as “low frequency” in methods) with the indicated dosing strategies for B-naphthoflavone/tamoxifen (e.g., single vs. repeat) should be reported.

2. Fig. 4 and Extended Data Fig. 4 – The authors use transcriptional readouts for PI3K/AKT/mTOR when biochemical readouts of signaling should be pursued to corroborate these findings in PIK3CAH1047R vs. WT clones both in vitro and in vivo, especially in confirming the effects of insulin and relevant inhibitors (e.g., LY294002). This could be done by assessing phosphorylation of INSR, IRS-2, AKT, and relevant downstream targets (mTOR, S6K, S6) by western blotting and/or Immunofluorescence in cell lines and in esophageal tissue in mice. How dynamic are the competitive fitness phenotypes observed? From a clinical relevance standpoint, one would like to know whether sustained inhibition of PI3K signaling in mutant cells or augmentation of PI3K/AKT signaling in normal cells is required to suppress mutant clone expansion (neither of which would be ideal for human translation), or whether transient modulation is sufficient. The authors could test the reversibility of insulin enhanced cellular fitness by performing insulin withdrawal studies (post-activation), using their

3D co-culture system and assess whether the competitive advantage of PIK3CAH1047R is restored.

3. Fig. 5 – Many of the genes enriched/depleted in CRISPR screen are not exclusively regulators of glycolysis (i.e, mTOR, Myc, SREBP1/2, etc.). Myc for example has been shown to regulate amino acid metabolism, most notably glutamine metabolism (Gao, P et al. Nature 2009, Bott AJ et al. Cell Metabolism, 2015), proline metabolism (Liu, W et al. PNAS 2012) and lipid metabolism (Eberlin, LS et al PNAS 2014, Edmunds, LR et al. J. Biol. Chem 2014). SREBP1 is a well-characterized master transcription factor of lipogenesis, responsive to serum fatty acid, cholesterol, and insulin levels (Debose-Boyd R et al. Trends in Biochemical Sciences 2018). Lastly, GLUT1, a glucose transporter and promoter of glycolysis, was enriched in the screen providing evidence against glycolysis-induced cellular fitness. Therefore, it is not clear that glycolysis is the prime consequential metabolic pathway responsible for PIK3CAH1047R-enhanced cellular fitness. To address this, the authors should treat their co-cultures with inhibitors for Glutaminolysis (GLS inhibitor/CB-839) and/or lipogenesis (TVB-2640 (FASN inhibitor), botulin (SREBP1/2 inhibitor)) and assess their role on enhanced cellular fitness. The authors should also confirm that there is indeed increased glucose uptake in PIK3CAH1047R clones to support aerobic glycolysis and determine whether co-cultures grown in low vs. high glucose levels modulates the difference in competitive advantage between the mutant and wildtype clones. The authors should also provide a rationale for why losing GLUT1 was not detrimental for mutant PI3K enhanced cell fitness. Lastly, while not necessary for the current findings to be interesting, mapping the metabolic fate of glucose via heavy isotope tracing (Jang C et al. Cell, 2018) in mutant vs WT clones could provide insights into which pathways (nucleotide biosynthesis, lipid synthesis, hexosamine synthesis. etc.) are functionally important for improved cellular fitness via PI3K activation in keratinocytes and esophageal tissue.

4. Fig. 6 – The authors treat keratinocytes with DCA (glycolysis inhibitor) however the effect is very modest as by the end of treatment, most of the cultures (~70%) are still PIK3CAH1047R despite a significant decrease in lactate secretion again, arguing against glycolysis being solely responsible for increased cellular fitness in PIK3CAH1047R. How do the authors explain this? Additional, more potent inhibitors of glycolysis such as 2-DG (Zhao J et al. Cell Death Discovery, 2019) should be tested – titrated for optimal non-lethal dose – to effectively inhibit glycolysis. For in vivo studies, cell-based pharmacodynamic markers are needed to corroborate whether the drug effects are direct (as hypothesized) or indirect. For example, the authors should show PI3K/AKT/mTOR activation by immunofluorescence in mutant and wild-type clones in vivo upon MET/DCA treatment to confirm expected differences in PI3K signaling between PIK3CAH1047R and WT clones is maintained, and the competitive fitness of mutant cells is blocked downstream at the level of glycolysis. As these agents may also modulate islet function and insulin sensitivity, immunofluorescence for pINSR on esophageal epithelial cells and serum insulin levels should be measured in these in vivo experiments. These results may help differentiate whether the effects of metformin and DCA are due to altering insulin production/sensitivity or directly acting on keratinocytes to mediate their effects. Lastly, the addition of MET appears to increase basal cell number in WT clones but decrease in mutant clones (Figs. 6i-j). Is this due to an off-target effect of metformin on mutant clones or saturated levels of glycolysis?

5. Fig. 7 – As described, Akitahet mice have reduced system insulin levels and hyperglycemia both of which could contribute to a competitive fitness imbalance between PIK3CAH1047R and WT clones but affecting distinct cells. Loss of insulin would be expected to reduce the relative fitness of WT cells, whereas high glucose might be predicted to increase the fitness of mutant cells that may be taking up more glucose for aerobic glycolysis. Evaluation of INSR activation (and downstream PI3K/AKT) in esophageal tissue of PIK3CAH1047R/wt; Akitahet comparing mutant versus wild-type clones would

help distinguish these possibilities. Furthermore, the authors could parse the effects on basal expansion of increased glucose availability vs. hypoinsulinemia on PIK3CAH1047R vs. WT clones by lowering glucose in a pancreas-independent manner (e.g., with an SGLT2 inhibitor; Nasiri AR et al. Cancer Metab, 2019) or by restoring sustained insulin levels in Akitahet mice using insulin pellets, as described previously (Wang Y et al, Cell reports, 2018).

6. Fig. 7 - It is not clear that the increase in PIK3CAH1047R clones in the esophagus of mice fed a high fat diet (HFD) is due to hyperinsulinemia, especially considering that hyperinsulinemia due to a HFD may result in a loss of insulin sensitivity in insulin-sensitive tissues, perhaps even in the esophageal keratinocytes. Immunofluorescence for changes in INSR activation and downstream signaling (as described above) due to HFD can validate that the increase in PIK3CAH1047R clones is due to increased activation in mutant clones or loss of insulin sensitivity in WT clones.

7. Fig. 7- Given the importance of insulin levels in altering mutant clone expansion in mice, the authors should show the diabetes status for human data showing an accumulation of PI3K mutant clones in esophageal tissue of obese vs non-obese patients and whether it alters the findings.

8. Extended Data Fig. 8. There are well-known sex-specific differences in weight gain, insulin resistance, and islet resilience comparing male and female mice fed a lard-based HFD. The authors state that there were no "gender" specific differences in their experiments in the Methods, but that would be unlikely based on short-term HFD treatment and the data presented. This should be clarified.

Minor concerns:

1. Axes labeling (throughout all figures) are too small. Larger labeling needed.
2. Typo on page 14; "... we induced Cre-Pik3caH1047R-YFP/wt mice at a level than used above and aged them for 3 months..."

Reviewer #2:

Remarks to the Author:

Title: Organismal metabolism regulates the expansion of oncogenic PIK3CA mutant clones in normal esophagus

This is an interesting study that looks to investigate the acute consequences of PI3KCA mutation in oesophageal epithelium. The authors show convincingly that PI3KCA mutant clones have a competitive advantage in the epithelium driven through metabolic rewiring. The role of PI3KCA in driving cancer is not so clear (obviously there are mutations) however the impact of PI3KCA in the cancer in marginal (although statistically significant). The authors importantly don't overplay this finding. Overall I feel this paper will be of interest to the community if a number of important issues are addressed.

Specific comments:

- Induction regime: There is no comment on the difference between X1 induction and X3 induction – how does it affect clone numbers per se and not just in terms of tumour numbers?

- Mouse numbers in cancer experiments

Page 6 – Fig 2c – The number of mice used is very high. How was power originally decided for these

experiments? I doubt the authors started with the assumption PI3KCA would only provide a very small amount of difference. If the authors go back in now with a power calculation now they know the difference, is the experiment appropriately powered?

- In Vitro Competition experiments

Pag 15-17 – In vitro competition experiments – My understanding is that Pik3caMut cells expressed EYFP, whilst wildtype cells expressed RYFP, whereby (as shown in Fig 4.a/b) Pik3caMut can be distinguished as positive green cells, whilst wildtype cells can be seen as yellow cells, which makes the set up perfect for cell competition studies and see how the cells outcompete each other. However, all subsequent images shown, particularly those of the cell competition experiment, are only with the RYFP marker, meaning that all it can be seen is a change in the wildtype cells, and not necessarily how the Pik3caMut cells are behaving. What was the reason for this choice? Co-staining of both cell types should have been taken.

- Pag 24 – extended data fig 6.h – as per the in vitro cell competition experiment in pag 15-17, why wasn't a co-stain for both PIK3camut and wildtype cells not performed to more directly determine the interaction and competition between the two cell types?

Functionally tested fitness genes

- Pag 22 – At the end of the paragraphs the authors report “Finally, depletion of most glycolysis pathway genes and some mutant-induced metabolism regulators, reduced cell fitness, including Ldha, which is not implicated in mitochondrial glucose oxidation (Fig. 5j and Extended Data Fig. 5f)”. From the data presented in Fig5 and till that point in the manuscript, the inhibition of specific targets can be seen, which can lead to the postulation that there is a reduction in fitness. However, a reduction in cell fitness is not directly shown or investigated, making this statement a bit of an over-reach (specifically at that point in the manuscript).

- Pag 23 – Same observation as above – The authors report “Targeting Atf4, Hif1 α /Hif1 β , Myc, Srebf1 or Srebf2 reduced cell fitness (Fig. 5j and Extended Data Fig. 5f), suggesting that PI3K regulates metabolic gene expression and cell fitness through multiple factors in parallel 45,46.” However, the data presented in Fig 5 and extended data Fig 5 shows the effect at the gene expression level of the CRISPR screen and not how the depletion of specific genes directly affects cell fitness. This will only be shown later in Fig 6 and thus this statement feels preemptive and a bit of an over-reach at that point in the manuscript.

Competition in overweight donors

Page 32 – The authors reported “Finally, the specific enrichment in overweight donors was caused by mutations present in the kinase domain (Extended Data Fig. 8k), including H1047R, in agreement with the increased fitness of Pik3caH1047R clones observed in mice under HFD (Fig. 7e and f).” – This statement is a bit misleading as the frequency of H1047R between overweight and non-overweight donors was similar, 9.7 and 9.3, respectively. Unless this is still statistically significant, but no Statistics were reported to make that conclusion.

Samples per patients

For the human-based analyses multiple samples per patient – 844 from 9 donors first and then 698 from 10 individuals – were taken. When determining the PIK3CA mutant clone density, including clone density of the specifically H1047R, was the analysis performed on equal number of samples per patient? If not, was a normalisation performed to take into account unequal numbers of samples per patient? How would this affect the results and conclusions? Wouldn't there be a risk of over/underestimating based on the mutational signature of a particular patient with more samples analysed compared to other patients?

Clonal dynamics in HFD

The authors showed early in the manuscript that supraphysiological doses of insulin, causing

overactivation of the PIK3 pathway, significantly reduced the competitive advantage of Pik3ca mut cells over the wildtype cells in mixed cultures. However, in their assessment of the mice on HFD, which was described as increasing mouse body weight and hyperinsulinemia, only small changes in wildtype clones and a significant increase in average mutant clones was seen. Was the expected phenotype of hyperinsulinemia assessed and seen in these mice? If yes, findings in these mice would deviate from the in vitro observations and should be commented on. If no, how can conclusions be made?

Minor comments

- Page 3 – Fig 1a – Difficult to interpret, particularly in terms of mutant density (unless I physically counting).
- Page 5 – Fig 1b – Quite a bit of green background-staining. This is seen in page 15 fig b too.
- Page 6 – Fig 2c – The number of macroscopic oesophageal tumours are being compared between uninduced vs x1 induction and uninduced vs x3, however this is not clear from the figure or figure legend and should be made clearer.
- Page 8 – Extended fig 2d: the figure is the same as fig 2a with no obvious reason to report the figure twice.
- Page 15 – Fig d: I found unclear how the comparison (Y axis) and figure generated.
- Page 17 – Fig 4g: Venn diagram not representative of an 82% vs 18% overlap.
-
- Figures arrangement with sections:
 - o Fig 5 – figures i, j,: should be included in fig 6 as they are not discussed or pertinent to this section.
 - o Extended data fig 5 – figures d, e, f : should be included in extended fig 5 as they are not discussed or pertinent to this section.
 - o There's an extended fig 8, but not a figure 8.
 - o Fig 7 g, h: should go under a separate fig 8 as they are not discussed or pertinent to this section and actually are relevant to extended fig 8 than 7.
 - o Similar observations for fig 3 and extended data fig 3 too.
- Typos:
 - o Pag 14 line 3 – word missing between a level and than.
 - o Pag 31 line 16 – Typo – midel instead of middle.

Reviewer #3:

Remarks to the Author:

The manuscript from Herms and Colum et al represents a thorough and important contribution to the body of work seeking to understand how homeostatic tissues cope with the frequent emergence of pre-malignant clones. Taking a cue from previously published analyses of patient samples, and using a suite of similar approaches to their previous work, the authors develop a mouse model to study activating mutations in PI3KCA in the esophagus. They find that cells heterozygous for PI3KCA mutations have a survival advantage over wild-type cells. Using mathematical modelling and lineage tracing experiments, they attribute this advantage to a skewing of fate dynamics towards progenitor self-renewal. Experiments largely conducted in a culture system allow them to use inhibitor treatments, RNA-sequencing studies, and CRISPR-screening approaches to determine that PI3KCA cells, due to upregulated mTOR/HIF signaling, ultimately shift to more glycolytic metabolism; this metabolic shift confers the growth advantage. All major results are then satisfyingly re-confirmed in vivo in the mouse model. Finally, in perhaps the most intriguing and novel part of the paper, they uncover links between metabolic diseases such as Type 1 diabetes and obesity, and heightened

expansion of PI3CA clones.

The mechanistic link between clonal competition, metabolism, and systemic disease is quite exciting and ultimately is what makes the paper worthy of publication in Nature Genetics.

I have only a few minor points:

1. The paper is very well written. However, for readers not familiar with the prior work of the Jones lab, it may be worth mentioning in the introduction that Notch1 mutations have been analysed in other studies – it just jumps out so prominently as a candidate! But of course there is excellent rationale for also studying PI3KCA.
2. Although the data is very clean and clearly presented, I am a bit confused about the set-up of the culture system. Since the mouse model made in this study is PI3KCA-T2A-YFP, are not all the cells in the co-culture experiments with the RYFP-WT cells YFP+? I understand that because of the adenovirus transduction method the RYFP cells may be relatively brighter, but in the image the PI3KCA mutant cells look YFP-negative (and maybe they are coming from a different mouse model that doesn't have YFP?). This should be somehow explained more clearly in the text (and I do wonder why in the first place R26-RFP wasn't used for WT cells! Would have been easier!).
3. A major outstanding question is why does a switch to glycolytic metabolism confer a fitness advantage? Addressing this is outside the scope of the current manuscript, but more discussion, and a better synthesis of the literature on cell competition and glycolysis, is warranted. The fly papers cited should include the 2014 paper from Laura Johnson's lab (de la Cova et al, Cell Metabolism) and also the paper from Pascal Meier's group (Banreti & Meier, Nature Communications, 2019).
4. I was initially troubled by the lack of earlier reference to the paper from the Beronja group on PI3KCA mutations in the skin (Ying et al, NCB, 2017), especially since in that case the result is almost opposite – that mutant clones shift their fate dynamics towards differentiation. When the paper is finally mentioned in the discussion, the authors here suggest that the different results perhaps can be explained by copy number (homozygous mutants in the case of the Ying paper). I appreciate that it would not be trivial for the authors to test this hypothesis, but a better attempt to reconcile these divergent results is warranted, at least by a more fleshed out discussion, if not by the addition of more data.

Author Rebuttal to Initial comments

Response to Reviewers

In the text below, our responses are in blue, changes to the text are in purple.

We are most grateful to the Reviewers for their insightful comments which have significantly strengthened our revised manuscript. In this study we have used a genetic approach in human, mouse and primary organotypic cell cultures with the strengths and limitations that each of these methods bring. We used a knock in strategy in mice to model the human *PIK3CA* mutants that colonize the normal esophagus. The knock in to the endogenous *Pik3ca* locus results in more subtle phenotypes compared with the more widely studied overexpression strains. We applied a new primary 3D cell culture system, epithelioids (1), to perform targeted CRISPR gene knockout studies of genes implicated as candidate regulators of cell fitness in expression studies. We have now added additional experiments suggested by

the Reviewers exploring the mutant phenotype. These include multiple *in vitro* competition experiments, western blotting analysis of primary cultures to show the effects of the *Pik3ca* mutation on downstream signalling +/- insulin/inhibitor treatment, and three additional CRISPR screens to explore cell competition in mutant and wild-type cells in the presence or absence of insulin.

The new results led us to substantially revise manuscript. Mutant cells clearly upregulate glycolysis-related HIF1 α target genes leading to a more glycolytic metabolism, and the deletion of most glycolysis genes negatively affects cell fitness. However, the additional data argues that the link between glycolysis and cell fate is complex and multiple other transcription factors and pathways downstream PI3K, such as lipogenesis, contribute to the increased fitness of mutant cells. We have revised our discussion of these results accordingly.

REVIEWER #1 (obesity and cancer, mouse models)

In the present manuscript, Herms and Bartomeu et al. report that a heterozygous PIK3CAH1047R mutation in the normal esophageal epithelium of mice leads to clonal expansion of progenitor cells due to increased PI3K/mTOR signaling and glycolysis. Leveraging glycolytic activators/inhibitors, murine models of type-1 diabetes, and high fat diet (HFD) induced obesity, the authors show that the metabolic environment can affect the proliferation of cells harboring oncogenic mutations in normal tissues, and – more importantly – interventions to balance signaling could limit their expansion.

The findings reported here are timely and of great importance as many studies – including from this group – have shown an accumulation of cells harboring oncogenic mutations across a plethora of seemingly healthy tissue types, yet the long-term consequences and underlining mechanisms responsible are not fully understood. There are, however, a few major concerns that should be addressed to strengthen the manuscript, most notably a lack of convincing evidence to support the claim that elevated aerobic glycolysis is responsible for the acquired cellular fitness in PIK3CAH1047R clones and a lack of biochemical studies to support the differences in insulin and PI3K signaling inferred from the experiments.

Major concerns:

1. Fig. 2 and Extended Data Fig. 1 – A strength of the manuscript is the generation of a new Cre-inducible knock-in oncogenic PI3K allele within the endogenous, which overcome some of the concerns related to expression level of prior models (e.g., ROSA26 knock-in), as evident with the signaling studies in 3T3 cells and primary keratinocytes from the model. Confirmation of PI3K/AKT signaling *in vivo* using immunofluorescence analysis (or western blots on directly sorted esophageal keratinocytes) for downstream pathway activation would strengthen the manuscript.

We agree with the Reviewer that demonstrating signalling differences between mutant and wild-type cells *in vivo*, and under the different conditions used in the study, would strengthen the manuscript. Unfortunately, no reliable methods for tissue-based analysis of PI3K pathway activation have been developed, despite extensive efforts from academic and pharma laboratories, especially in the cancer

field. Addressing this question is even more challenging in a clonal experiment where groups of mutant cells are widely scattered across the tissue. To detect these clones, we normally prepare epithelial wholemounts, a procedure that requires incubation of the tissue for 3 h in EDTA at 37°C before fixation; which disrupts phospho-protein staining.

With the clonal induction protocols as they were described in our manuscript, conventional tissue sections contained few if any mutant cells, so obtaining a sufficient number of cells for statistical analysis proved infeasible.

In an attempt to address this, we treated *Ahcre^{ERT}Pik3ca^{H1047R-EYFP}* mice with a high dose of recombination-inducing drugs (two 1-mg tamoxifen injections) and then waited 6 months to allow mutant clones to expand before harvesting tissue and staining cryosections for P-AKT (S473) and GFP, the latter staining the EYFP reporter expressed by mutant cells (**Reviewer Figure 1a**). GFP-positive cells were detected but GFP staining in sections gave rise to much higher cytoplasmic background than in tissue wholemounts. We used optical sectioning with confocal microscopy, with an optical section thickness of approximately 1.2 µm, to image the cell nuclei without overlying cytoplasm. This allowed us to resolve the low level signal of nuclear localised YFP detected by the GFP antibody, from the cytoplasmic background present in all cells. Given that the expression of YFP-NLS is directly linked to the *Pik3ca^{H1047R}* allele, this enabled us to score clusters of cells with positive *nuclear* GFP as *Pik3ca* mutant clones (**Reviewer Figure 1b**). Using this scoring rule to assign cells as mutant or wild-type, we next proceeded with P-AKT staining. Positive P-AKT signals were mostly observed in the basal layer and first suprabasal tissue layers (**Reviewer Figure 1c**). We also observed individual cells or groups of cells with a very bright P-AKT staining (**Reviewer Figure 1c**), consistent with the cell-cell heterogeneity in PI3K activation observed in cell lines in *in vitro* conditions (2), but unrelated to the GFP labelling. Unfortunately, and despite our best efforts, the interpretation of these staining patterns is unclear, and we did not find sufficient mutant cells in the sections containing mutant clones for reliable statistical analysis.

In addition to our observations above, there are additional considerations with regards to assessing the activity of the mutant *in vivo* by immunostaining, given the low-level activation of *Pik3ca* from the endogenous *Pik3ca* locus in the oesophagus epithelium. Firstly, we note that to detect signalling differences between wild-type and mutant cells by western blotting *in vitro*, cells had to be starved of growth factors (**Extended Data Figure 1c**). This suggests that signalling differences in PI3K pathway resulting from mutant expression in esophagus *in vivo* are likely to be modest compared to those observed in *Pik3ca* mutant overexpression studies (3-5). Secondly, the PI3K/AKT pathway activation response of single cells in culture to insulin or IGF1 is intrinsically heterogeneous (6). Such cell-to-cell variability may be even higher *in vivo* where access to growth factors is also dictated by distance to blood vessels and diffusion capacity of each growth factor. The analysis of large numbers of mutant and wild-type cells per esophagus would be required to overcome such signalling noise. Thirdly, little is known about the signalling patterns of those pathways in the mouse esophagus. It is likely that signalling staining varies across the day/night cycle in nocturnal feeders such as mice, and the time after the feeding-induced insulin peak might be relevant in transient signalling pathways like PI3K/AKT. Unfortunately, we do not have sufficient resource to generate a large enough cohort of highly induced mice to sample the multiple

time points to generate enough data to resolve how staining of P-AKT and other proteins varies between wild-type and mutant cells across a 24-hour period.

Taken together, we believe that the combination of genetic evidence *in vivo* in mouse and human, the published literature for the *Pik3ca*^{H1047R} mutant (<https://aacrjournals.org/cancerres/article/74/3/641/599228/The-Structural-Basis-of-PI3K-Cancer-Mutations-From>) and phospho-protein studies *in vitro* argues that the *Pik3ca*^{H1047R} mutant drives clonal expansion in esophagus via activation of PI3K signalling (7, 8).

Reviewer Figure 1. a, Scheme of the protocol. *Pik3ca* fl-H1047R-T2A-YFP-NLS mice were induced twice with 1 mg tamoxifen and 80 mg/kg β -naphthoflavone, culled 6 months later and esophagi collected. Cryosections were stained and imaged by confocal microscopy. **b** and **c**, Representative optical confocal sections of approximately 1.2 μ m. Immunofluorescence signal of antibodies to GFP (b, green) or P-AKT(S473) (c, red) and DAPI (blue) and wheat germ agglutinin (grey) staining of representative sections of mouse esophagus. Scale bars: 20 μ m. White and red circles in **b** indicate uninduced and induced mutant cells respectively. Yellow selections in **c** indicate areas occupied by induced basal cells.

More specific details on the percent recombination (described as “low frequency” in methods) with the indicated dosing strategies for β -naphthoflavone/tamoxifen (e.g., single vs. repeat) should be reported.

Following this Reviewer’s suggestion, we quantified the percentage of recombined epithelium obtained with the tamoxifen doses used for lineage tracing and for mutation induction in the tumorigenesis experiments. We now include this information in the methods Lineage tracing and Chemically induced mutagenesis sections:

“Lineage tracing

To induce low frequency expression of EYFP in the mouse esophagus, 10–16 week-old transgenic mice were given a single intraperitoneal (i.p.) of 80 mg/kg β -naphthoflavone (MP Biomedicals 156738) and 1 mg tamoxifen (Sigma Aldrich N3633) (*Ahcre^{ERT}Rosa26^{flConfetti}-Pik3ca^{fl-H1047R-T2A-EYFP-NLS/wt}*) or 0.25 mg tamoxifen (other mouse strains), resulting in **2.9 \pm 0.9% and 0.17 \pm 0.04% (mean \pm s.d.) recombined epithelium by area respectively 10 days after induction**. Following induction, between three and eight mice per time point were culled and the esophagus collected. Time points analyzed included 10 days, 1, 3 and 6 months after induction. As expression from the endogenous *Pik3ca* locus is very low, immunostaining was necessary to detect EYFP-NLS reporter expression (**Extended Data Figure 2b**). The total number of clones quantified for each figure is shown in Supplementary Table 1. Normalized clone-size distributions were built for each experimental condition and time point from the observed relative frequencies $f_{m,n}$ of clones of a certain size, containing m basal and n suprabasal cells, resulting in two-dimensional histograms, (displayed as heatmaps using CloneSizeFreq_2Dheat package (https://github.com/gp10/CloneSizeFreq_2Dheat)). A 2D histogram of the residuals or differences observed between conditions in the relative frequencies of each particular clone size (i.e., each cell on the grid) was generated when appropriate.

Chemically induced mutagenesis

To generate mutations in the esophageal epithelium, mice were treated with diethylnitrosamine (DEN) (Sigma, catalog no. N0756) at 40 mg per 1,000 ml sweetened drinking water for 24 h on 3 days a week (Monday, Wednesday and Friday) for 8 weeks. When indicated, mice were induced using one or three intraperitoneal (i.p.) injections of 80 mg/kg β -naphthoflavone (MP Biomedicals 156738) and 1 mg tamoxifen (Sigma Aldrich N3633), **resulting in 2.9 \pm 0.9% and 10.2 \pm 2.8% (mean \pm s.d.) recombined epithelium by area respectively of recombination respectively 10 days after the last induction**. After

each dosage mice received sweetened water until the next DEN treatment. Control mice received sweetened water as vehicle for the length of the treatment. After the 8 weeks, all mice were administered normal water.”

2. Fig. 4 and Extended Data Fig. 4 – The authors use transcriptional readouts for PI3K/AKT/mTOR when biochemical readouts of signaling should be pursued to corroborate these findings in PIK3CAH1047R vs. WT clones both *in vitro* and *in vivo*, especially in confirming the effects of insulin and relevant inhibitors (e.g., LY294002). This could be done by assessing phosphorylation of INSR, IRS-2, AKT, and relevant downstream targets (mTOR, S6K, S6) by western blotting and/or Immunofluorescence in cell lines and in esophageal tissue in mice.

Please refer to the comment above for the challenges in assessment of phosphorylation in clonally induced *in vivo* samples using immunostaining. To biochemically assess PI3K pathway activation by primary cells in 3D culture, we performed Western blot analysis of pAKT (S473 and T308), pGSK3 and pS6, in starvation medium (0.1% FCS minimal medium without added growth factors). We also analysed the effect of adding a supraphysiological dose of insulin or the PI3K inhibitor LY294002. The results confirm that mutant cells have increased PI3K signalling. The PI3K inhibitor reduces PI3K signalling in both wild-type and mutant cells, while insulin increases PI3K signalling in both wild-type and mutant cells (**Reviewer Figure 2**). Following the Reviewer’s suggestion these results have been added to **Figure 4**, as panel **e** and the transcriptional readouts for PI3K activation have been moved to **Extended Data Figure 4**.

Reviewer Figure 2. Uninduced or Induced cells were cultured overnight in starvation medium (STV, mFAD with 0.1% FCS without insulin). Next, cells were cultured for 1 h either in STV, or STV plus LY294002 (50 μM), or STV with Insulin (5 μg/ml). Samples were lysed and analysed by western blot using antibodies against P-AKT(S473), P-AKT(T308), AKT, P-GSK3β, GSK3β, P-S6, S6 and α-tubulin. The blots shown are representative of 3 biological replicates.

We describe these results in the revised text as follows: (Page 19)

Over-activation of PI3K signaling, using supraphysiological doses of insulin, abrogated the differences in differentiation and gene expression between wild-type and mutant cells (**Fig. 4d-f and Extended Data Fig. 4d**)(9).

Conversely, leveling down the signaling activity of the PI3K pathway in both wild-type and mutant cells by treating mixed cultures with the PI3K inhibitor LY294002 or the mTOR inhibitor Rapamycin also reduced the mutant cell advantage (**Fig. 4e, j and Extended Data Fig. 4g**).

How dynamic are the competitive fitness phenotypes observed? From a clinical relevance standpoint, one would like to know whether sustained inhibition of PI3K signaling in mutant cells or augmentation of PI3K/AKT signaling in normal cells is required to suppress mutant clone expansion (neither of which would be ideal for human translation), or whether transient modulation is sufficient. The authors could test the reversibility of insulin enhanced cellular fitness by performing insulin withdrawal studies (post-activation), using their 3D co-culture system and assess whether the competitive advantage of PIK3CAH1047R is restored.

Following the Reviewer's comment, we performed the suggested experiment to check the reversibility of inhibition of mutant cell advantage by insulin in mixed mutant/wild-type primary 3D cell cultures. We compared mutant and wild-type cell competition in co-culture experiments for 28 days in control (CTL) and supraphysiological insulin (INS) conditions with treatment with insulin for 14 days followed by culture without insulin for a further 14 days (INS/CTL condition). As expected, we observed a strong competitive advantage of mutant over wild-type in control condition which was significantly reduced in the presence of insulin. In the INS/CTL condition the proportion of wild-type cells was significantly lower than in cultures treated with insulin for 28 days and slightly higher than in CTL cultures (**Reviewer Figure 3**). We conclude that when insulin is removed from the medium, the mutant cell advantage is restored. We have added these results to **Extended Data Fig 4** and comment on them in the results section as follows:

"Of note, the inhibitory effect of high dose insulin on mutant cell advantage was reversible, given that mutant cells were able to outcompete wild-type cells again upon removal of insulin after 15d of treatment (**Extended Data Fig. 4f**)."

Reviewer Figure 3. Experimental scheme (top panel) and quantification by flow cytometry (bottom panel) of the proportion of *WT-RYFP* cells mixed with *Pik3ca^{H1047R/wt}* cells at the end of the experiment versus the start of each experiment. Cells were cultured for 28 days in minimal medium (CTL), 28 days with minimal medium with 5 $\mu\text{g/ml}$ insulin (INS), or 14 days with minimal medium with 5 $\mu\text{g/ml}$ insulin followed by 14 days with minimal medium. Red lines indicate mean values of biological replicates in which cells each genotype were generated from a different animal. Two-tailed paired *t*-test. Source data are shown in Supplementary Table 1.

3. Fig. 5 – Many of the genes enriched/depleted in CRISPR screen are not exclusively regulators of glycolysis (i.e., mTOR, Myc, SREBP1/2, etc.). Myc for example has been shown to regulate amino acid metabolism, most notably glutamine metabolism (Gao, P et al. Nature 2009, Bott AJ et al. Cell Metabolism, 2015), proline metabolism (Liu, W et al. PNAS 2012) and lipid metabolism (Eberlin, LS et al PNAS 2014, Edmunds, LR et al. J. Biol. Chem 2014). SREBP1 is a well-characterized master transcription factor of lipogenesis, responsive to serum fatty acid, cholesterol, and insulin levels (Debose-Boyd R et al. Trends in Biochemical Sciences 2018). Lastly, GLUT1, a glucose transporter and promoter of glycolysis, was enriched in the screen providing evidence against glycolysis-induced cellular fitness. Therefore, it is not clear that glycolysis is the prime consequential metabolic pathway responsible for PIK3CAH1047R-enhanced cellular fitness. To address this, the authors should treat their co-cultures with inhibitors for Glutaminolysis (GLS inhibitor/CB-839) and/or lipogenesis (TVB-2640 (FASN inhibitor), botulin (SREBP1/2 inhibitor)) and assess

their role on enhanced cellular fitness. The authors should also confirm that there is indeed increased glucose uptake in PIK3CAH1047R clones to support aerobic glycolysis and determine whether co-cultures grown in low vs. high glucose levels modulates the difference in competitive advantage between the mutant and wildtype clones. The authors should also provide a rationale for why losing GLUT1 was not detrimental for mutant PI3K enhanced cell fitness. Lastly, while not necessary for the current findings to be interesting, mapping the metabolic fate of glucose via heavy isotope tracing (Jang C et al. Cell, 2018) in mutant vs WT clones could provide insights into which pathways (nucleotide biosynthesis, lipid synthesis, hexosamine synthesis. etc.) are functionally important for improved cellular fitness via PI3K activation in keratinocytes and esophageal tissue.

We are most grateful for these comments and the proposed experiments. We accept that, although our data suggested that glycolysis impacts cell fitness, we should have not assumed this was the *only* pathway determining mutant cell advantage. As the Reviewer correctly points out, most of the effectors in the CRISPR screen directly regulate multiple pathways. Therefore, we have rewritten the text to make clear that, although mutant cells show increased glycolysis and the knock-out of most glycolysis genes reduces cell fitness (we discuss *Glut1* below), increased glycolysis may only partially explain increased mutant fitness.

Firstly, as the Reviewer suggested, we have now confirmed that mutant cells show a trend towards increased glucose uptake (**Reviewer Figure 4**). We have not currently included this data in the revised version of the manuscript as we have now reduced the focus on glycolysis as the major cause of mutant cell fitness advantage. We would submit that combination of the RNAseq with the Seahorse experiment confirms mutant cells are more glycolytic than wild-type, as is already described for this *Pik3ca* mutation in other contexts (10).

Reviewer Figure 4. *Pik3ca*^{H1047R/wt} primary cells were infected with a null adenovirus (Adnull) or a *Cre* recombinase expressing adenovirus (AdCre) to induce expression of the mutant allele. Cells were incubated with ¹⁴C-Glucose for 3 h, lysed and radioactivity quantified in the protein lysate. Red lines indicate mean values of biological replicates, each from a different animal.

As the Reviewer highlights, deletion of the predominant glucose transporter *Glut1* promotes cell fitness. Recently, cytoplasmic glucose levels have been shown to increase during keratinocyte differentiation together with a metabolic switch to oxidative metabolism. Free glucose binds and regulates the function of transcription factors and RNA splicing factors that modulate the differentiation (11, 12). Deletion of *Glut1* may thus impair differentiation conferring a fitness advantage on *Glut1* mutant clones by reducing intracellular levels of glucose. Activation of glycolysis may also reduce cytosolic glucose levels achieving a similar effect to *Glut1* deletion. This may explain why targeting glycolysis in the CRISPR/Cas9 screen (via knockout of *Aldoa* or *Ldha*) reduces cell fitness while targeting glucose uptake (via *Glut1* deletion) increases fitness. Finally, cytosolic glucose is also used in multiple pathways including glycogen synthesis, the pentose phosphate pathway and glycosylation which has also key roles in regulating keratinocyte differentiation (13). One or more of these pathways may also play a role explaining the effects of gene deletions on fitness.

Regarding mutant cell fitness and DCA treatment, as mentioned by the Reviewer, reducing the glycolytic capacity by channelling pyruvate into mitochondria with DCA at least partially reduces mutant cell expansion in mixed mutant/wild-type cultures. However, removal of glucose from the medium (except the glucose present in foetal calf serum) or inhibition of glucose uptake/retention with 2-deoxyglucose had no effect on the fitness advantage of mutant over wild-type cells (**Reviewer Figure 5a**).

Taken together, the above results suggest that the effects of disrupting glycolysis on the relative fitness of wild-type and mutant cells may depend on the level at which the pathway is targeted.

To gain more detailed genetic insights into mutant cell fitness, we repeated the CRISPR/Cas9 screen in *Pik3ca* mutant cells (in this screen, all cells were mutant). The effect of *Glut1* deletion is the same in mutant as in wild-type cells (**Reviewer Figure 5b and c**). Interestingly, when *Ldha*, *Eno1*, *Pgam1* or *Aldoa* are deleted, they show a larger effect on mutant than on wild-type cell fitness. This is suggestive of glycolysis having a specific role in mutant cell fitness (**Reviewer Figure 5b and c**). We also repeated the screens in the presence of high dose insulin, finding reduced differences between mutant and wild-type cells. In the presence of insulin, deletion of *Ldha*, *Eno1*, *Pgam1* or *Aldoa* affected cell competition among wild-type cells in a similar degree than cell competition among *Pik3ca* mutant cells (**Reviewer Figure 5c-d**). We conclude that the activation of the PI3K/mTOR pathway with insulin generates a similar degree of dependency on glycolysis as the activation of the PI3K/mTOR pathway by the *Pik3ca*H1047R mutation.

Reviewer Figure 5. **a**, Flow cytometry quantification of the proportion of WT-RYFP cells in mixed cultures with *Pik3ca*^{H1047R/wt} cells, at 15 days normalized to baseline value. +/- Gluc indicates treatment in minimal medium with/without glucose and 2-DG indicates treatment with 5 mM 2-deoxyglucose. Each dot represents a primary culture from an animal and lines correspond to mean values. n=4 primary cultures from individual animals per condition. **b**, Illustration of the screen gene targets related to the glycolysis pathway in CRISPR screens performed in uninduced (WT, left panel) or induced (*Pik3ca*^{wt}, right panel) primary esophageal keratinocytes from the *Rosa26*^{Cas9/wt} *Pik3ca*^{H1047R/wt} mouse strain. Box color indicates the Log₂ (fold change) between the 3 weeks and 0 weeks timepoints as indicated by the color range bar. Significantly enriched or depleted genes (FDR<0.1 and >10% Fold change difference) are highlighted in

bold red. Yellow boxes indicate the enzyme isoform most expressed in esophageal primary cells. Source data are shown in **Supplementary Table 1. c-d**, Plot showing the average Log₂ (fold change) of the gRNA targeting the indicated genes in the two CRISPR screen conditions specified. Panels indicate the gene-sets corresponding to Glycolysis pathway genes. c, CRISPR screens performed in *Pik3ca*^{H1047R/wt} cells (y-axis) versus WT cells (x-axis) in control (CTL, top) or insulin-treated condition (INS, bottom). d, CRISPR screens performed in WT cells in CTL (x-axis) versus INS treated (y-axis). Yellow and green areas of each graph indicate the values with higher absolute Log₂ (fold change) in wild-type or *Pik3ca*^{H1047R/wt} cells respectively. Linear regression is shown in black with its slope and R² values indicated in each graph. Identity line is shown in orange. Error bars indicate the standard deviation between multiple replicates of the CRISPR screening. n=2-3 independent screens. Source data are shown in **Supplementary Table 1**.

As suggested by the Reviewer other pathways downstream PI3K may also be important for mutant cell competition given that PI3K pathway regulates multiple signalling and metabolic pathways (14)), and this specific *Pik3ca* mutant is reported to regulate lipogenesis and glutaminolysis (15, 16). We performed the additional mutant/wild-type cell competition experiments proposed by the Reviewer. We found that that glutaminolysis inhibition with CB839 did not alter mutant fitness. However, inhibition of lipogenesis using the FASN inhibitor TVB-2640 reduced the competitive advantage of mutant cells (**Reviewer Figure 6**). We then inhibited lipogenesis at a different stage using TOFA, an ACC inhibitor. TOFA treatment also significantly reduced mutant cell competitiveness, confirming a role for lipogenesis in mutant cell competition (**Reviewer Figure 6**). Betulin, a SREBP inhibitor had no significant effect on mutant cell fitness. This argues that the effect of the *Pik3ca* mutant esophageal is not mainly due to transcriptional changes (**Reviewer Figure 6**). Consistent with this, we observed no changes in gene expression in most lipogenesis related genes comparing mutant and wild-type cells cultured either in standard media (CTL) or in the presence of a high concentration of Insulin (INS) except for the expression of the mitochondrial citrate exporter (*Slc25a1*) and the *Scd2* genes which was slightly increased in mutant cells. Interestingly, these differences were abrogated by treating with insulin, suggesting that they could be due to the differential activation of the PI3K pathway in mutant and wild-type cells (**Reviewer Figure 6**).

Reviewer Figure 6. (Left panel) Flow cytometry quantification of the proportion of wildtype (R26-YFP) cells in mixed cultures with *Pik3ca*^{H1047R/wt} cells, at 15 days normalized to baseline value. Cells were treated

in minimal medium (CTL) or minimal medium plus Betulin (6 µg/ml), CB839 (10 µM), TOFA (30 µM) or TVB-2640 (0.1 µM) during the whole experiment. Each dot represents a primary culture from an animal and lines correspond to mean values. n=4 primary cultures from individual animals per condition. Two tailed paired t-test. Source data are shown in Supplementary Table 1. (Right panel) RNA-seq gene expression analysis showing differences between wild-type (WT, uninduced) and *Pik3ca*^{H1047R/wt} mutant (induced) primary esophageal keratinocytes, maintained in minimal FAD medium. Heatmaps comparing wild-type and *Pik3ca*^{H1047R/wt} primary cells in control conditions (CTL) or treated with 5 µg/ml insulin (+INS). Lipogenesis related genes are shown. n=4 mice per group, from paired induced-uninduced samples. Stars show significant differences in CTL samples. None of the genes were differentially expressed between wild-type and *Pik3ca*^{H1047R/wt} mutants in the +INS condition. **p<0.01, ***p<0.001, n.s.= not significant. Wald test corrected for multiple testing using the Benjamini and Hochberg method.

We conclude that the switch to glycolysis is a consequence of the PI3K pathway activation in mutant cells but is only one potential cause of the mutant cell competitive advantage, as mutant cells still show increased fitness when using other metabolic substrates. The new data obtained following the Reviewer's suggestions demonstrates that lipogenesis inhibition is also able to reduce the competitive differences between mutant and wild-type cells, arguing, lipogenesis is also important player in mutant cell advantage. In light of this, we no longer attribute the upregulation of glycolysis as the sole cause of the cell fate phenotype of the mutant in the revised version of the manuscript.

We have made the following changes to the manuscript:

The CRISPR data in **Reviewer Figure 5** are now included in **Figures 5** and **Extended Data Figure 5**. **Reviewer Figure 6** is incorporated into **Extended Data Figures 6** and **7** which are discussed in the text as follows:

'The same CRISPR screen performed in a *Pik3ca*^{H1047R/wt} mutant cells in minimal media yielded similar results (**Figure 5b**), with only small differences in the degree of enrichment/depletion for most gRNAs comparing mutant and wild-type cells. The log₂ (fold changes) of gRNAs targeting PI3K signaling after 3 weeks strongly correlated in mutant and wild-type cells, indicating the pathway is of similar importance for competitive fitness in both cell populations (**Figure 5c** and **Extended Data Figure 5c**). The log₂ (fold changes) of gRNAs targeting components of the mTOR pathway were less correlated, indicating that wild-type cells appear more dependent on the mTOR pathway for competitive fitness than mutant cells. Similar correlations were found in CRISPR/Cas9 screens performed in WT cells treated with supraphysiological doses of insulin (INS) that overactivate the PI3K pathway, confirming that the advantage of *Pik3ca*^{H1047R/wt} is due to a higher activation of the PI3K/mTOR pathway. In agreement, differences between wild-type and mutant cells were reduced when screens were performed under INS treatment (**Figure 5c** and **Extended Data Figure 5c**). These results are consistent with *Pik3ca*^{H1047R} mutation promoting cell fitness by mTOR dependent and mTOR independent pathways.'

'Next, we investigated whether the metabolic switch observed in *Pik3ca* mutant cells impacted their competitive fitness or was just a consequence of the increased HIF1α signaling. The connection between glycolysis and cell competition is controversial. Metabolic reprogramming towards glycolysis alters cell competitive fitness in *Drosophila* (17, 18). In mammalian keratinocytes, glycolysis has been linked to cell

fate regulation and differentiation suggesting that glycolytic activation may confer a proliferative advantage in the esophagus (19-21). gRNAs targeting most glycolysis pathway genes were depleted in the screen, including *Ldha*, which is not implicated in mitochondrial glucose oxidation, implying glycolysis regulates cell fitness (**Extended Data Fig. 5c-e**). Intriguingly, however, deletion of the glucose transporter *Glut1* showed the opposite effect. This argues that modulation of glycolysis at different levels may have different effects on competitive fitness. This may be explained by effects on differentiation. Cytosolic glucose has been shown to promote keratinocyte differentiation while activation of downstream glycolysis inhibits keratinocyte differentiation (12, 22). Consistent with a link between glycolysis and mutant cell fitness advantage, mutant cells are more sensitive to the loss of *Ldha*, *Aldoa*, *Eno1* and *Pgam1* than wild-type cells (**Extended Data Fig. 5c-e**) and similar effects were observed in INS treated WT cells (**Extended Data Fig. 5f**). ‘

‘Is glycolysis the only metabolic pathway responsible for the increased fitness of mutant cells? Surprisingly, removing glucose from culture medium or treating with the glucose uptake inhibitor 2-deoxyglucose did not affect *Pik3ca*^{H1047R/wt} cell fitness advantage in mixed mutant/wild-type cultures (**Extended Data Figure 7g**). Conversely, treatment with DCA, which by inhibiting pyruvate dehydrogenase kinase-1 favors mitochondrial glucose oxidation over aerobic glycolysis to lactate, modestly but significantly reduced mutant cell advantage (**Extended Data Figure 7h-j**) (23-25). In agreement with the *in vitro* results, DCA treatment *in vivo* for 28 days reduced mutant clone size and increased the proportion of differentiated mutant cells towards the levels observed in wild-type clones (**Extended Data Figure 7k-n**). A non-parametric two-factor analysis (see methods) revealed that DCA significantly decreased the differences between wild-type and mutant basal clone size distributions (**Extended Data Fig. 7o**). Because of the effect of DCA in multiple organs *in vivo*, it is important to note that its effect on *Pik3ca*^{H1047R/wt} cell fitness could be via the direct targeting of esophageal glycolysis or by reducing insulin blood levels or a combination of both (23-25). These results suggest that glycolysis activation only partially explains the competitive fitness advantage of *Pik3ca*^{H1047R/wt} over wild-type cells.

A metabolic switch to aerobic glycolysis is frequently accompanied by an increased glutaminolysis and lipogenesis, which are regulated by the PI3K/AKT/mTOR pathway (14, 26, 27). The *Pik3ca*^{H1047R} mutation is described to activate *de novo* lipogenesis and glutaminolysis (15, 16). We therefore checked the effect of glutaminolysis and lipogenesis inhibitors on mutant-wild-type cell competition in culture. The glutaminolysis inhibitor CB839 did not significantly affect *Pik3ca*^{H1047R} cell fitness (**Extended Data Fig. 7p**). However, inhibiting either *Fasn* or *Acc*, two of the main steps in lipogenesis pathway, reduced the advantage of mutant over wild-type cells (**Extended Data Fig. 7p**). Treatment with the *Srebp1c* inhibitor Betulin did not significantly alter mutant cell behavior, arguing that the changes in *Pik3ca* mutant cells are not mediated by transcriptional changes (**Extended Data Fig. 7p**). Consistent with this, mutant cells had similar expression of most *Srebp1c* lipogenesis-related targets (*Acly*, *Acaca*, *Acacb*, *Fasn* or *Scd1*), only the mitochondrial citrate exporter (*Slc25a1*) and *Scd2* showed increased expression in mutant cells (**Extended Data Fig. 6l**).

In summary, *Pik3ca* mutant cells show metabolic rewiring with increased HIF1 α signaling and glycolysis. This is only partially responsible for their competitive advantage. The demonstration that lipogenesis

inhibition reduces mutant cell fitness suggests that multiple metabolic pathways downstream PI3K pathway promote mutant cell competitiveness.’

4. Fig. 6 – The authors treat keratinocytes with DCA (glycolysis inhibitor) however the effect is very modest as by the end of treatment, most of the cultures (~70%) are still PIK3CAH1047R despite a significant decrease in lactate secretion again, arguing against glycolysis being solely responsible for increased cellular fitness in PIK3CAH1047R. How do the authors explain this? Additional, more potent inhibitors of glycolysis such as 2-DG (Zhao J et al. Cell Death Discovery, 2019) should be tested – titrated for optimal non-lethal dose – to effectively inhibit glycolysis.

Please refer to the previous comment with respect to 2-DG and modest DCA effects *in vitro*.

For *in vivo* studies, cell-based pharmacodynamic markers are needed to corroborate whether the drug effects are direct (as hypothesized) or indirect. For example, the authors should show PI3K/AKT/mTOR activation by immunofluorescence in mutant and wild-type clones *in vivo* upon MET/DCA treatment to confirm expected differences in PI3K signaling between PIK3CAH1047R and WT clones is maintained, and the competitive fitness of mutant cells is blocked downstream at the level of glycolysis. As these agents may also modulate islet function and insulin sensitivity, immunofluorescence for pINSR on esophageal epithelial cells and serum insulin levels should be measured in these *in vivo* experiments. These results may help differentiate whether the effects of metformin and DCA are due to altering insulin production/sensitivity or directly acting on keratinocytes to mediate their effects. Lastly, the addition of MET appears to increase basal cell number in WT clones but decrease in mutant clones (Figs. 6i-j). Is this due to an off-target effect of metformin on mutant clones or saturated levels of glycolysis?

As discussed above we no longer attribute glycolysis as the sole mechanism for the mutant cell advantage. We would submit that resolving the complex mechanisms by which DCA and metformin reduce mutant cell advantage *in vivo* is beyond of the scope of this manuscript with its focus on a genetic approach *in vivo* and in primary organotypic cultures. We have revised this section and now use the DCA results only to support the argument that the increased glycolysis of mutant cells may partially explain the mutant cell fitness advantage. The *in vitro* data suggest that part of the reduction in mutant advantage after DCA treatment is keratinocyte specific. However, as the Reviewer points out, this inhibition is only partial, we now highlight that DCA may also act through other mechanisms such as reducing insulin levels in serum (25, 28). We have now added a sentence commenting on this to the manuscript:

“Because of the impact of DCA on multiple organs *in vivo*, it is important to note that its effect on *Pik3ca*^{H1047R/wt} cell fitness could be via the direct targeting of esophageal glycolysis or by reducing insulin blood levels or a combination of both (23-25).”

However, if the Reviewer and editorial team consider that DCA results are inconclusive, we will remove them from the manuscript.

Regarding metformin, as the Reviewer pointed out, and in light of the 2DG and glucose deprivation results, we wish to be cautious in attributing its effect on mutant fitness advantage to direct activation of glycolysis in the esophageal epithelium. Metformin has multiple effects on the organism (29).

However, it is widely accepted that metformin has an antidiabetic effect. Therefore, we moved the metformin results to a new figure together with the high fat diet and Akita mice results. The aim of this section is to demonstrate that pro- and anti-diabetic conditions can modify the mutant cell advantage. Therefore, we now only use Metformin as a drug reported to reduce glycaemia and insulin levels in mice (30) alongside two pro-diabetic models (a genetic type 1 diabetes model, Akita, and a diet-induced overweight model). These three metabolic alterations modulate PI3K mutant cell expansion. Finding the mechanism(s) by which metformin reduced mutant advantage *in vivo* is complex, likely multifactorial and we submit is out of the scope of the paper. Even if we could prove that metformin modified insulin sensitivity in the esophagus, it would be challenging to establish a causal relationship between this and mutant cell fate advantage and separate it from the effects of metformin on glycolysis and/or other pathways in keratinocytes. Although *in vitro* metformin treatment experiments suggest that metformin effect on mutant advantage may be, at least partially, caused by a direct effect on esophageal keratinocytes, we now acknowledge that it may also have systemic effects that directly impact in the competitive advantage of mutant cells (see below).

As the Reviewer points out, there is a trend to increase wild-type average basal clone size after metformin treatment, but it is not significant. The statistics now have been included (**Figure 6c, Reviewer Figure 7**). Also, metformin does not significantly alter the ratio of suprabasal/basal wild-type cells (**Extended Data Figure 8a and Reviewer Figure 7**). However, metformin does modify wild-type clone size distributions (**Figure 6b and Extended Data Figure 8b and Reviewer Figure 7**), suggesting that metformin may act both on wild-type clones and on mutant cells, reducing the differences between the two populations. We now comment this possibility in the main text:

“Such effect might be caused by a dual effect on mutant and wild-type cells, as, although it does not significantly affect the average clone size or stratification ratio, metformin alters wild-type clone size distribution (**Fig. 6b and Extended Data Fig. 8b**).”

We now discuss the *in vivo* metformin data in the text as follows:

‘Collectively our results argue that *Pik3ca*^{H1047R/wt} mutant cells have increased activation of the Insulin/PI3K signaling pathway, which provides them with higher cell fitness. Therefore, organismal conditions affecting the Insulin/PI3K signaling pathway may alter the fitness of *Pik3ca*^{H1047R} mutant clones *in vivo* (31). To test this hypothesis, we analyzed mutant clonal expansion in three different organismal metabolic scenarios that affect the Insulin/PI3K axis, we tested the effect of metformin a widely used antidiabetic agent that increases insulin sensitivity and mouse lifespan (30, 32) and two pro-diabetic contexts, namely (1) a genetic model of type I diabetes (Akita mice), which harbors a mutation in the insulin-2 gene that results in reduced circulating insulin levels with age (33, 34), and (2) a high fat diet (HFD) model in mice

which alters PI3K signaling and promotes insulin-resistance, increased body mass and hyperinsulinemia (35-37).

Metformin treatment reduced mutant clone size and increased the proportion of differentiated mutant cells towards the levels observed in wild-type clones (Fig. 6a-c and Extended Data Fig. 8a). A non-parametric two-factor analysis (see methods) revealed that metformin significantly decreased the differences between wild-type and mutant basal clone size distributions (Extended Data Fig. 8b). Such effect might be caused by a dual effect on mutant and wild-type cells, as, although it does not significantly affect the average clone size or stratification ratio, metformin alters wild-type clone size distribution (Fig. 6b and Extended Data Fig. 8b). Metformin treatment *in vitro* activates glycolysis in wild-type cells (Extended Data Fig. 8c) and reduced the expansion of *Pik3ca*^{H1047R/wt} cells in *in vitro* competition experiments (Extended Data Fig. 8d-f), suggesting that the *in vivo* effect of metformin could be partially explained by a direct effect on the metabolic differences between wild-type and mutant esophageal cells. In summary, we conclude that metformin reduces the fitness advantage of *Pik3ca* mutant clones in mouse esophagus.

Reviewer Figure 7. a, Protocol: *Cre-RYFP* control and *Cre-Pik3ca*^{H1047R-YFP/wt} mice were induced and treated with/without metformin (MET, see methods). Clone sizes were analyzed at 28 days. (**b**, left and center panels) Heatmaps represent clone size frequency with number of basal and first suprabasal layer cells indicated, in animals from **a**. Black dots and dashed lines indicate geometric median clone size. (**b**, right

panels) Heatmaps showing differences between treatment and control in *Cre-RYFP* (upper panels) or *Cre-Pik3ca^{H1047R-YFP/wt}* (lower panels) animals. 2D Kolmogorov-Smirnov test. **c**, Average basal clone sizes for each strain and treatment from **a**. Data comprises clones with at least one basal cell. Bars are SD. Two-tailed unpaired *t*-test. N=431-917 clones from 5-10 animals per condition (see Supplementary Table 1 for numbers). **d-e**, *Cre-RYFP* control and *Cre-Pik3ca^{H1047R-YFP/wt}* mice were induced and treated with/without MET (see methods). Proportion of suprabasal cells per clone (**d**) and basal clone size distributions (**e**) for each strain and treatment were analyzed at 28 days. Only first suprabasal layer cells were analyzed in **d**. Dots indicate mean and lines standard deviation in **e**. Kolmogorov-Smirnov test to compare wild-type and mutant basal clone distributions. Contrast ART-C Post-hoc test of differences between distributions (methods). n=431-917 clones from 5-10 animals per condition (see Supplementary Table 1).

5. Fig. 7 – As described, Akitahet mice have reduced system insulin levels and hyperglycemia both of which could contribute to a competitive fitness imbalance between PIK3CAH1047R and WT clones but affecting distinct cells. Loss of insulin would be expected to reduce the relative fitness of WT cells, whereas high glucose might be predicted to increase the fitness of mutant cells that may be taking up more glucose for aerobic glycolysis. Evaluation of INSR activation (and downstream PI3K/AKT) in esophageal tissue of PIK3CAH1047R/wt; Akitahet comparing mutant versus wild-type clones would help distinguish these possibilities. Furthermore, the authors could parse the effects on basal expansion of increased glucose availability vs. hypoinsulinemia on PIK3CAH1047R vs. WT clones by lowering glucose in a pancreas-independent manner (e.g., with an SGLT2 inhibitor; Nasiri AR et al. *Cancer Metab*, 2019) or by restoring sustained insulin levels in Akitahet mice using insulin pellets, as described previously (Wang Y et al, *Cell reports*, 2018).

Please refer to the comment below.

6. Fig.–7 - It is not clear that the increase in PIK3CAH1047R clones in the esophagus of mice fed a high fat diet (HFD) is due to hyperinsulinemia, especially considering that hyperinsulinemia due to a HFD may result in a loss of insulin sensitivity in insulin-sensitive tissues, perhaps even in the esophageal keratinocytes. Immunofluorescence for changes in INSR activation and downstream signaling (as described above) due to HFD can validate that the increase in PIK3CAH1047R clones is due to increased activation in mutant clones or loss of insulin sensitivity in WT clones.

We agree with the Reviewer that it would be interesting to understand the mechanism behind the increased advantage of mutant clones both in the Akita background and on a high fat diet. We further agree that it is unclear whether the increased expansion of mutant clones in the diabetic context is due to hypoinsulinemia or hyperglycemia, a combination of both, or an indirect effect of diabetes on another parameter. We support the hypothesis that the Reviewer elaborated, suggesting that loss of insulin/insulin resistance could have more effect on wild-type cells while hyperglycemia may enhance mutant fitness. This is consistent with observations with both models. In low insulin (or lower insulin sensitivity) conditions, only mutant cells exhibit sustained PI3K pathway activation to drive clonal growth. Because our *in vitro* results suggested that lack of glucose does not affect mutant advantage, we favour

insulin levels/sensitivity as the most important factor in influencing mutant advantage. However, for the technical reasons described above, performing the experiments required is infeasible.

The main aim of the manuscript is to analyse genetic selection of *Pik3ca* mutant clones and describe how these cells, through modestly greater PI3K signalling than their neighbours, undergo reduced cell differentiation and clonal expansion in normal epithelium. In addition, we show that three metabolic conditions known to modulate insulin signalling alter mutant clone expansion. However, each of these conditions is complex, and their impacts on the esophagus or other epithelia are unstudied. Unravelling these mechanisms is a major undertaking, beyond the scope of this manuscript. We have modified the discussion to reflect this as follows:

“In vivo models of metabolic conditions such as diabetes and diet-induced overweight increase mutant cell expansion while the antidiabetic drug metformin decreases it (Fig.8). Although further work is needed to define the mechanism behind their effect on mutant advantage, our results suggest that reducing insulin signaling in peripheral tissues, via lower insulin levels or insulin sensitivity could enhance mutant fitness. ”

7. Fig. 7- Given the importance of insulin levels in altering mutant clone expansion in mice, the authors should show the diabetes status for human data showing an accumulation of PI3K mutant clones in esophageal tissue of obese vs non-obese patients and whether it alters the findings.

We agree with the Reviewer that reporting the diabetes status would be relevant for this study. None of the donors were previously diagnosed as diabetic, we now state this in the methods section:

“The Body Mass Index (BMI) of each donor was used to classify them into non-overweight (BMI<25) or overweight (BMI>25). None of the donors used in the study had a previous diagnosis of diabetes.”

Unfortunately, we do not have the information on insulin or blood glucose levels of the donors to perform deeper analysis on diabetic status. Our research ethics only allows very limited sampling during a 15 min window post organ retrieval in the transplant donors from whom esophageal samples are obtained, so this was not performed in the present study. We agree it would be very interesting to repeat this analysis in another cohort of donors in the future.

8. Extended Data Fig. 8. There are well-known sex-specific differences in weight gain, insulin resistance, and islet resilience comparing male and female mice fed a lard-based HFD. The authors state that there were no “gender” specific differences in their experiments in the Methods, but that would be unlikely based on short-term HFD treatment and the data presented. This should be clarified.

Following this Reviewer comment, we analysed data by sex (see graphs below). As the Reviewer points out, there are sex differences in weight and insulin levels between males and females. However, we observed similar trends in weight and insulin levels after high fat diet in both sexes (Reviewer Figure 8 a and b). When we analysed clone sizes and stratification ratios by sex, we found that HFD increased clone size and decreased stratification in both sexes (Reviewer Figure 8 c and d). Overall, we did not observe a

consistent trend towards different behaviour in males and females, therefore we did not include this analysis in the manuscript. However, if the Reviewer and the editor consider it desirable, we are happy to include it. We have corrected the term “gender” to “sex” specific differences.

Reviewer Figure 8. a-b, Mice were fed for 4 weeks with a normal chow or high-fat diet. **a**, Body weight measured at weeks 0 and 4 post diets classifying animals per gender. Each dot corresponds to one animal, lines link two weights of the same animal (n=10-16 mice). Two-tailed paired *t*-test. **b**, Insulin levels in blood, measured at the end of the experiment in animals from a. Two-tailed unpaired *t*-test. **c-d**, Wild-type (*Cre-RYFP*, *YFP*) and mutant (*Cre-Pik3ca^{H1047R-YFP/wt}*, PI3K) mice were induced fed a normal chow (CTL) or high fat diet (HFD) and tissues collected 28 days post-induction. **c**, Average basal clone sizes for each strain and treatment and sex, considering all clones with at least one basal cell. Dots indicate the average clone size of a mouse. n=5-9 mice. Two-tailed unpaired *t*-test. Source data are shown in Supplementary Table 1. **d**, Average proportion of suprabasal cells per clone for each strain and treatment classifying animals per gender, only first suprabasal cells were counted. Each dot corresponds to one animal. n=5-9 mice. Two-tailed unpaired *t*-test.

Minor concerns:

1. Axes labeling (throughout all figures) are too small. Larger labeling needed.

Following the reviewer's suggestion, we have increased the font size of all the graphs labelling by 1 p.

2. Typo on page 14; "... we induced Cre-Pik3caH1047R-YFP/wt mice at a level than used above and aged them for 3 months..."

We removed these words to simplify the sentence as they were only aimed to explain that in EdU experiments we injected 1mg TAM. After this Reviewer comment we realised that this was not properly specified in the methods section, which we have now modified as follows:

"Analysis of *in vivo* proliferation by EdU labeling

Three months after induction with one intraperitoneal (i.p.) injection of 80 mg/kg β -naphthoflavone (MP Biomedicals 156738) and 1 mg tamoxifen (Sigma Aldrich N3633), 10 μ g of EdU in PBS was administered by intraperitoneal injection 1 h before culling. Tissues were collected and stained with EdU-Click-iT kit and immunofluorescence as explained below. EdU-positive basal cells were quantified from a minimum of 10 z-stack images. "

REVIEWER #2 (in vivo, metabolism)

Title: Organismal metabolism regulates the expansion of oncogenic PI3KCA mutant clones in normal esophagus

This is an interesting study that looks to investigate the acute consequences of PI3KCA mutation in oesophageal epithelium. The authors show convincingly that PI3KCA mutant clones have a competitive advantage in the epithelium driven through metabolic rewiring. The role of PI3KCA in driving cancer is not so clear (obviously there are mutations) however the impact of PI3KCA in the cancer is marginal (although statistically significant). The authors importantly don't overplay this finding. Overall I feel this paper will be of interest to the community if a number of important issues are addressed.

Specific comments:

- **Induction regime:** There is no comment on the difference between X1 induction and X3 induction – how does it affect clone numbers per se and not just in terms of tumour numbers?

We thank the Reviewer for spotting this. We now introduced the following sentence in the methods section:

“Chemically induced mutagenesis

To generate mutations in the esophageal epithelium, mice were treated with diethylnitrosamine (DEN) (Sigma, catalog no. N0756) at 40 mg per 1,000 ml sweetened drinking water for 24 h on 3 days a week (Monday, Wednesday and Friday) for 8 weeks. When indicated, mice were induced using one or three intraperitoneal (i.p.) injections of 80 mg/kg β -naphthoflavone (MP Biomedicals 156738) and 1 mg tamoxifen (Sigma Aldrich N3633), **resulting in 2.9 \pm 0.9% and 10.2 \pm 2.8% (mean \pm s.d.) recombined epithelium by area respectively of recombination respectively 10 days after the last induction.** After each dosage mice received sweetened water until the next DEN treatment. Control mice received sweetened water as vehicle for the length of the treatment. After the 8 weeks, all mice were administered normal water.”

- Mouse numbers in cancer experiments

Page 6 – Fig 2c – The number of mice used is very high. How was power originally decided for these experiments? I doubt the authors started with the assumption PI3KCA would only provide a very small amount of difference. If the authors go back in now with a power calculation now they know the difference, is the experiment appropriately powered?

Power calculations were not feasible as the effect size was not known. As the Reviewer suggests, we did not start assuming the differences would be so small. We expected that the protocols used would result

in variability in the number of tumours per mouse (38). We performed three independent experiments for the 1x induction regime and two for the 3x induction regime with 35 and 23 mice in total respectively. As it can be seen in Fig. 2c, reproduced below, there is indeed large variation in the number of tumours per animal. Had we known the effect size provided by the 3x induction regime, 12 mice (not 23) would have been sufficient to detect a difference in the mutant animals according to power calculations.

Main Text Fig. 2: Effect of heterozygous *Pik3ca^{H1047R}* expression in mouse esophageal tumorigenesis. a, Protocol 1: *Cre-Pik3ca^{H1047R-YFP/wt}* mice were induced 1 or 3 times with BNF and TAM, and treated with diethylnitrosamine (DEN) for 8 weeks starting 4 weeks post-induction. Uninduced mice were used as controls. Tissues were collected before exceeding the permitted humane endpoint or at 1 year after DEN. **b**, Typical DEN-treated esophagus opened and flattened epithelial side up showing four tumors (yellow arrows). Scale bar, 2 mm. **c**, Number of macroscopic esophageal tumors in DEN-treated mice, uninduced, or induced 1 or 3 times before DEN treatment. Mann-Whitney test versus uninduced (n= 33, 35 and 23 animals, respectively). Red lines indicate average values. **d**, Protocol 2. *Cre-Pik3ca^{H1047R-YFP/wt}* mice were treated with DEN for 8 weeks followed by BNF and TAM induction. A subgroup of animals was then treated with the tumor promoter sorafenib (SOR) for 6 weeks. Tissues were collected before exceeding the permitted humane endpoint (see methods) or 1 year post-DEN treatment. Control groups received all treatments but were uninduced. **e-f**, Number (**e**) and size (**f**) of macroscopic esophageal tumors. Mann-Whitney test (n= 33, 13, 7 and 13 mice, as they appear in the graph). Red lines indicate average values. **g**, Frequency of missense mutations (MM) for the indicated driver genes detected in human ESCCs from data collected from the TCGA and ICGC databases. Driver genes were selected using the Intogen tool

(<https://www.intogen.org/search>). Only driver genes with missense mutation frequency >2% are shown. Source data in Supplementary Table 1.

With respect to the effect size of the differences in tumour size between induced or uninduced animals, observed in the induction post DEN+SOR treatment regime, (Figure 2f) the experiment is correctly powered. Power calculations estimate that 56 tumours would be required to detect a difference of this size and 51 tumours were analysed.

- In Vitro Competition experiments

Pag 15-17 – In vitro competition experiments – My understanding is that *Pik3ca*Mut cells expressed EYFP, whilst wildtype cells expressed RYFP, whereby (as shown in Fig 4.a/b) *Pik3ca*Mut can be distinguished as positive green cells, whilst wildtype cells can be seen as yellow cells, which makes the set up perfect for cell competition studies and see how the cells outcompete each other. However, all subsequent images shown, particularly those of the cell competition experiment, are only with the RYFP marker, meaning that all it can be seen is a change in the wildtype cells, and not necessarily how the *Pik3ca*Mut cells are behaving. What was the reason for this choice? Co-staining of both cell types should have been taken.-
 Pag 24 – extended data fig 6.h – as per the in vitro cell competition experiment in pag 15-17, why wasn't a co-stain for both *PIK3camut* and wildtype cells not performed to more directly determine the interaction and competition between the two cell types?

As the Reviewer points out, both *Pik3ca*^{H1047R/wt} cells and YFP-WT cells express YFP when they are induced. After this Reviewer comment we realised that the green labelling of YFP staining of *Pik3ca* mutant cells and yellow labelling of YFP staining of WT-YFP cells in Extended Figure 4 was very confusing. All colours in the confocal microscope images shown are pseudo-colours generated by the imaging software, but both genotypes have a YFP reporter and are fluorescently identical, so we have now changed the green labelling to yellow to be consistent with the rest of the YFP staining *in vitro* shown in the manuscript (see Reviewer Figure 9).

Reviewer Figure 9. a-b, Generation of fully induced primary WT-RYFP (a) and *Pik3ca*^{H1047R/wt} (b) esophageal keratinocyte cultures. Primary esophageal keratinocytes were isolated from uninduced *Rosa26*^{RYFP/RYPF} animals (a) or uninduced *Pik3ca*^{H1047R-YFP/wt} animals (b). Cells were incubated either with Cre-expressing adenovirus (Ad-Cre) or null adenovirus (Ad-Null). Right panels show representative images of Ad-Cre or Ad-Null treated cultures stained for YFP (yellow) and DAPI (blue). Scale bars, 20 μm. **c,** *In vitro* cell competition experimental protocol. *Pik3ca*^{H1047R/wt} and *Pik3ca*^{wt/wt} primary keratinocytes obtained as in b were mixed with WT-RYFP cells (as in a) and maintained at confluence. The proportion of WT-RYFP cells was quantified by flow cytometry. The proportion of WT-RYFP after treatment was normalized to the initial WT-RYFP cell proportion.

Due to the low expression of the *Pik3ca* endogenous locus in the oesophagus, we can only detect the induced *Pik3ca*^{H1047R-YFP} cells by microscopy with a long immunostaining protocol, which includes a 5-day incubation with the primary anti-GFP antibody (which recognises YFP as they are very similar antigens). We tried but were unable to optimize a staining protocol that allows us to detect the induced *Pik3ca*^{H1047R-YFP} cells by flow cytometry. To be able to rapidly quantify by flow cytometry or microscopy the cell competition between mutant and wild-type cells under different conditions and times, we decided to analyse the competition between either induced or uninduced *Pik3ca* mutant cells versus a fluorescent wild-type strain.

We selected the WT-YFP strain as wild-type for cell competition experiments because the allele has been extensively characterized in esophageal progenitors *in vivo* and shown to be a neutral reporter (39). Critically, in a mixed culture with uninduced *Pik3ca*^{H1047R-T2A-YFP} cells, the proportion of WT-YFP remains constant (Figure 4c). This argues that an advantage of the induced *Pik3ca*^{H1047R-T2A-YFP} cells over RYFP-WT cells would be an advantage over the uninduced *Pik3ca*^{H1047R-T2A-YFP} cells as well. We did not perform immunostaining to detect the mutant cells because the YFP expression levels are around 4 orders of magnitude higher in WT-YFP cells than in *Pik3ca*^{H1047R-YFP} mutant cells (see histogram in Extended Fig 4 c,

Reviewer Figure 9). This means that when immunostaining and microscope settings are setup to detect the mutant cells, the WT-YFP cells show excessive intensity that masks their neighbouring cells. Therefore, when a fully induced population of mutant cells was mixed to WT-YFP cells, only WT-YFP cells were detected in the microscopy experiments used to detect the wild-type cells.

Functionally tested fitness genes

- Pag 22 – At the end of the paragraphs the authors report “Finally, depletion of most glycolysis pathway genes and some mutant-induced metabolism regulators, reduced cell fitness, including *Ldha*, which is not implicated in mitochondrial glucose oxidation (Fig. 5j and Extended Data Fig. 5f)”. From the data presented in Fig5 and till that point in the manuscript, the inhibition of specific targets can be seen, which can lead to the postulation that there is a reduction in fitness. However, a reduction in cell fitness is not directly shown or investigated, making this statement a bit of an over-reach (specifically at that point in the manuscript).

Former **Figure 5j** and **Ext Data Figure 5f** summarized the results of a CRISPR screen. In this experiment, we infected Cas9-expressing primary oesophageal cells with lentiviruses bearing gRNAs targeting multiple genes. After few days, we collected the time 0 sample and subsequently we collected the 3 week time-point. The enrichment or depletion of the gRNAs targeting each gene was determined using MageCK software (40), which identifies how targeting each of these genes affects the cell selection during the 3 weeks of the experiment (with a given fold change and false discovery rate) (**Fig 5j**). We then validate one of the targets, *Hif1a*, in the competition between mutant and wild-type cells. We have revised the text as follows:

‘Next, we investigated whether the metabolic switch observed in *Pik3ca* mutant cells impacted their competitive fitness or was just a consequence of the increased HIF1 α signaling. The connection between glycolysis and cell competition is controversial. Metabolic reprogramming towards glycolysis alters cell competitive fitness in *Drosophila* (17, 18). In mammalian keratinocytes, glycolysis has been linked to cell fate regulation and differentiation suggesting that glycolytic activation may confer a proliferative advantage in the esophagus (19-21). gRNAs targeting most glycolysis pathway genes were depleted in the screen, including *Ldha*, which is not implicated in mitochondrial glucose oxidation, implying glycolysis regulates cell fitness (**Extended Data Fig. 5c-e**). Intriguingly, however, deletion of the glucose transporter *Glut1* showed the opposite effect. This argues that modulation of glycolysis at different levels may have different effects on competitive fitness. This may be explained by effects on differentiation. Cytosolic glucose has been shown to promote keratinocyte differentiation while activation of downstream glycolysis inhibits keratinocyte differentiation (12, 22). Consistent with a link between glycolysis and mutant cell fitness advantage, mutant cells are more sensitive to the loss of *Ldha*, *Aldoa*, *Eno1* and *Pgam1* than wild-type cells (**Extended Data Fig. 5c-e**) and similar effects were observed in INS treated WT cells (**Extended Data Fig. 5f**). ‘

- Pag 23 – Same observation as above – The authors report “Targeting *Atf4*, *Hif1 α* /*Hif1 β* , *Myc*, *Srebf1* or *Srebf2* reduced cell fitness (Fig. 5j and Extended Data Fig. 5f), suggesting that PI3K regulates metabolic gene expression and cell fitness through multiple factors in parallel 45,46.” However, the data presented

in Fig 5 and extended data Fig 5 shows the effect at the gene expression level of the CRISPR screen and not how the depletion of specific genes directly affects cell fitness. This will only be shown later in Fig 6 and thus this statement feels preemptive and a bit of an over-reach at that point in the manuscript.

The results presented in former **Fig. 5j** and **Extended Data Fig. 5f** did not show the gene expression levels. Enrichment or depletion refer to enrichment or depletion of cells that incorporated a gRNA targeting each gene (please refer to the previous comment for a more detailed explanation of the analysis). The enrichment or depletion of these cells indicates that gene regulates cell fitness. With the aim of clarifying this we have modified the text as follows:

'The PI3K/mTOR pathway modulates signaling and metabolism at transcriptional and posttranscriptional levels through multiple downstream effectors (14). Although gRNAs targeting the Foxo transcription factors did not largely modify cell fitness, gRNAs targeting *Atf4*, *Hif1α/Hif1β*, *Myc*, *Sreb1* or *Sreb2* reduced cell fitness (**Fig. 5b and Extended Data Fig. 5c**), suggesting that PI3K might partially regulate cell fitness modulating gene expression through multiple transcription factors in parallel (14, 41).'

We also modified the scheme and text of the CRISPR screening figure (**Reviewer Figure 10, new Figure 5**) to clarify the experimental setup:

Reviewer Figure 10. CRISPR screening of targets affecting cell competition in primary esophageal keratinocytes. a, Overview of experimental steps performed during a CRISPR-Cas9 targeted cell competition screen. Primary cultures from *Pik3ca*^{H1047R/wt} *Rosa26*^{Cas9/wt} mice were induced (*Pik3ca*^{*/wt}) or uninduced (WT). Cells were harvested and infected with a lentiviral gRNA library targeting PI3K/mTOR related genes and plated in inserts. 5 days later, a 0 week time-point was collected and the rest of cultures maintained in a minimal medium (CTL) or minimal medium

supplemented with 5 µg/ml insulin (INS) for 3 weeks changing medium 2-3 times per week. Then a 3 week time-point was collected. gRNA relative abundance between the 3w and 0w time-points is expressed as Log_2 (fold change). Volcano plot shows the enrichment score and the Log_2 (fold change) of the genes targeted by the CRISPR screening in the WT background in a control situation. Significantly depleted or enriched gRNAs ($\text{FDR} < 0.1$ and $> 10\%$ Fold change difference) are depicted in blue or orange respectively, while unchanged gRNAs are depicted in grey. $n = 2-3$ biological replicates, 10 gRNA per gene. **b**, Illustration of the results of the CRISPR screening in the control condition in the WT background (left panel) or *Pik3ca*^{H1047R/wt} background (right panel) showing gene targets related to the PI3K pathway and downstream transcriptional regulation with a color code indicating its Log_2 (fold change) between the 3 w and 0 w time-points. Genes targeted by gRNAs that are significantly enriched or depleted ($\text{FDR} < 0.1$ and $> 10\%$ Fold change difference) are indicated with a bold red labeling. Pathway activation or inhibition are shown using green and red arrows respectively. Yellow boxes indicate the enzyme isoform most expressed in esophageal primary cells. Source data are shown in **Supplementary Table 1**.

Competition in overweight donors

Page 32 – The authors reported “Finally, the specific enrichment in overweight donors was caused by mutations present in the kinase domain (Extended Data Fig. 8k), including H1047R, in agreement with the increased fitness of *Pik3ca*H1047R clones observed in mice under HFD (Fig. 7e and f).” – This statement is a bit misleading as the frequency of H1047R between overweight and non-overweight donors was similar, 9.7 and 9.3, respectively. Unless this is still statistically significant, but no Statistics were reported to make that conclusion.

The Reviewer is correct, the frequency of *PIK3CA*^{H1047R} mutant clones versus all *PIK3CA* mutant clones in lean and obese people is similar. The difference is in the proportion of all pathogenic mutant clones per cm^2 per donor. We lack the statistical power to detect differences in individual mutations. We have changed the text to:

“Finally, although the number of samples was not large enough to see significant differences in any specific mutation, an increase in the density of pathogenic variants of the kinase domain was observed in overweight individuals. This was consistent with the increased fitness of *Pik3ca*^{H1047R} clones observed in mice fed a HFD (Fig. 7e and f).”

Samples per patients

For the human-based analyses multiple samples per patient – 844 from 9 donors first and then 698 from 10 individuals – were taken. When determining the *PIK3CA* mutant clone density, including clone density of the specifically H1047R, was the analysis performed on equal number of samples per patient? If not, was a normalisation performed to take into account unequal numbers of samples per patient? How would this affect the results and conclusions?

The analysis was not performed on an equal number of samples from each donor. The mutation frequency was normalized by the total area sequenced from each donor to be able to compare donors with different numbers of samples sequenced. Thanks to this Reviewer comment we realized this was inadequately

explained in the methods section. We have therefore introduced the following sentence in the Human DNA sequencing section:

“*Pik3ca* mutant clone density was calculated as the number of missense mutant clones classified as stated above and normalized by the total area sequenced in each particular donor.”

Wouldn't there be a risk of over/underestimating based on the mutational signature of a particular patient with more samples analysed compared to other patients?

All the analyses are made per donor, therefore will not be affected by such bias, as a heavily mutated donor has the same weight as a less mutated donor. We have introduced a sentence in the methods section to clarify this point:

“All analyses were performed per donor and therefore are independent of the area of tissue sequenced in each donor.”

Clonal dynamics in HFD

The authors showed early in the manuscript that supraphysiological doses of insulin, causing overactivation of the PIK3 pathway, significantly reduced the competitive advantage of *Pik3ca* mut cells over the wildtype cells in mixed cultures. However, in their assessment of the mice on HFD, which was described as increasing mouse body weight and hyperinsulinemia, only small changes in wildtype clones and a significant increase in average mutant clones was seen. Was the expected phenotype of hyperinsulinemia assessed and seen in these mice? If yes, findings in these mice would deviate from the in vitro observations and should be commented on. If no, how can conclusions be made?

We indeed measured blood insulin levels and confirmed that HFD treated mice show hyperinsulinemia (**Extended Figure 8k, below**).

Main text Extended Data Figure 8k. Insulin levels in blood in mice fed on Chow or high fat diet for 4 weeks, each dot is one mouse. Error bars are mean \pm s.e.m. Two-tailed unpaired *t*-test.

High insulin levels in blood reflect a reduced insulin sensitivity of peripheral tissues (42), therefore HFD is expected to reduce insulin sensitivity and therefore intracellular insulin signalling. Consistent with this, a similar effect on mutant clonal expansion is seen both in hypoinsulinemic Akita mice or hyperinsulinemic HFD mice.

In low stimulus conditions (such as growth factor starvation) the H1047R mutant increases PI3K signalling (**Extended Data Figure 1c-f**), suggesting that mutant cells are less dependent on external insulin/growth factor signalling to activate PI3K pathway. This led us to hypothesise that under HFD or Akita mice WT cells would have lower overall activation of the PI3K signalling, which would confer an additional competitive advantage to mutant cells, offering an explanation to the increased mutant advantage caused by both diabetic models.

The supraphysiological insulin doses used *in vitro* are much higher than the *in vivo* insulin concentrations even in hyperinsulinemia. At this dose (typically used in primary keratinocyte culture), insulin activates insulin receptor and IGF-1 receptor which leads to overactivation of the PI3K pathway, which was the aim in our experiment (43). When we overactivated the pathway both in wildtype and mutant cells, we observed a reduction in the gene expression differences between mutant and wild-type cells and a mutant cell fitness advantage over wildtype cells. This is consistent with mutant fitness advantage being due to differential activation of the PI3K pathway between wildtype and mutant cells.

After this Reviewer comment, we realized this possible explanation was not addressed in the text which we have revised as follows:

Results:

“As previously reported, HFD significantly increased mouse body weight and hyperinsulinemia as compared to control diet, suggesting the onset of insulin resistance (**Extended Data Fig. 8j and k**)(37, 42, 44).”

Discussion:

“*In vivo* models of metabolic conditions such as diabetes and diet-induced overweight increase mutant cell expansion while the antidiabetic drug metformin decreases it (**Fig.8**). Although further work is needed to define the mechanism behind their effect on mutant advantage, our results suggest that reducing insulin signaling in peripheral tissues, via lower insulin levels or insulin sensitivity could enhance mutant fitness. “

Minor

comments

- Page 3 – Fig 1a – Difficult to interpret, particularly in terms of mutant density (unless I physically counting).

Figure 1a is only illustrative, intended to convey that *Pik3ca* mutant clones coexist with multiple other mutants in the normal esophagus, although the proportions of mutants did reflect the actual coverage of

mutant clones, it was not meant to be a quantitative representation. In light of this Reviewer comment, in order to aid the interpretation of this figure we substituted the Fig1a panel with a representative circle plot representation similar to the ones published in the 2018 paper from which the data was reanalysed. The plot represents the mutant clones found on a 1cm² of normal esophagus of one of the donors analysed in that paper (a 49-51 year old male). In addition, in order to get quantitative information on clonal density, we have now introduced a graph showing the density of mutant clones for each gene (taking all 9 donors into account) in **Figure 1b** (see below) to assist the interpretation of the figure.

Reviewer Figure 11. a, Schematic representation of the mutant clones in an average 1 cm² of normal esophageal epithelium from a 48-51 year old male donor from (45). To generate the figure a number of samples from the donor are randomly selected and the mutant clones detected are represented as circles and randomly distributed in space. **b**, Average VAF (top graph) and frequency (bottom graph) of missense mutations (MM) detected more than once per gene, arranged from largest to smallest. *PIK3CA* highlighted in red. n=844 samples from 9 donors. **c**, Distribution of *PIK3CA* MMs classified into pathogenic/gain of function (Path/GoF) or Unknown/No effect (Unkn/NE) (methods). VAF distribution of synonymous mutations in all genes is also shown. Medians (red) and quartiles (grey lines) are represented. Two-tailed Mann-Whitney Test. n=23, 26 and 603 mutant clones respectively from 9 donors. **d**, Frequency of MM codons in the p110 α protein. Path/GoF mutations are shown in red. n=41 mutant clones from 9 donors. **e**, Comparison of the VAF distribution of *PIK3CA*^{H1047R} MMs with other *PIK3CA* MMs classified as Path/GoF or Unkn/NE. Medians (red) and quartiles (grey lines) are represented. Two-tailed Mann-Whitney Test. n=8, 15 and 26 mutant clones respectively from 9 donors. Source data are shown in Supplementary Table 1.

Page 5 – Fig 1b – Quite a bit of green background-staining. This is seen in page 15 fig b too.

The staining of the *Pik3ca* mutant clones usually has background probably due to the low expression of *Pik3ca* in the oesophagus. We chose to show representative images that reflect this.

- Page 6 – Fig 2c – The number of macroscopic oesophageal tumours are being compared between uninduced vs x1 induction and uninduced vs x3, however this is not clear from the figure or figure legend and should be made clearer.

We apologize, we omitted a line describing what is being compared, we thank the Reviewer for spotting this. Now we modified the figure legend to:

“c, Number of macroscopic esophageal tumors in DEN-treated mice, uninduced, or induced 1 or 3 times before DEN treatment. Mann-Whitney test versus uninduced (n= 33, 35 and 23 animals, respectively). Red lines indicate average values.”

- Page 8 – Extended fig 2d: the figure is the same as fig 2a with no obvious reason to report the figure twice.

We repeated the same scheme in the **Extended data Figure 2d** and the associated main **Figure 2a** to help the reader to understand the experiments without switching from one figure to the other. We are happy to remove the ED figure panel if the reviewer thinks this is best.

- Page 15 – Fig d: I found unclear how the comparison (Y axis) and figure generated.

The Y-axis in **Extended Data Figure 4e** correspond to the quantification of the proportion of WT-YFP cells in mixed cultures with *Pik3ca*^{H1047R/wt} mutant cells after 28 days in competition with the indicated treatments, with respect to the proportion of WT-YFP cells found in sibling cultures at the beginning of the experiment. We have modified the figure legend with the aim to clarify the experimental setup (see below). We also have introduced a scheme of this experiment alongside the graph. We hope this will clarify the details of the protocol:

Reviewer Figure 12. Experimental scheme (top panel) and quantification by flow cytometry (lower panel) of the proportion of WT-RYFP cells mixed with *Pik3ca^{H1047R/wt}* cells at the end of the experiment versus the start of each experiment (lower panel). Cells were treated either in minimal FAD medium or treated with EGF at 10 ng/ml, and insulin (INS) at 2 ng/ml (physiological) or 5 µg/ml (supraphysiological) doses for 28 days. Each dot represents a primary culture from an animal (n=4-16 primary cultures from individual animals, per condition). Red lines indicate mean values. Two-tailed paired t-test. Source data are shown in Supplementary Table 1.

- Page 17 – Fig 4g: Venn diagram not representative of an 82% vs 18% overlap.

Following the Reviewer’s suggestion we have changed the Venn diagram in **Figure 4g** to make it more representative of the overlap:

Reviewer Figure 13. Venn diagram showing the proportion of genes up-regulated in *Pik3ca*^{H1047R/wt} cells which are also up-regulated by insulin treatment of wild-type cells.

- Figures arrangement with sections:

o Fig 5 – figures i, j,: should be included in fig 6 as they are not discussed or pertinent to this section.

Following this Reviewer suggestion, we have re-arranged the CRISPR screening results in a new Figure 5 and Extended Data Figure 5.

o Extended data fig 5 – figures d, e, f : should be included in extended fig 5 as they are not discussed or pertinent to this section.

See previous comment.

o There's an extended fig 8, but not a figure 8.

Unlike other journals, in *Nature Genetics* not every extended data figure is directly linked to one main data figure.

o Fig 7 g, h: should go under a separate fig 8 as they are not discussed or pertinent to this section and actually are relevant to extended fig 8 than 7.

Following this Reviewer's suggestion, we have placed the human DNA sequencing results in a new Figure 7 and Extended Data Figure 9.

o Similar observations for fig 3 and extended data fig 3 too.

We apologise for any confusion caused. These figures are where they are first cited in the text. It is true that both figures are relevant for two sections that describe different aspects of the same experiments, we can merge the sections into one if the Reviewer prefers it, however we thought that splitting these results into two sections would be clearer for the reader.

- Typos:

o Pag 14 line 3 – word missing between a level and than.

We removed these words to simplify the sentence as they were only aimed to explain that in EdU experiments we injected 1mg TAM. After this Reviewer comment, we realised that this was not properly specified in the methods section; therefore, we now extended this as follows:

“Analysis of *in vivo* proliferation by EdU labeling

Three months after induction with one intraperitoneal (i.p.) injection of 80 mg/kg β -naphthoflavone (MP Biomedicals 156738) and 1 mg tamoxifen (Sigma Aldrich N3633), 10 μ g of EdU in PBS was administered by intraperitoneal injection 1 h before culling. Tissues were collected and stained with EdU-Click-iT kit and immunofluorescence as explained below. EdU-positive basal cells were quantified from a minimum of 10 z-stack images. “

o Pag 31 line 16 – Typo – midel instead of middle.

We thank the Reviewer for spotting this typo, we have corrected it in the revised version of the manuscript.

REVIEWER #3 (expertise in cell competition)

The manuscript from Herms and Colum et al represents a thorough and important contribution to the body of work seeking to understand how homeostatic tissues cope with the frequent emergence of pre-malignant clones. Taking a cue from previously published analyses of patient samples, and using a suite of similar approaches to their previous work, the authors develop a mouse model to study activating mutations in *PI3KCA* in the esophagus. They find that cells heterozygous for *PI3KCA* mutations have a survival advantage over wild-type cells. Using mathematical modelling and lineage tracing experiments, they attribute this advantage to a skewing of fate dynamics towards progenitor self-renewal. Experiments largely conducted in a culture system allow them to use inhibitor treatments, RNA-sequencing studies, and CRISPR-screening approaches to determine that *PI3KCA* cells, due to upregulated mTOR/HIF signaling, ultimately shift to more glycolytic metabolism; this metabolic shift confers the growth advantage. All major results are then satisfyingly re-confirmed in vivo in the mouse model. Finally, in perhaps the most intriguing and novel part of the paper, they uncover links between metabolic diseases such as Type 1 diabetes and obesity, and heightened expansion of *PI3KCA* clones.

The mechanistic link between clonal competition, metabolism, and systemic disease is quite exciting and ultimately is what makes the paper worthy of publication in Nature Genetics.

I have only a few minor points:

1. The paper is very well written. However, for readers not familiar with the prior work of the Jones lab, it may be worth mentioning in the introduction that *Notch1* mutations have been analysed in other studies – it just jumps out so prominently as a candidate! But of course there is excellent rationale for also studying *PI3KCA*.

We thank the Reviewer for this comment; we have introduced the following text

“Analysis of published DNA-sequencing data identified 57 missense *PIK3CA* mutant clones in 17 cm² of histologically normal human esophageal epithelium (45). Of 72 cancer-related genes analyzed, missense mutations in *PIK3CA* had the second highest average variant allele fraction (VAF), indicating they form particularly large clones (**Fig. 1a and b**), only surpassed by *NOTCH1* inactivating mutations (previously demonstrated to provide a strong clonal advantage in the esophagus) (46). “

2. Although the data is very clean and clearly presented, I am a bit confused about the set-up of the culture system. Since the mouse model made in this study is *PI3KCA*-T2A-YFP, are not all the cells in the co-culture experiments with the RYFP-WT cells YFP+? I understand that because of the adenovirus transduction method the RYFP cells may be relatively brighter, but in the image the *PI3KCA* mutant cells look YFP-negative (and maybe they are coming from a different mouse model that doesn't have YFP?). This should

be somehow explained more clearly in the text (and I do wonder why in the first place R26-RFP wasn't used for WT cells! Would have been easier!).

As the Reviewer points out, both genetic mouse models (*Pik3ca*^{H1047R-T2A-YFP} and *Rosa26*^{RYFP}) express YFP after recombination. The rationale for the development of the PI3KCA-T2A-YFP construct was to use the YFP reporter to discriminate mutant cells from their wild-type neighbours. However, the level of expression of YFP from the *Pik3ca* locus in the esophagus is low, requiring immunostaining using anti-GFP antibodies (which also binds to YFP) for detection.

Due to the low expression of the YFP in mutant cells, it was convenient for us to perform the *in vitro* competition experiments using the WT-YFP strain which does not require anti-YFP immunostaining to be visualized by confocal microscopy. Furthermore, the WT-YFP strain had been extensively used in our group to characterize the clonal behaviour of esophageal progenitors *in vivo* and shown to be a neutral reporter (39). After recombination, WT-YFP cells, which express YFP from the *Rosa26* locus are about 4 orders of magnitude brighter than *PI3KCA*^{H1047R-T2A-YFP} cells which express YFP from the *Pik3ca* locus (see below histogram in panel **Extended Figure 4c**). Such a difference in fluorescence allows to easily differentiate them by FACS or immunofluorescence without staining. In the images presented, the *PI3KCA*^{H1047R-T2A-YFP} cells look negative because, although they express YFP its expression is undetectable with the settings used to detect WT-YFP cells.

Main text Extended Data Figure 4c. *In vitro* cell competition experimental protocol. *Pik3ca*^{H1047R/wt} and *Pik3ca*^{wt/wt} primary keratinocytes obtained as in **b** were mixed with *WT-RYFP* cells (as in **a**) and maintained at confluence. The proportion of *WT-RYFP* cells was quantified by flow cytometry. The proportion of *WT-RYFP* after treatment was normalized to the initial *WT-RYFP* cell proportion.

3. A major outstanding question is why does a switch to glycolytic metabolism confer a fitness advantage? Addressing this is outside the scope of the current manuscript, but more discussion, and a better synthesis of the literature on cell competition and glycolysis, is warranted. The fly papers cited should include the 2014 paper from Laura Johnson's lab (de la Cova et al, Cell Metabolism) and also the paper from Pascal Meier's group (Banreti & Meier, Nature Communications, 2019).

We are most grateful for the Reviewer's suggestions, we have now included the specified references in the results section:

"The connection between glycolysis and cell competition is controversial. Metabolic reprogramming towards glycolysis alters cell competitive fitness in *Drosophila* (17, 18). In mammalian keratinocytes, glycolysis has been linked to cell fate regulation and differentiation suggesting that glycolytic activation may confer a proliferative advantage in the esophagus (19-21).

4. I was initially troubled by the lack of earlier reference to the paper from the Beronja group on PI3KCA mutations in the skin (Ying et al, NCB, 2017), especially since in that case the result is almost opposite – that mutant clones shift their fate dynamics towards differentiation. When the paper is finally mentioned in the discussion, the authors here suggest that the different results perhaps can be explained by copy number (homozygous mutants in the case of the Ying paper). I appreciate that it would not be trivial for the authors to test this hypothesis, but a better attempt to reconcile these divergent results is warranted, at least by a more fleshed out discussion, if not by the addition of more data.

We now modified the discussion in light of this comment. Gene dosage is critical in defining the phenotype of signalling mutants.

"Overexpression of the *Pik3ca*^{H1047R} mutant and excessive activation of the PI3K/AKT/mTOR pathway in non-transformed cells may induce senescence, or differentiation in the skin (47-51) suggesting that the degree of activation of the PI3K pathway is critical for mutant cell fitness (52). These observations stress the importance of using mouse models that express heterozygous activating *Pik3ca* mutants from the endogenous promoter to model the corresponding mutants in human epithelia (8, 53)."

Response References

1. Herms A, Fernandez-Antoran D, Alcolea MP, Kalogeropoulou A, Banerjee U, Piedrafita G, et al. Epithelioids: Self-sustaining 3D epithelial cultures to study long-term processes. *bioRxiv*. 2023:2023.01.03.522589.
2. Yuan TL, Wulf G, Burga L, Cantley LC. Cell-to-cell variability in PI3K protein level regulates PI3K-AKT pathway activity in cell populations. *Current biology*. 2011;21(3):173-83.
3. Wilson MR, Reske JJ, Holladay J, Wilber GE, Rhodes M, Koeman J, et al. ARID1A and PI3-kinase pathway mutations in the endometrium drive epithelial transdifferentiation and collective invasion. *Nat Commun*. 2019;10(1):3554.
4. Deming DA, Leystra AA, Nettekoven L, Sievers C, Miller D, Middlebrooks M, et al. PIK3CA and APC mutations are synergistic in the development of intestinal cancers. *Oncogene*. 2014;33(17):2245-54.
5. Adams JR, Xu K, Liu JC, Agamez NMR, Loch AJ, Wong RG, et al. Cooperation between Pik3ca and p53 Mutations in Mouse Mammary Tumor Formation. *Cancer Research*. 2011;71(7):2706-17.
6. Gross SM, Rotwein P. Akt signaling dynamics in individual cells. *J Cell Sci*. 2015;128(14):2509-19.
7. Kinross KM, Montgomery KG, Mangiafico SP, Hare LM, Kleinschmidt M, Bywater MJ, et al. Ubiquitous expression of the Pik3caH1047R mutation promotes hypoglycemia, hypoinsulinemia, and organomegaly. *The FASEB Journal*. 2015;29(4):1426-34.
8. Kinross KM, Montgomery KG, Kleinschmidt M, Waring P, Ivetac I, Tikoo A, et al. An activating Pik3ca mutation coupled with Pten loss is sufficient to initiate ovarian tumorigenesis in mice. *The Journal of clinical investigation*. 2012;122(2):553-7.
9. Boucher J, Tseng Y-H, Kahn CR. Insulin and insulin-like growth factor-1 receptors act as ligand-specific amplitude modulators of a common pathway regulating gene transcription. *J Biol Chem*. 2010;285(22):17235-45.
10. Ilic N, Birsoy K, Aguirre AJ, Kory N, Pacold ME, Singh S, et al. *PIK3CA* mutant tumors depend on oxoglutarate dehydrogenase. *Proceedings of the National Academy of Sciences*. 2017;114(17):E3434-E43.
11. Lopez-Pajares V, Bhaduri A, Zhao Y, Gowrishankar G, Donohue L, Guo MG, et al. Glucose modulates transcription factor dimerization to enable tissue differentiation. *bioRxiv*. 2022:2022.11.28.518222.
12. Miao W, Porter DF, Lopez-Pajares V, Siprashvili Z, Meyers RM, Bai Y, et al. Glucose dissociates DDX21 dimers to regulate mRNA splicing and tissue differentiation. *Cell*. 2023;186(1):80-97.e26.
13. Dabelsteen S, Pallesen EMH, Marinova IN, Nielsen MI, Adamopoulou M, Rømer TB, et al. Essential Functions of Glycans in Human Epithelia Dissected by a CRISPR-Cas9-Engineered Human Organotypic Skin Model. *Developmental cell*. 2020;54(5):669-84.e7.

14. Hoxhaj G, Manning BD. The PI3K-AKT network at the interface of oncogenic signalling and cancer metabolism. *Nature reviews Cancer*. 2020;20(2):74-88.
15. Ricoult SJH, Yecies JL, Ben-Sahra I, Manning BD. Oncogenic PI3K and K-Ras stimulate de novo lipid synthesis through mTORC1 and SREBP. *Oncogene*. 2016;35(10):1250-60.
16. Lau CE, Tredwell GD, Ellis JK, Lam EW, Keun HC. Metabolomic characterisation of the effects of oncogenic PIK3CA transformation in a breast epithelial cell line. *Sci Rep*. 2017;7:46079.
17. Banreti AR, Meier P. The NMDA receptor regulates competition of epithelial cells in the *Drosophila* wing. *Nat Commun*. 2020;11(1):2228.
18. de la Cova C, Senoo-Matsuda N, Ziosi M, Wu DC, Bellosta P, Quinzii CM, et al. Supercompetitor status of *Drosophila* Myc cells requires p53 as a fitness sensor to reprogram metabolism and promote viability. *Cell Metab*. 2014;19(3):470-83.
19. Hamanaka RB, Mutlu GM. PFKFB3, a Direct Target of p63, Is Required for Proliferation and Inhibits Differentiation in Epidermal Keratinocytes. *J Invest Dermatol*. 2017;137(6):1267-76.
20. Sutter CH, Olesen KM, Bhujju J, Guo Z, Sutter TR. AHR Regulates Metabolic Reprogramming to Promote SIRT1-Dependent Keratinocyte Differentiation. *J Invest Dermatol*. 2019;139(4):818-26.
21. Cliff TS, Wu T, Boward BR, Yin A, Yin H, Glushka JN, et al. MYC Controls Human Pluripotent Stem Cell Fate Decisions through Regulation of Metabolic Flux. *Cell stem cell*. 2017;21(4):502-16.e9.
22. Hamanaka RB, Mutlu GM. PFKFB3, a Direct Target of p63, Is Required for Proliferation and Inhibits Differentiation in Epidermal Keratinocytes. *Journal of Investigative Dermatology*. 2017;137(6):1267-76.
23. Michelakis ED, Webster L, Mackey JR. Dichloroacetate (DCA) as a potential metabolic-targeting therapy for cancer. *Br J Cancer*. 2008;99(7):989-94.
24. James MO, Jahn SC, Zhong G, Smeltz MG, Hu Z, Stacpoole PW. Therapeutic applications of dichloroacetate and the role of glutathione transferase zeta-1. *Pharmacol Ther*. 2017;170:166-80.
25. Lingohr MK, Thrall BD, Bull RJ. Effects of Dichloroacetate (DCA) on Serum Insulin Levels and Insulin-Controlled Signaling Proteins in Livers of Male B6C3F1 Mice. *Toxicological Sciences*. 2001;59(1):178-84.
26. Yang L, Venneti S, Nagrath D. Glutaminolysis: A Hallmark of Cancer Metabolism. *Annual Review of Biomedical Engineering*. 2017;19(1):163-94.
27. Koundouros N, Poulogiannis G. Reprogramming of fatty acid metabolism in cancer. *British Journal of Cancer*. 2020;122(1):4-22.
28. Katayama Y, Kawata Y, Moritoh Y, Watanabe M. Dichloroacetate, a pyruvate dehydrogenase kinase inhibitor, ameliorates type 2 diabetes via reduced gluconeogenesis. *Heliyon*. 2022;8(2):e08889.

29. Foretz M, Guigas B, Viollet B. Metformin: update on mechanisms of action and repurposing potential. *Nature Reviews Endocrinology*. 2023;19(8):460-76.
30. Martin-Montalvo A, Mercken EM, Mitchell SJ, Palacios HH, Mote PL, Scheibye-Knudsen M, et al. Metformin improves healthspan and lifespan in mice. *Nat Commun*. 2013;4:2192.
31. Hopkins BD, Goncalves MD, Cantley LC. Insulin-PI3K signalling: an evolutionarily insulated metabolic driver of cancer. *Nat Rev Endocrinol*. 2020;16(5):276-83.
32. Wiernsperger NF, Bailey CJ. The antihyperglycaemic effect of metformin: therapeutic and cellular mechanisms. *Drugs*. 1999;58 Suppl 1:31-9; discussion 75-82.
33. Yoshioka M, Kayo T, Ikeda T, Koizumi A. A novel locus, Mody4, distal to D7Mit189 on chromosome 7 determines early-onset NIDDM in nonobese C57BL/6 (Akita) mutant mice. *Diabetes*. 1997;46(5):887-94.
34. Oyadomari S, Koizumi A, Takeda K, Gotoh T, Akira S, Araki E, et al. Targeted disruption of the Chop gene delays endoplasmic reticulum stress-mediated diabetes. *The Journal of clinical investigation*. 2002;109(4):525-32.
35. Han J-W, Zhan X-R, Li X-Y, Xia B, Wang Y-Y, Zhang J, et al. Impaired PI3K/Akt signal pathway and hepatocellular injury in high-fat fed rats. *World J Gastroenterol*. 2010;16(48):6111-8.
36. García-Prieto CF, Hernández-Nuño F, Rio DD, Ruiz-Hurtado G, Aránguez I, Ruiz-Gayo M, et al. High-fat diet induces endothelial dysfunction through a down-regulation of the endothelial AMPK-PI3K-Akt-eNOS pathway. *Mol Nutr Food Res*. 2015;59(3):520-32.
37. Turner N, Kowalski GM, Leslie SJ, Risis S, Yang C, Lee-Young RS, et al. Distinct patterns of tissue-specific lipid accumulation during the induction of insulin resistance in mice by high-fat feeding. *Diabetologia*. 2013;56(7):1638-48.
38. Frede J, Greulich P, Nagy T, Simons BD, Jones PH. A single dividing cell population with imbalanced fate drives oesophageal tumour growth. *Nature cell biology*. 2016;18(9):967-78.
39. Doupe DP, Alcolea MP, Roshan A, Zhang G, Klein AM, Simons BD, et al. A single progenitor population switches behavior to maintain and repair esophageal epithelium. *Science (New York, NY)*. 2012;337(6098):1091-3.
40. Li W, Xu H, Xiao T, Cong L, Love MI, Zhang F, et al. MAGeCK enables robust identification of essential genes from genome-scale CRISPR/Cas9 knockout screens. *Genome biology*. 2014;15(12):554.
41. Sorge S, Theelke J, Yildirim K, Hertenstein H, McMullen E, Müller S, et al. ATF4-Induced Warburg Metabolism Drives Over-Proliferation in Drosophila. *Cell reports*. 2020;31(7):107659.
42. Czech MP. Insulin action and resistance in obesity and type 2 diabetes. *Nat Med*. 2017;23(7):804-14.
43. Kuhn C, Hurwitz SA, Kumar MG, Cotton J, Spandau DF. Activation of the insulin-like growth factor-1 receptor promotes the survival of human keratinocytes following ultraviolet B irradiation. *International Journal of Cancer*. 1999;80(3):431-8.

44. Park S-Y, Cho Y-R, Kim H-J, Higashimori T, Danton C, Lee M-K, et al. Unraveling the Temporal Pattern of Diet-Induced Insulin Resistance in Individual Organs and Cardiac Dysfunction in c57bl/6 Mice. *Diabetes*. 2005;54(12):3530-40.
45. Martincorena I, Fowler JC, Wabik A, Lawson ARJ, Abascal F, Hall MWJ, et al. Somatic mutant clones colonize the human esophagus with age. *Science (New York, NY)*. 2018;362(6417):911-7.
46. Abby E, Dentre SC, Hall MWJ, Fowler JC, Ong SH, Sood R, et al. Notch1 mutations drive clonal expansion in normal esophageal epithelium but impair tumor growth. *Nature genetics*. 2023;55(2):232-45.
47. Ying Z, Sandoval M, Beronja S. Oncogenic activation of PI3K induces progenitor cell differentiation to suppress epidermal growth. *Nature cell biology*. 2018;20(11):1256-66.
48. Eser S, Reiff N, Messer M, Seidler B, Gottschalk K, Dobler M, et al. Selective Requirement of PI3K/PDK1 Signaling for Kras Oncogene-Driven Pancreatic Cell Plasticity and Cancer. *Cancer Cell*. 2013;23(3):406-20.
49. Jung SH, Hwang HJ, Kang D, Park HA, Lee HC, Jeong D, et al. mTOR kinase leads to PTEN-loss-induced cellular senescence by phosphorylating p53. *Oncogene*. 2019;38(10):1639-50.
50. Astle MV, Hannan KM, Ng PY, Lee RS, George AJ, Hsu AK, et al. AKT induces senescence in human cells via mTORC1 and p53 in the absence of DNA damage: implications for targeting mTOR during malignancy. *Oncogene*. 2012;31(15):1949-62.
51. Alimonti A, Nardella C, Chen Z, Clohessy JG, Carracedo A, Trotman LC, et al. A novel type of cellular senescence that can be enhanced in mouse models and human tumor xenografts to suppress prostate tumorigenesis. *The Journal of clinical investigation*. 2010;120(3):681-93.
52. Madsen RR, Knox RG, Pearce W, Lopez S, Mahler-Araujo B, McGranahan N, et al. Oncogenic PIK3CA promotes cellular stemness in an allele dose-dependent manner. *Proc Natl Acad Sci U S A*. 2019;116(17):8380-9.
53. Berenjeno IM, Pineiro R, Castillo SD, Pearce W, McGranahan N, Dewhurst SM, et al. Oncogenic PIK3CA induces centrosome amplification and tolerance to genome doubling. *Nat Commun*. 2017;8(1):1773.

Decision Letter, first revision:

31st May 2024

Dear Dr Jones,

Thank you for submitting your revised manuscript "Organismal metabolism regulates the expansion of oncogenic PIK3CA mutant clones in normal esophagus" (NG-A61642R). It has now been seen by the original referees and their comments are below. The reviewers find that the paper has improved in

revision, and therefore we'll be happy in principle to publish it in Nature Genetics, pending revisions to satisfy the referees' final requests and to comply with our editorial and formatting guidelines.

Sincerely,

Safia Danovi, PhD
Senior Editor, Nature Genetics
ORCID: 0009-0007-7822-5479

Reviewer #1 (Remarks to the Author):

The authors have made a good faith effort to address the concerns raised in the initial review, and the revised manuscript has been greatly strengthened. The authors should be applauded for adopting a more cautious approach in the revised manuscript in terms of mechanistic claims, how they interpret the data regarding metabolic dependencies, and in recognizing the intricate interplay of mechanisms by which DCA, metformin, HFD, and Akita influence mutant cell fitness in vivo. The acknowledgment of potential indirect effects, particularly of DCA on systemic insulin levels, adds depth to the discussion. While the systemic (on insulin signaling/sensitivity) versus direct effects (on esophageal cells) of these perturbations is not experimentally determined, this reviewer acknowledges that interpretable experiments to address this would not be feasible in the short-term. Nevertheless, there are a few remaining issues that the authors should address prior to publication.

1. While biochemical assessment of PI3K/insulin signaling in vivo in mutant/wild-type clones at baseline and following metabolic perturbation (metformin, HFD, DCA, Akita) would have further strengthened the study, this reviewer acknowledges and accepts the technical challenges presented by the authors and the possibility of limited sensitivity, background issues, and variability of immunofluorescence tissue studies. The authors to denote this as a limitation of their study in the Discussion.

2. The authors explored alternative metabolic pathways (e.g. glutamine, lipogenesis) regulated by PI3K based on additional CRISPR screens that support their claims. Through these studies, the authors introduce new data supporting a role for enhanced de novo lipogenesis in the fitness advantage afforded PIK3CA mutant cells. It is not clear, however, how the in vitro lipogenesis inhibitor studies were done. Lipogenic gene programs are highly sensitive to environmental lipid levels, it would be informative to culture the cells in a low-lipid environment (e.g., 1% FBS) and assess cellular fitness in the presence and absence of SREBP1, FASN, and ACC inhibitors, with and without insulin supplementation. TOFA is not an optimal inhibitor of ACC given the high doses needed to see an effect (30 μ M) and thus could

result in off-target effects. Other more potent ACC inhibitors are available including those that have been shown to be effective in vivo (e.g., ND-646; PMID 27643638). To validate the importance of lipogenesis in the increased cellular fitness, the authors should determine whether the fitness advantage can be rescued in cells treated with inhibitors of de novo lipogenesis via treatment of palmitate or oleate (given the importance of Scd2 in their cells). Collectively, these experiments could help substantiate the claim that increased de novo lipogenesis is determinant in the increased cellular fitness observed in PIK3CA mutant cells.

3. The additional analyses conducted to explore sex-specific differences in response to HFD treatment are appreciated. While the authors report no consistent sex-based trends in behavior, the inclusion of this analysis in the manuscript could offer valuable information to the field, given the well-documented sex differences in metabolic response. Therefore, it is suggested to include this analysis, or at least a summary of these findings, in the manuscript to provide a comprehensive view of your study's results.

Reviewer #2 (Remarks to the Author):

In this revision, the authors investigate the expansion of oncogenic PI3KCA clones in the oesphagus and how organismal metabolism regulates.

In the first first version, i have a number of significant concerns (both metholodogical and functional). However these have been tackled very well by the authors with a very considered and robust rebuttal which has very much improved the paper.

Therefore i believe this will be an important study.

Reviewer #3 (Remarks to the Author):

The revised manuscript from the Jones group is an impressive update to an already comprehensive study. The authors have clearly gone to great lengths to address the concerns raised by all three reviewers. The new figure 5 (and all associated extended data) is particularly commendable and raises many new questions for exciting follow up study. Without question, the paper will be of great interest to the field.

-typo in figure 3i in the spelling of "differentiating"

Author Rebuttal, first revision:

Response to reviewers: NG-A61642R

We are most grateful to the reviewers for the time and trouble they have taken to improve the paper with their thoughtful critique and positive suggestions. We respond to the reviewer comments below, our remarks are in blue.

REVIEWER #1: The authors have made a good faith effort to address the concerns raised in the initial review, and the revised manuscript has been greatly strengthened. The authors should be applauded for adopting a more cautious approach in the revised manuscript in terms of mechanistic claims, how they interpret the data regarding metabolic dependencies, and in recognizing the intricate interplay of mechanisms by which DCA, metformin, HFD, and Akita influence mutant cell fitness in vivo. The acknowledgment of potential indirect effects, particularly of DCA on systemic insulin levels, adds depth to the discussion. While the systemic (on insulin signaling/sensitivity) versus direct effects (on esophageal cells) of these perturbations is not experimentally determined, this reviewer acknowledges that interpretable experiments to address this would not be feasible in the short-term.

We are most grateful to the reviewer for their comments which resulted in a substantial revision of the text and recognition of the complexities of the metabolic effects that may be involved, especially in vivo.

Nevertheless, there are a few remaining issues that the authors should address prior to publication.

1. While biochemical assessment of PI3K/insulin signaling in vivo in mutant/wild-type clones at baseline and following metabolic perturbation (metformin, HFD, DCA, Akita) would have further strengthened the study, this reviewer acknowledges and accepts the technical challenges presented by the authors and the possibility of limited sensitivity, background issues, and variability of immunofluorescence tissue studies. The authors to denote this as a limitation of their study in the Discussion.

We agree that we have been unable to demonstrate changes in PI3K/Insulin signaling in mutant and wild type clones in esophageal epithelium directly. We have now discuss this as a limitation of the study in the discussion as follows:

“The limitations of the study include the lack of direct evidence for signaling differences between *Pik3ca*^{H1047R/wt} clones and wild type cells in vivo and showing that they are altered in the various metabolic states we studied. Changes induced by targeting oncogenic *Pik3ca* alleles to the endogenous allele in mice are difficult to detect with immunostaining, suggesting they are likely to be modest compared to those in *Pik3ca* mutant overexpression studies⁶³⁻⁶⁵. “

2. The authors explored alternative metabolic pathways (e.g. glutamine, lipogenesis) regulated by PI3K based on additional CRISPR screens that support their claims. Through these studies, the authors introduce new data supporting a role for enhanced de novo lipogenesis in the fitness advantage afforded PIK3CA mutant cells. It is not clear, however, how the in vitro lipogenesis inhibitor studies were done. Lipogenic gene programs are highly sensitive to environmental lipid levels, it would be informative to culture the cells in a low-lipid environment (e.g., 1% FBS) and assess cellular fitness in the presence and absence of SREBP1, FASN, and ACC inhibitors, with and without insulin supplementation. TOFA is not an optimal inhibitor of ACC given the high doses needed to see an effect (30 μ M) and thus could result in off-target effects. Other more potent ACC inhibitors are available including those that have been shown to be effective in vivo (e.g., ND-646; PMID 27643638). To validate the importance of lipogenesis in the increased cellular fitness, the authors should determine whether the fitness advantage can be rescued in cells treated with inhibitors of de novo lipogenesis via treatment of palmitate or oleate (given the importance of Scd2 in their cells). Collectively, these experiments could help substantiate the claim that increased de novo lipogenesis is determinant in the increased cellular fitness observed in PIK3CA mutant cells.

We have now added details to the methods section on how the lipogenesis inhibitor studies were done. We agree that an additional program of investigation of the metabolic consequences of expression of the PIK3CA mutant is desirable but would submit that this lies beyond the scope of this first report. We have added a comment on investigation of the effects of lipogenesis and other pathways to cell fitness to our comment on the limitations of the study in the discussion.

“More work is also needed to characterize the relative contributions of the different metabolic changes such as aerobic glycolysis and lipogenesis induced by the mutant allele, for example exploring the effects of different inhibitors and culture conditions on mutant fitness in vitro. “

3. The additional analyses conducted to explore sex-specific differences in response to HFD treatment are appreciated. While the authors report no consistent sex-based trends in behavior, the inclusion of this analysis in the manuscript could offer valuable information to the field, given the well-documented sex differences in metabolic response. Therefore, it is suggested to include this analysis, or at least a summary of these findings, in the manuscript to provide a comprehensive view of your study's results.

We now include a summary of these findings in the supplementary note (Supplementary Note Figure 2). We also commented them in the main text:

“However, HFD substantially altered mutant global clone size distribution, average clone size, and decreased the proportion of suprabasal cells in mutant clones, both in males and females (Fig. 6h, i and Extended Data Fig. 9l and m and Supplementary Figure 2).”

REVIEWER #3: The revised manuscript from the Jones group is an impressive update to an already comprehensive study. The authors have clearly gone to great lengths to address the concerns raised by all three reviewers. The new figure 5 (and all associated extended data) is particularly commendable and raises many new questions for exciting follow up study. Without question, the paper will be of great interest to the field.

We are most grateful to the reviewer for their positive assessment of the work.

-typo in figure 3i in the spelling of “differentiating”

We have corrected this

Final Decision Letter:

31st Jul 2024

Dear Dr Jones,

I am delighted to say that your manuscript "Organismal metabolism regulates the expansion of oncogenic PIK3CA mutant clones in normal esophagus" has been accepted for publication in an upcoming issue of Nature Genetics.

Due to the importance of these deadlines, we ask that you please let us know now whether you will be difficult to contact over the next month. If this is the case, we ask you provide us with the contact

information (email, phone and fax) of someone who will be able to check the proofs on your behalf, and who will be available to address any last-minute problems.

Your paper will be published online after we receive your corrections and will appear in print in the next available issue. You can find out your date of online publication by contacting the Nature Press Office (press@nature.com) after sending your e-proof corrections.

Please note that *Nature Genetics* is a Transformative Journal (TJ). Authors may publish their research with us through the traditional subscription access route or make their paper immediately open access through payment of an article-processing charge (APC). Authors will not be required to make a final decision about access to their article until it has been accepted. Find out more about Transformative Journals

Authors may need to take specific actions to achieve compliance with funder and institutional open access mandates. If your research is supported by a funder that requires immediate open access (e.g. according to Plan S principles) then you should select the gold OA route, and we will direct you to the compliant route where possible. For authors selecting the subscription publication route, the journal's standard licensing terms will need to be accepted, including <https://www.nature.com/nature-portfolio/editorial-policies/self-archiving-and-license-to-publish>. Those licensing terms will supersede any other terms that the author or any third party may assert apply to any version of the manuscript.

If you have not already done so, we strongly recommend that you upload the step-by-step protocols used in this manuscript to protocols.io. protocols.io is an open online resource that allows researchers to share their detailed experimental know-how. All uploaded protocols are made freely available and are assigned DOIs for ease of citation. Protocols can be linked to any publications in which they are used and will be linked to from your article. You can also establish a dedicated workspace to collect all your lab Protocols. By uploading your Protocols to protocols.io, you are enabling researchers to more readily reproduce or adapt the methodology you use, as well as increasing the visibility of your protocols and papers. Upload your Protocols at <https://protocols.io>. Further information can be found at <https://www.protocols.io/help/publish-articles>.

Sincerely,

Safia Danovi, PhD
Senior Editor, Nature Genetics
ORCID: 0009-0007-7822-5479